# GATED INFERENCE NETWORK: INFERENCING AND LEARNING STATE-SPACE MODELS

## ABSTRACT

State-space models (SSMs) perform predictions by learning the underlying dynamics of observed sequence. We propose a new SSM approach in both high and low dimensional observation space, which utilizes Bayesian filtering-smoothing to model system's dynamics more accurately than RNN-based SSMs and can be learned in an end-to-end manner. The designed architecture, which we call the *Gated Inference Network* (GIN), is able to integrate the uncertainty estimates and learn the complicated dynamics of the system that enables us to perform estimation and imputation tasks in both data presence and absence. The proposed model uses the GRU cells into its structure to complete the data flow, while avoids expensive computations and potentially unstable matrix inversions. The GIN is able to deal with any time-series data and gives us a strong robustness to handle the observational noise. In the numerical experiments, we show that the GIN reduces the uncertainty of estimates and outperforms its counterparts , LSTMs, GRUs and variational approaches.

## 1 INTRODUCTION

State estimation and inference in the states in dynamical systems is one of the most interesting problems that has lots of application in signal processing and time series Rauch et al. (1965). In some cases, learning state space is a very complicated task due to the relatively high dimension of observations and measurements, which only provides the partial information about the states. Noise is another significant issue in this scenario, where it is more likely to obtain a noisy observation. Time series prediction and estimating the next scene, e.g, the state prediction or next observation prediction, is another substantial application that again requires the inference within the states which comes from the observations. Classical memory networks such as LSTMs (Hochreiter & Schmidhuber, 1997), GRUs (Cho et al., 2014) and simple RNNs like (Wilson & Finkel, 2009) and (Yadaiah & Sowmya, 2006) fail to give some intuition about the uncertainties and dynamics. A group of approaches perform the Kalman Filtering (KF) in the latent state which usually requires a deep encoder for feature extraction. Krishnan et al. (2017), Ghalamzan et al. (2021) and Hashempour et al. (2020) belong to these group of works. However the mentioned solutions have some restrictions, where they are not able to deal with high dimensional non-linear systems and the classic KF approach is computationally expensive, e.g matrix inversion issue. Likewise, indirect optimization of an objective fuction by using variational inference, like the work of Kingma & Welling (2013), increases the complexity of the model. Moreover, in the variational inference approaches that usually implemented in the context of variational auto encoders for dimension reduction, they do not have access to the loss directly and have to minimize its lower bound instead, which reduce the ability of learning dynamics and affect the performance of the model. KalmanNet Revach et al. (2021) and Ruhe & Forré (2021) use GRU in their structure for the state update. However, they are only able to deal with low-dimensional state space and cannot handle complex high dimensional inputs because of directly using classic Bayesian equations and matrix inversion issue. Moreover, their structure require the full, or at least partial, dynamic information.

The mentioned restrictions for KF and its variants and variational models in addition the necessity of having a metric to measure the uncertainty, motivate us to introduce the GIN, an end to end structure with dynamics learning ability using Bayesian properties for filtering-smoothing. The contributions of GIN are: (i) modeling high-low dimensional sequences: we show the eligibility of the GIN to infer both cases by a simple adjustment in the observation transferring functions in the

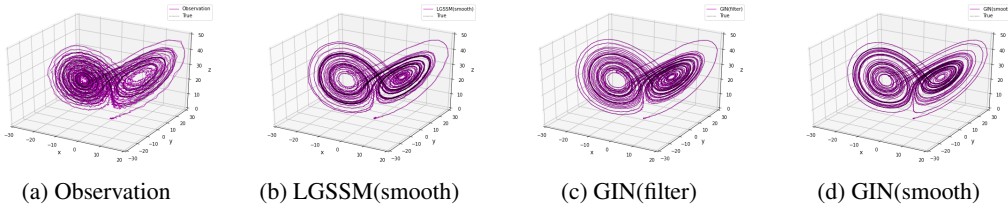

| (a) Observation | (b) LGSSM(smooth) | (c) GIN(filter) | (d) GIN(smooth) |

Figure 1: Inferred 5k length trajectories for Lorenz attractor.

proposed structure, where we conduct three experiments of high dimensional non-linear dynamics and two experiments of low dimensional chaotic observation. (ii) Learning/using dynamics: the ability of learning the dynamics(in the lack of them) and utilizing available dynamics(in the presence of them) alongside modeling high-low dimensional observations makes the GIN applicable to a wide range of applications. To attain more accurate inference of observed dynamical system, we apply GRU cells that increases the modeling capability of the Kalman filtering-smoothing. By conduction an ablation study of the GIN being replaced by a linearized Gaussian state transition, we show the GIN is able for better learning state space representation with disentangled dynamics features. (iii) Direct optimization: We show that the posterior and smoothing inference distribution of the state-space model is tractable while dynamics and parameters are estimated by neural networks. Despite variational approaches, this allows us to use recursive Bayesian updates for direct likelihood maximization. (iv) Noise robustness: verified by the numerical results, inferencing for highly distorted sequences is feasible with the GIN. (v) Missing data imputation: by using Bayesian properties, the GIN decides whether to keep the previous information in the memory cell or update them by the obtained observation. Experimental results show the out-performance of the GIN over the SOTA studies in the imputation task.

## 2 RELATED WORKS

To deal with complex sensory inputs, some approaches integrate a deep auto encoders into their architecture. Among these works, Embed to Control (E2C) (Watter et al., 2015) uses a deep encoder to obtain the observation and a variational inference about the states. However, these methods are not able to deal with missing data problem and imputation task since they do not rely on memory cells and are not recurrent. Another group of works like BackpropKF (Haarnoja et al., 2016) and RKN (Becker et al., 2019) apply CNNs for dimension-reduction and output both the uncertainty vector and observation, where they move away from variational inference and borrow Bayesian properties for the inference. However, these methods cannot handle the cases with the available knowledge of the dynamics and impose restrictive assumptions over covariance matrices, while the GIN provides a principled way for using the available partial dynamics information and release any assumption over covariance. Toward learning state space (system identification) a group of works like Wang et al. (2007), Ko & Fox (2011) and Frigola et al. (2013) propose algorithms to learn GPSSMs based on maximum likelihood estimation with the iterative EM algorithm. Frigola et al. (2013) obtain sample trajectories from the smoothing distribution, then conditioned on this trajectory they conduct M step for the model's parameters. Switching linear dynamics systems (SLDS) (Ghahramani & Hinton, 2000), use additional latent variables to switch among different linear dynamics, where the approximate inference algorithms can be utilized to model switching linearity for reducing approximation errors ,however, this approach is not as flexible as general non-linear dynamical systems because the switch transition model is assumed independent of states and observations. To address this problem, Linderman et al. (2017) performs SLDS method through augmentation with a Polya-gamma-distributed variable and a stick-breaking process, however, this approach employs Gibbs sampling for inferring the parameters and therefore is not scalable to large datasets. Auto-regressive Hidden Markov Models (ARHMM) explain time series structures by defining a mapping from past observations to the current observation. (Salinas et al., 2020) is a ARHMM approach, in which target values are used as inputs directly. However, this dependency of the model on the targets makes the model more vulnerable to noise. This issue is addressed in DSSM (Rangapuram et al., 2018), another ARHMM approach, where the target values are only incorporated through the likelihood term. Other group of works consider EM-based variational-inference like Structured Inference Networks (SIN) (Krishnan et al., 2017), where it utilizes a RNN to update the state. Kalman Variational Autoencoder (KVAE) (Fraccaro et al., 2017) and Extended KVAE (EKVAE) (Klushyn et al., 2021) use the original KF equations and apply both filtering and smoothing.

However, these EM-based variational inference methods are not able to estimate the states directly because of optimizing the lower bound of likelihood. Extra complexity is another issue with these approaches, while they are addressed by the proposed structure and direct end-to-end optimization in the GIN. We compare the GIN with these approaches in the experiment section and provide an empirical complexity analysis in appendix A.8.2. We provide a detailed discussion of recent related work in appendix A.8.3.

## 3 GATED INFERENCE NETWORK FOR SYSTEM IDENTIFICATION

In the context of System Identification (SI) the GIN is similar to a Hammerstein-Wiener (HW) model (Schoukens & Tiels, 2017) (Gilabert et al., 2005), in the sense that it estimates the system parameters directly from the observations, which is in the figure 2. $e(.)$ and $d(.)$ are implemented with non-linear functions, e.g. auto encoders-MLPs. Transition block in figure 2 represents the dynamics of the system that allows for the inference using the Gaussian state space filtering-smoothing equations. However unlike a HW model, we employ non-linear GRU cells in the transition block that calculate the Kalman Gain (KG) and smoothing gain (SG) in an appropriate manner by circumventing the computational complexity, i.e matrix inversion issues. GRU cells empower the whole system by applying non-linearity to the linearized Gaussian state space models (LGSSMs). Numerical results indicate that by the proposed structure, having a good inference for even the complex non-linear systems with high dimensional observations is feasible. To achieve this, we assume the state fits into Gaussian state space models (GSSMs), which are commonly used to model sequences of vectors.

In most cases, the dynamics of the system might not be available or hard to obtain; while the process noise and observation noise are unknown (our first three experiments). Accordingly, we construct GIN to learn unknown variables from data in an end to end fashion, then we utilize the constructed KG and SG during inference time to obtain the filtered-smoothed states. The proposed architecture is depicted in figure 4. In the presence of dynamics (our last two experiments), auto-encoder and *Dynamics Network* in figure 4 are replaced by MLP to model the observation noise.

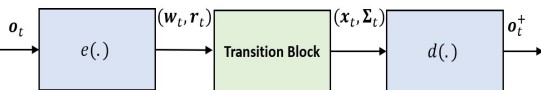

Figure 2: The GIN as a HW model for system identification. By appropriate structure selection for $e(.)$ and $d(.)$, the GIN can handle high-low dimensional observations. The proposed architectures for each case are depicted separately with further details in appendix figures 10 and 11. The relation between the internal variables, $\mathbf{w}_t$ and $\mathbf{x}_t$, is simulated by the transition block.

## 4 PARAMETERIZATION

In this section we show the parameterization of the inference model. Firstly, we refer the readers to the summary of Kalman filtering-smoothing background for the completeness in appendix A.2. In the rest of the paper we use the following notations, where the original observations are $\mathbf{o}_{1:T}$, the transferred observations are $\mathbf{w}_{1:T}$ and $\mathbf{R}_{1:T}$ are diagonal matrices with $\mathbf{r}_{1:T}$ diagonal elements that correspond to the covariance of transferred observation noise. $\mathbf{x}_{1:T}$ corresponds to the states and $\mathbf{Q}_{1:T}$ are diagonal matrices with $\mathbf{q}_{1:T}$ diagonal elements which are the covariance of state process noise. $(\mathbf{x}_t^-, \mathbf{\Sigma}_t^-)$ are the mean vector and covariance matrix of the prior state at time step $t$, i.e. $p(\mathbf{x}_t|\mathbf{w}_{1:t-1})$, and $(\mathbf{x}_t^+, \mathbf{\Sigma}_t^+)$ are the mean vector and covariance matrix of the posterior state at time step $t$, i.e. $p(\mathbf{x}_t|\mathbf{w}_{1:t})$. We define the transition matrices $\mathbf{F}_{1:T}$ and emission matrices $\mathbf{H}_{1:T}$ as the dynamics of the model, where lack of dynamics in the first



Figure 3: Graphical model for high dimensional observations. $dyn_t$ is the model's dynamics at time $t$.

three experiments means that $(\mathbf{F}_{1:T}, \mathbf{H}_{1:T})$ are not know and are going to be trained(graphical model is in figure 3), while the presence of dynamics in our last two experiments means that $(\mathbf{F}_{1:T}, \mathbf{H}_{1:T})$ are known(graphical models are in figures 9a and 9b). The dynamics $(\mathbf{F}_{1:T}, \mathbf{H}_{1:T})$ and noise matrices

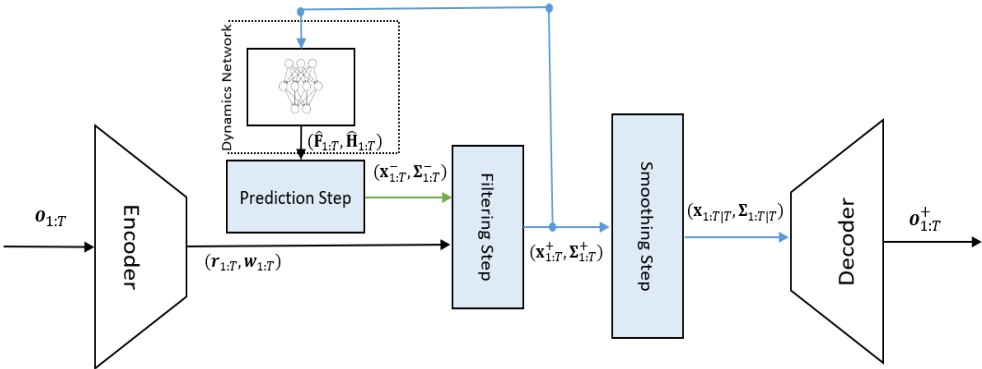

Figure 4: The high level structure of the GIN for high dimensional observation in the lack of dynamics, while for low dimensional cases auto-encoder is replaced by MLPs and dynamics are directly used. The transferred observation $\mathbf{w}_t$ and its uncertainty $\mathbf{r}_t$, are obtained from the encoder(MLPs). In each time step, the last posterior mean $\mathbf{x}_{t-1}^+$ is fed to the *Dynamic Network* to compute $\hat{\mathbf{F}}_t$ and $\hat{\mathbf{H}}_t$. In the *Prediction Step* the next priors $(\mathbf{x}_t^-, \boldsymbol{\Sigma}_t^-)$ are obtained by using new dynamics and the last posteriors. In the filtering step, by using the priors $(\mathbf{x}_t^-, \boldsymbol{\Sigma}_t^-)$ and the observation $(\mathbf{w}_t, \mathbf{r}_t)$, the next posteriors $(\mathbf{x}_t^+, \boldsymbol{\Sigma}_t^+)$ are obtained. Applying smoothing operation over the obtained posteriors $(\mathbf{x}_t^+, \boldsymbol{\Sigma}_t^+)$ is feasible in the smoothing step. Finally, the decoder(MLP) is utilized to produce $\mathbf{o}_t^+$, which can be the high-low dimensional noise free estimates.

$(\mathbf{R}_{1:T}, \mathbf{Q}_{1:T})$ construct the system parameters $\gamma_{1:T} = (\mathbf{F}_{1:T}, \mathbf{H}_{1:T}, \mathbf{Q}_{1:T}, \mathbf{R}_{1:T})$. Given original observations $\mathbf{o}_{1:T}$ and transferred observations $\mathbf{w}_{1:T}$, we want to find good estimate of the latent states $\mathbf{x}_{1:T}$. To achieve this, we want to infer the marginal distributions $p(\mathbf{x}_t|\mathbf{w}_{1:t})$ for the online inference approach or filtering; and $p(\mathbf{x}_t|\mathbf{w}_{1:T})$ for the full inference approach or smoothing.

We introduce an advantageous prediction parameterization as $p_{\gamma_t}(\mathbf{x}_t|\mathbf{x}_{t-1}, \mathbf{w}_{1:t-1}) = \mathcal{N}(\mathbf{F}_t\mathbf{x}_{t-1}, \mathbf{Q}_t)$, where $\mathbf{x}_{t-1} \sim p_{\gamma_{t-1}}(\mathbf{x}_{t-1}|\mathbf{w}_{1:t-1}) = \mathcal{N}(\mathbf{x}_{t-1}^+, \boldsymbol{\Sigma}_{t-1}^+)$. Then, $p_{\gamma_t}(\mathbf{x}_t|\mathbf{w}_{1:t-1}) = \mathcal{N}(\mathbf{F}_t\mathbf{x}_{t-1}^+, \mathbf{F}_t\boldsymbol{\Sigma}_{t-1}^+\mathbf{F}_t^T + \mathbf{Q}_t) = \mathcal{N}(\mathbf{x}_t^-, \boldsymbol{\Sigma}_t^-)$ is obtained by marginalizing out $\mathbf{x}_{t-1}$ and the Gaussianity of $p_{\gamma_t}(\mathbf{x}_t|\mathbf{w}_{1:t-1})$ results from the Gaussianity of prediction parameterization. By having $p_{\gamma_t}(\mathbf{x}_t|\mathbf{w}_{1:t-1})$ and observing $\mathbf{w}_t$, filtering parameterization is introduced as:

$$p_{\gamma_t}(\mathbf{x}_t|\mathbf{w}_{1:t}) = \mathcal{N}\left(\mathbf{x}_t^- + \mathbf{K}_t[\mathbf{w}_t - \mathbf{H}_t\mathbf{x}_t^-], \ \boldsymbol{\Sigma}_t^- - \mathbf{K}_t[\mathbf{H}_t\boldsymbol{\Sigma}_t^-\mathbf{H}_t^T + \mathbf{R}_t]\mathbf{K}_t^T\right) = \mathcal{N}(\mathbf{x}_t^+, \boldsymbol{\Sigma}_t^+) \ (1)$$

where $\mathbf{K}_t$ is KG. After observing all transferred observations $\mathbf{w}_{1:T}$, one can do backward induction and propagate to the previous states using the chain rule. This procedure, known as smoothing, can be parameterized as:

$$p_{\gamma_t}(\mathbf{x}_t|\mathbf{w}_{1:T}) = \mathcal{N}\left(\mathbf{x}_t^+ + \mathbf{J}_t[\mathbf{x}_{t+1|T} - \mathbf{F}_{t+1}\mathbf{x}_t^+], \ \boldsymbol{\Sigma}_t^+ + \mathbf{J}_t\left[\boldsymbol{\Sigma}_{t+1|T} - \boldsymbol{\Sigma}_{t+1}^-\right]\mathbf{J}_t^T\right) \quad (2)$$

where $\mathbf{J}_t$ is SG and we use short handed notation $\mathcal{N}(\mathbf{x}_{t|T}, \boldsymbol{\Sigma}_{t|T})$ instead of (2) . These parameterizations give us some insight to 1-illustrate a tractable way to construct $p_\gamma(\mathbf{x}|\mathbf{w})$ and accordingly obtain the posterior and smoothened states, based on which $\mathbf{o}^+$ is constructed and 2- appropriately modeling $\gamma$ and KG(SG) with neural networks.

To construct the KG and SG networks, we have to find appropriate inputs containing related information to attain the KG and SG. In (1) and (2), KG and SG are given by (3) and (4), respectively.

$$\mathbf{K}_t = \boldsymbol{\Sigma}_t^-\mathbf{H}_t^T.\left[\mathbf{H}_t\boldsymbol{\Sigma}_t^-\mathbf{H}_t^T + \mathbf{R}_t\right]^{-1} \propto (\boldsymbol{\Sigma}_t^-, \mathbf{R}_t) \qquad (3)$$

$$\mathbf{J}_t = \boldsymbol{\Sigma}_t^+\mathbf{F}_{t+1}^T.\left[\mathbf{F}_{t+1}\boldsymbol{\Sigma}_t^+\mathbf{F}_{t+1}^T + \mathbf{Q}_{t+1}\right]^{-1} = \boldsymbol{\Sigma}_t^+\mathbf{F}_{t+1}^T\boldsymbol{\Sigma}_{t+1}^- \propto \boldsymbol{\Sigma}_{t+1}^- \qquad (4)$$

(3) is proportional to the prior covariance at time $t$, $\boldsymbol{\Sigma}_\mathbf{t}^-$, and the observation noise matrix, $\mathbf{R}_\mathbf{t}$, while (4) is proportional to prior covariance matrix at time $t+1$, $\boldsymbol{\Sigma}_{t+1}^-$. Our encoder(MLP) directly maps the observation noise matrix from the observation space, but the state covariance is a recursive function of previous states. Consequently, we consider $GRU^{KG}$ and $GRU^{SG}$ which are networks including

GRU that map $[\mathbf{f}(\boldsymbol{\Sigma_t^-}), \mathbf{R_t}]$ and $\mathbf{f}(\boldsymbol{\Sigma_{t+1}^-})$ to the KG and SG, respectively. $GRU^{KG}$ considers $\mathbf{R}_t$, a diagonal matrix with $r_t$ elements in figure 4, as a part of its input to incorporate the effects of observation noise. In the case of high dimensional state space, due to the high dimension of $\boldsymbol{\Sigma_t^-}$ and $\boldsymbol{\Sigma_{t+1}^-}$, $\mathbf{f}$ is a convolutional layer with pooling to extract the valuable information of the covariance matrix that reduces its size, while for the low dimension of $\boldsymbol{\Sigma_t^-}$ and $\boldsymbol{\Sigma_{t+1}^-}$, $\mathbf{f}$ is the identity function.

**Learning The Process Noise.** In the filtering procedure, the process noise in time $t$ is obtained as

$$\mathbf{Q}_t = \boldsymbol{\Sigma}_t^- - \mathbf{F}_t \boldsymbol{\Sigma}_{t-1}^+ \mathbf{F}_t^T. \tag{5}$$

where $\boldsymbol{\Sigma}_t^-$, $\mathbf{F}_t$ and $\boldsymbol{\Sigma}_{t-1}^+$ are prior state covariance, transition matrix and posterior state covariance at time $t$. It is shown that $\mathbf{Q}_t$ can be written as a function of $\mathbf{F}_t$ as

$$\mathbf{Q}_t = \boldsymbol{\Sigma}_t^- - \mathbf{F}_t \boldsymbol{\Sigma}_{t-1}^+ \mathbf{F}_t^T = \boldsymbol{\Sigma}_t^- - \mathbf{F}_t \big[ \boldsymbol{\Sigma}_{t-1}^- - \mathbf{K}_{t-1}[\mathbf{H}_{t-1} \boldsymbol{\Sigma}_{t-1}^- \mathbf{H}_{t-1}^T + \mathbf{R}_{t-1}]^{-1} \mathbf{K}_{t-1}^T \big] \mathbf{F}_t^T \tag{6}$$

while the derivations are rather lengthy, therefore, we refer to the appendix materials A.3. From (32), the relation of the process noise with the transition matrix indicates that $\mathbf{F}_t$ can possess the effects of $\mathbf{Q}_t$ if we learn it in an appropriate manner. $\hat{\mathbf{F}}_t(\mathbf{Q}_t)$ notation means that the learned transition matrix $\hat{\mathbf{F}}_t$ comprises the effects of $\mathbf{Q}_t$, while for the simplicity we use $\hat{\mathbf{F}}_t$ abbreviation. Therefore, it is possible to rewrite (5) as

$$\boldsymbol{\Sigma}_t^- = \hat{\mathbf{F}}_t \boldsymbol{\Sigma}_{t-1}^+ \hat{\mathbf{F}}_t^T. \tag{7}$$

Another way to have a a meaningful inference about the process noise matrix is to obtain it from (30) as a recursive function of $\mathbf{x}_{t-1}^+$ and $\mathbf{Q}_{t-1}$. Intuitively, $\mathbf{g}$ function in (30) that we call it *Q Network*, can be implemented by a memory cell, e.g., a GRU cell, to keep the past status of $\mathbf{Q}$ ,however, it increases the complexity of the model. Equivalently, one can obtain $\mathbf{Q}_t$ directly from $\mathbf{x}_{t-1}^+$ with MLP as stated in (32). Both of these solutions can be utilized when the dynamics are known, i.e. we cannot learn the effects of $\mathbf{Q}_t$ jointly with $\mathbf{F}_t$ as $\mathbf{F}_t$ is not trainable.

**Prediction Step.** Similar to the model based Kalman Filter, by using dynamics of the system and transition, the next priors are obtained from the current posterior by

$$\mathbf{x}_t^- = \hat{\mathbf{F}}_t \mathbf{x}_{t-1}^+ , \qquad \boldsymbol{\Sigma}_t^- = \hat{\mathbf{F}}_t \boldsymbol{\Sigma}_{t-1}^+ \hat{\mathbf{F}}_t^T \tag{8}$$

where $\hat{\mathbf{F}}_t$ is the learned transition matrix comprises the effects of the process noise from previous section. By which, it is feasible to predict state mean and the state covariance matrix.

**Filtering Step.** To obtain the next posteriors based on the new observation $(\mathbf{w}_t, \mathbf{r}_t)$, i.e. the output of $e(.)$ in figure 2, we have to use the obtained KG matrix from $GRU^{KG}$ network and learned emission matrix $\hat{\mathbf{H}}_t$ to complete updating the state mean vector and state covariance matrix. This procedure is given by

$$\mathbf{S}_t^- = \hat{\mathbf{H}}_t . \boldsymbol{\Sigma}_t^- . \hat{\mathbf{H}}_T^T + \mathbf{R}_t, \qquad \mathbf{K}_t = \boldsymbol{\Sigma}_t^- \hat{\mathbf{H}}_t^T \mathbf{M}_t \mathbf{M}_t^T, \qquad \mathbf{M}_t = GRU^{KG}(\mathbf{f}(\boldsymbol{\Sigma}_t^-), \mathbf{R}_t) \tag{9}$$

$$\mathbf{x}_t^+ = \mathbf{x}_t^- + \mathbf{K}_t . [\mathbf{w}_t - \hat{\mathbf{H}}_t \mathbf{x}_t^-], \qquad \boldsymbol{\Sigma}_t^+ = \boldsymbol{\Sigma}_t^- + \mathbf{K}_t . \mathbf{S}_t^- . \mathbf{K}_t^T. \tag{10}$$

In addition to avoiding the matrix inversion that arises in the computation of Kalman gain and applying non-linearity to handle more complex dynamics, the architecture of KG network, $GRU^{KG}$, can reduce the dimension of the input to its corresponding GRU cell, and thus reduces the total amount of parameters quadratically. Additionally, positive $\mathbf{r_t}$ vector and Cholesky factor consideration, $\mathbf{M}_t \mathbf{M}_t^T$ in (9), guarantee the positive definiteness of the resulted covariance matrices.

**Smoothing Step.** After obtaining filtered states $(\mathbf{x}_{1:T}^+, \boldsymbol{\Sigma}_{1:T}^+)$ in filtering step, we employ smoothing properties of Bayesian to get smoothed version of the states. In this stage, we use $\mathbf{J}_{1:T}$ matrices obtained from $GRU^{SG}$ network, learned transition matrices $\hat{\mathbf{F}}_{1:T}$ and filtered states $(\mathbf{x}_{1:T}^+, \boldsymbol{\Sigma}_{1:T}^+)$. The procedure in each smoothing step is given by:

$$\mathbf{J}_t = \boldsymbol{\Sigma}_t^+ \hat{\mathbf{F}}_{t+1}^T \mathbf{N}_t \mathbf{N}_t^T, \qquad \mathbf{N}_t = GRU^{SG}(\mathbf{f}(\boldsymbol{\Sigma}_{t+1}^-)) \tag{11}$$

$$\mathbf{x}_{t|T} = \mathbf{x}_t^+ + \mathbf{J}_t \big[ \mathbf{x}_{t+1|T} - \hat{\mathbf{F}}_{t+1} \mathbf{x}_t^+ \big], \qquad \boldsymbol{\Sigma}_{t|T} = \boldsymbol{\Sigma}_t^+ + \mathbf{J}_t \big( \boldsymbol{\Sigma}_{t+1|T} - \hat{\mathbf{F}}_{t+1} \boldsymbol{\Sigma}_t^+ \hat{\mathbf{F}}_{t+1}^T \big) \mathbf{J}_t^T \tag{12}$$

where the first smoothing state is set to the last filtering state, i.e. $(\mathbf{x}_{T|T}, \boldsymbol{\Sigma}_{T|T}) = (\mathbf{x}_T^+, \boldsymbol{\Sigma}_T^+)$ .

**Learning Dynamics.** We can model the dynamics in each time step $t$ as a function of the transferred observations $\mathbf{w}_{1:t-1}$. However, conditioning on the noisy observations can distort the procedure of learning the dynamics. Instead, we use the state $\mathbf{x}_{t-1}^+$ in GSSM that includes the history of the system with considerable lower noise distortion to increase system's noise robustness, where we generate time correlated noise in our experiments to show this robustness(see appendix A.5). In other words, original transition and emission equations, $\mathbf{x}_t = f(\mathbf{x}_{t-1}) + \mathbf{q}_t$ and $\mathbf{w}_t = h(\mathbf{x}_t) + \mathbf{r}_t$, are modeled as $\mathbf{x}_t = \hat{\mathbf{F}}_t(\mathbf{x}_{t-1}^+)\mathbf{x}_{t-1} + \mathbf{q}_t$ and $\mathbf{w}_t = \hat{\mathbf{H}}_t(\mathbf{x}_{t-1}^+)\mathbf{x}_t + \mathbf{r}_t$. We learn $K$ state transition and emission matrices $\hat{\mathbf{F}}^k$ and $\hat{\mathbf{H}}^k$, and combine each one with the state dependent coefficient $\alpha^k(\mathbf{x}_{t-1}^+)$.

$$\hat{\mathbf{F}}_t(\mathbf{x}_{t-1}^+) = \sum_{k=1}^{K} \alpha_t^k(\mathbf{x}_{t-1}^+)\hat{\mathbf{F}}_t^k, \quad \hat{\mathbf{H}}_t(\mathbf{x}_{t-1}^+) = \sum_{k=1}^{K} \alpha_t^k(\mathbf{x}_{t-1}^+)\hat{\mathbf{H}}_t^k \tag{13}$$

where a separated neural network with softmax output is utilized to learn $\alpha^k(\mathbf{x}_{t-1}^+)$ that we call it *Dynamics Network*. This formulation enables us to follow Bayesian methodology. Despite classic LGSSMs that are not able to learn the dynamics, e.g. EKF and UKF, the trainable dynamics in the GIN are function of the states. For the notation simplicity, we have used $(\hat{\mathbf{F}}_t, \hat{\mathbf{H}}_t)$ instead of $(\hat{\mathbf{F}}_t(\mathbf{x}_{t-1}^+), \hat{\mathbf{H}}_t(\mathbf{x}_{t-1}^+))$ in the paper.

## 5 FITTING

For the state estimation task, by implementing $p(\mathbf{w}_{1:T}|\mathbf{o}_{1:T})$, $p(\mathbf{x}_t|\mathbf{w}_{1:T})$ and $p(\mathbf{s}_t|\mathbf{x}_t)$ with encoder, smoothing parameterization and decoder, we maximise the log-likelihood of output $p(\mathbf{s}_t|\mathbf{o}_{1:T}) = \int p(\mathbf{s}_t|\mathbf{x}_t)p(\mathbf{x}_t|\mathbf{w}_{1:T})p(\mathbf{w}_{1:T}|\mathbf{o}_{1:T})d\mathbf{x}_t d\mathbf{w}_{1:T}$ ,where $\mathbf{s}_t$ is the estimated state, i.e. equal to $\mathbf{o}_t^+$ in figure 4. For the image imputation task, in addition to the state likelihood, we add the reconstruction pseudo-likelihood for inferring images by using Bernoulli distributions as $p(\mathbf{i}_t|\mathbf{x}_t)$, i.e. the decoder in figure 4 maps both state $\mathbf{s}_t$ and image $\mathbf{i}_t : \mathbf{o}_t^+ = [\mathbf{i}_t, \mathbf{s}_t]$. Further details of distribution assumptions and hyper parameter optimization can be found in the appendix A.4 and A.8. After training phase, forecasting desired number of time steps is applicable by plugging the new value $\mathbf{x}_t = \hat{\mathbf{F}}_t\mathbf{x}_{t-1}$ recursively in the model, and so on.

**Likelihood for Inferring States.** Consider the ground truth sequence is defined as $\mathbf{s}_{1:T}$. We determine the log likelihood of the states as:

$$\mathcal{L}(\mathbf{s}_{1:T}) = \sum_{t=1}^{T} \log \mathcal{N}\left(\mathbf{s_t} \Big| \mathrm{dec}_{\mathrm{mean}}(\mathbf{x_{t|T}}), \mathrm{dec}_{\mathrm{covar}}(\mathbf{\Sigma_{t|T}})\right) \tag{14}$$

where the $\mathrm{dec}_{\mathrm{mean}}(.)$ and $\mathrm{dec}_{\mathrm{covar}}(.)$ determines those parts of the decoder that are used to obtain the state mean and state variance, respectively. We use Wishart distribution as a prior for our estimated covariance matrix, which pushes the estimated covariance toward a scale of identity matrix and the scale is a hyper parameter. Such prior prevents getting high log-likelihood due to the high uncertainty.

**Likelihood for inferring images.** For the imputation task, consider the ground truth as the sequence of images and their corresponding states, which are defined as $[\mathbf{s}_{1:T}, \mathbf{i}_{1:T}]$ and the dimension of $\mathbf{i}_t$ is $D_o$. We determine the log likelihood:

$$\mathcal{L}(\mathbf{o}_{1:T}^+) = \mathcal{L}(\mathbf{s}_{1:T}) + \lambda \sum_{t=1}^{T} \sum_{k=0}^{D_o} \mathbf{i}_t^{(k)} \log\left(\mathrm{dec}_k(\mathbf{x_{t|T}})\right) + \left(1 - \mathbf{i}_t^{(k)}\right) \log(1 - \mathrm{dec}_k(\mathbf{x_{t|T}})). \tag{15}$$

$\mathrm{dec}_k(\mathbf{x}_t)$ defines the corresponding part of the decoder that maps the $k$-th pixel of $\mathbf{i}_t$ image and $\lambda$ constant determines the importance of the reconstruction. The first term in RHS is obtained from (14) and we abbreviate the second term as $\mathcal{L}(\mathbf{i}_{1:T})$.

## 6 EVALUATION AND EXPERIMENTS

We divide our experiments into two parts, first the tasks in which the observation space is high dimensional like sequence of images, and second the applications that the observation is in low dimension by itself so there is no need to include encoder for dimension reduction. The training algorithms of both cases are added in the appendix section A.11.

Table 1: Double pendulum state estimation. $(x_1, x_3)$ refers to the position of the first joint, while $(x_2, x_4)$ is for the second joint.

| Model | $SE^{Pos}_{x_1}$ | $SE^{Pos}_{x_3}$ | $SE^{Pos}_{x_2}$ | $SE^{Pos}_{x_4}$ | Log Likelihood |
|---|---|---|---|---|---|
| LSTM (units=50) | 0.163 | 0.171 | 0.148 | 0.167 | $3.901 \pm 0.706$ |
| LSTM (units=100) | 0.154 | 0.147 | 0.134 | 0.152 | $4.053 \pm 0.565$ |
| GRU (units=50) | 0.189 | 0.183 | 0.179 | 0.177 | $3.886 \pm 0.369$ |
| GRU (units=100) | 0.164 | 0.156 | 0.162 | 0.145 | $3.976 \pm 0.231$ |
| KVAE ($n$=2$m$) | 0.193 | 0.188 | 0.178 | 0.149 | $3.679 \pm 0.101$ |
| KVAE ($n$=3$m$) | 0.171 | 0.159 | 0.151 | 0.162 | $3.801 \pm 0.116$ |
| RKN ($n$=2$m$) | 0.134 | 0.129 | 0.139 | 0.118 | $4.176 \pm 0.294$ |
| LGSSM$_{filter}$($n$=3$m$) | 0.125 | 0.119 | 0.121 | 0.107 | $4.192 \pm 0.127$ |
| LGSSM$_{smooth}$($n$=3$m$) | 0.109 | 0.111 | 0.104 | 0.101 | $4.231 \pm 0.154$ |
| GIN$_{filter}$($n$=2$m$) | 0.115 | 0.109 | 0.119 | 0.109 | $4.224 \pm 0.105$ |
| GIN$_{filter}$($n$=3$m$) | 0.093 | 0.091 | 0.098 | 0.089 | $4.329 \pm 0.151$ |
| GIN$_{smooth}$($n$=2$m$) | 0.091 | 0.104 | 0.101 | 0.092 | $4.308 \pm 0.123$ |
| GIN$_{smooth}$($n$=3$m$) | **0.079** | **0.083** | **0.085** | **0.077** | **$4.477 \pm 0.168$** |

Table 2: Pendulum state estimation. By consider $n = 3m$, intuitively the last part of the state is dedicated to the acceleration information causing a more lieklihood.

| Model | $SE^{Pos}_{x_1}$ | $SE^{Pos}_{x_2}$ | Log Likelihood |
|---|---|---|---|
| LSTM (units=25) | 0.092 | 0.094 | $5.891 \pm 0.151$ |
| LSTM (units=100) | 0.089 | 0.087 | $5.751 \pm 0.215$ |
| GRU (units=30) | 0.095 | 0.089 | $5.986 \pm 0.168$ |
| GRU (units=100) | 0.091 | 0.089 | $5.698 \pm 0.205$ |
| KVAE ($n$=2$m$) | 0.104 | 0.095 | $5.786 \pm 0.098$ |
| KVAE ($n$=3$m$) | 0.088 | 0.093 | $5.858 \pm 0.113$ |
| RKN ($n$=2$m$) | 0.078 | 0.075 | $6.161 \pm 0.23$ |
| LGSSM$_{filter}$ | 0.077 | 0.073 | $6.211 \pm 0.265$ |
| LGSSM$_{smooth}$ | 0.071 | 0.069 | $6.242 \pm 0.109$ |
| GIN$_{filter}$($n$=2$m$) | 0.073 | 0.07 | $6.192 \pm 0.239$ |
| GIN$_{filter}$($n$=3$m$) | 0.067 | 0.066 | $6.315 \pm 0.220$ |
| GIN$_{smooth}$($n$=2$m$) | 0.065 | 0.067 | $6.292 \pm 0.173$ |
| GIN$_{smooth}$($n$=3$m$) | **0.059** | **0.057** | **$6.445 \pm 0.165$** |

## 6.1 HIGH DIMENSIONAL OBSERVATION WITH LACK OF DYNAMICS

We include three high dimensional experiments. The first two experiments are single pendulum and double pendulum, where the dynamics of the latter one is more complicated. The last experiment is visual odometry task. Intuitive python code in appendix A.12.1.

### 6.1.1 SINGLE PENDULUM AND DOUBLE PENDULUM

The inputs of the encoder are the images with size of $24 \times 24$. The angular velocity is disturbed by transition noise which follows Normal distribution with $\sigma = 0.1$ as its standard deviation at each step. In the pendulum experiment, we perform the filtering-smoothing by the GIN where the observation is distorted with high observation noise. Furthermore, we compare GIN with LGSSM, where the GRU cells are omitted from the GIN structure and classic filtering-smoothing equations are used, instead. The log-likelihood and squared error (SE) of positions for single and double pendulum are given in table 2 and 1, respectively. Generated samples from trained smooth-filter distributions are in appendix figures 16-33.

By randomly deleting the half of images from the generated sequences, we conduct the image imputation task to our model by predicting those missing parts, while the missingness applied to train and test are not same, but random. The results are in table 3 and 4.The GIN outperforms all the other models, although the variational inference models have more complex structures in KAVE

Table 3: Image imputation task for the different models. Models contain boolean masks determining the available and missed images. For uninformed masks, a black image is considered as the input of the cell whenever the image is missed, which requires the model to infer the accurate dynamics for the generation. We conduct uninformed experiment as well.

| Model | Log Likelihood |
|---|---|
| E2C | $-95.539 \pm 1.754$ |
| SIN | $-101.268 \pm 0.567$ |
| KVAE (informed smooth) | $-14.217 \pm 0.236$ |
| KVAE (unformed smooth) | $-39.260 \pm 5.399$ |
| EKVAE (informed smooth) | $-12.897 \pm 0.524$ |
| EKVAE (unformed smooth) | $-29.246 \pm 3.328$ |
| RKN (informed) | $-12.782 \pm 0.0160$ |
| RKN (uninformed) | $-12.788 \pm 0.0142$ |
| LGSSM(informed smooth) | $-12.695 \pm 0.048$ |
| GIN (informed smooth) | $-12.215 \pm 0.027$ |
| GIN (unformed smooth) | $-12.246 \pm 0.029$ |

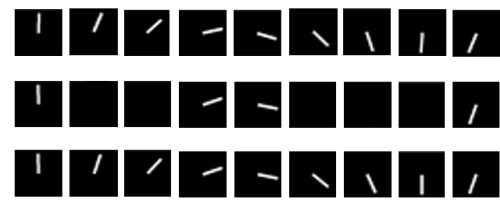

Figure 5: Pendulum image imputation. Each figure, beginning from up to down, indicates the ground truth, uninformed observation and the imputation results of the GIN(smoothed). Missingness is applied randomly for train and test.

Table 4: Image imputation for double pendulum.

| Model | Log Likelihood |
|---|---|
| KVAE (informed smooth) | -15.917 ± 0.294 |
| KVAE (unformed smooth) | -38.544 ± 6.419 |
| EKVAE (informed smooth) | -13.917 ± 0.414 |
| EKVAE (unformed smooth) | -33.548 ± 4.516 |
| RKN (informed) | -13.832 ± 0.023 |
| RKN (uninformed) | -13.898 ± 0.0191 |
| LGSSM(informed smooth) | -13.775 ± 0.013 |
| GIN (informed smooth) | -13.284 ± 0.021 |
| GIN (unformed smooth) | -13.351 ± 0.019 |

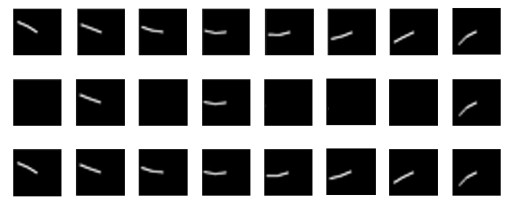

Figure 6: Double pendulum image imputation. Each figure, beginning from up to down, indicates the ground truth, uninformed observation and the imputation results of the GIN(smoothed).

and EKVAE. We include the results using the MSE as well, to illustrate that our approach is also competitive in prediction accuracy (See A.10).

### 6.1.2 VISUAL ODOMETRY OF KITTI DATASET

We also evaluate the GIN with the higher dimensional observations for the visual odometry task on the KITTI dataset Geiger et al. (2012). This dataset consists of 11 separated image sequences with their corresponding labels. In order to extract the positional features, we use a feature extractor network proposed by Zhou et al. in Zhou et al. (2017). The obtained features are considered as the observations of the GIN, i.e. $(\mathbf{w}, \mathbf{r})$. Additionally, we compare the results with LSTM, GRU, DeepVO Wang et al. (2017) and KVAE. The results are in table 5 and figure 8, where the common evaluation scheme for the KITTI dataset is exploited. The results of the KVAE degrades substantially as we have to reduce the size of the transferred observation to prevent the complexity of matrix inversion.

### 6.2 LOW DIMENSIONAL OBSERVATION WITH PRESENCE OF DYNAMICS

We conduct two experiments, Lorenz attractor problem and the real world dynamics NCLT dataset, where we are aware of the dynamics. Intuitive python code in appendix A.12.2. However, the GIN is able to deal the cases in which the dynamics are not known. To show this, we conduct additional experiment in the appendix, where we do not give the dynamics information of Lorenz attractor and NCLT dataset to the model, so that they will be learned(see figures 42-46).

### 6.2.1 LORENZ ATRRACTOR

The Lorenz system is a system of ordinary differential equations that describes a non-linear dynamic system used for atmospheric convection. Due to nonlinear dynamics of this chaotic system (see A.6), it can be a good evaluation for the GIN cell. We evaluate the performance of the GIN on a trajectory of 5k length. Each point in the generated trajectories is distorted with an observation noise that follows Gaussian distribution with standard deviation $\sigma = 0.5$. The likelihood with Gaussian distribution is calculated and maximized in the training phase. The mean square error (MSE) of the test data for various number of train-

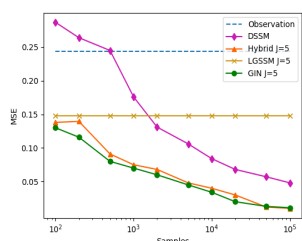

Figure 7: MSE of Lorenz attractor.

Table 5: Comparison of model performance on KITTI dataset. See 34, 35, 36, 37, 38 and 39 figures in A.9 for the visualization results.

| Seq | LSTM | | GRU | | DeepVO | | KVAE | | LGSSM | | GIN | |
|---|---|---|---|---|---|---|---|---|---|---|---|---|
| | $t_{rel}(\%)$ | $r_{rel}(°)$ | $t_{rel}(\%)$ | $r_{rel}(°)$ | $t_{rel}(\%)$ | $r_{rel}(°)$ | $t_{rel}(\%)$ | $r_{rel}(°)$ | $t_{rel}(\%)$ | $r_{rel}(°)$ | $t_{rel}(\%)$ | $r_{rel}(°)$ |
| 03 | 8.99 | 4.55 | 9.34 | 3.81 | 8.49 | 6.89 | 12.14 | 4.38 | 7.51 | 3.98 | 6.98 | 3.27 |
| 04 | 11.88 | 3.44 | 12.36 | 2.89 | 7.19 | 6.97 | 13.17 | 4.73 | 9.12 | 2.64 | 9.14 | 2.28 |
| 05 | 8.96 | 3.43 | 10.02 | 3.43 | 2.62 | 3.61 | 11.47 | 5.14 | 6.11 | 3.21 | 4.38 | 2.51 |
| 06 | 9.66 | 2.8 | 10.99 | 3.22 | 5.42 | 5.82 | 10.93 | 3.98 | 6.70 | 3.51 | 6.14 | 2.90 |
| 07 | 9.83 | 5.48 | 13.70 | 6.52 | 3.91 | 4.60 | 12.73 | 4.68 | 6.59 | 3.49 | 7.21 | 2.98 |
| 10 | 13.58 | 3.49 | 13.37 | 3.25 | 8.11 | 8.83 | 14.79 | 10.91 | 9.32 | 2.90 | 8.37 | 2.59 |
| mean | 10.53 | 3.87 | 11.63 | 3.85 | 5.96 | 6.12 | 12.53 | 5.63 | 7.55 | 3.28 | 7.03 | 2.75 |

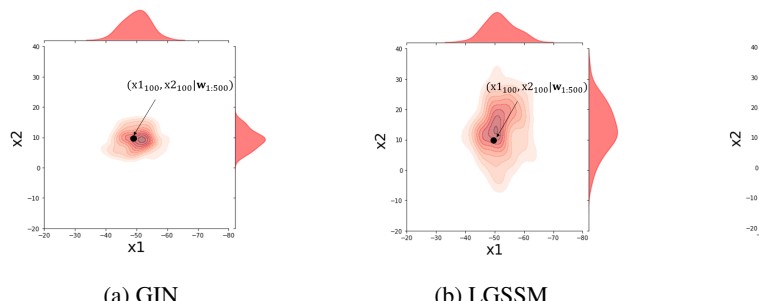

(a) GIN  (b) LGSSM  (c) KVAE

Figure 8: Generated samples from smoothing distribution for the joint position $(x1, x2)$, equivalent to $(o1^+, o2^+)$ in figure 4, at 100-th time step of visual odometry experiment. The ground truth is shown with a black point.

ing samples are depicted in figure 7. *Hybrid* is a graphical GNN based model Garcia Satorras et al. (2019) and DSSM Rangapuram et al. (2018) is a version of LGSSM using LSTM cells.

Due to the non-linearity of the dynamics of this system, LGSSM has to use linearization and then use the linearized dynamics to model the transition. The DSSM model performs better for lager amount of data (>10K) because it needs to learn the dynamics. The results of the Hybrid GNN and the GIN are similar, while the results of the GIN are slightly improved. Although, the core of both models is based on the GRU cell, this enhancement may come from the structure of the GIN that learns the observation and process noises separately. Inferred trajectories are in figure 1.

### 6.2.2 REAL WORLD DYNAMICS: MICHIGAN NCLT DATASET

To evaluate the performance of the GIN on a real world dataset, the Michigan NCLT dataset Carlevaris-Bianco et al. (2016) is utilized that encompasses a collection of navigation data gathered by a segwey robot moving inside of the University of Michigan's North Campus. The states in each time, $\mathbf{x}_t \in \mathbb{R}^4$, comprise the position and the velocity in each direction and the observations, $\mathbf{y}_t \in \mathbb{R}^2$, include noisy positions. The ultimate purpose is to localize the real position of the segway robot, while only the noisy GPS observations are available. We apply the GIN to find the current location of the segway robot.

In this experiment, we randomly select the session 2012-01-22 captured in a cloudy situation with the length of 6.1 Km. By sampling with 1Hz and removing the unstable GPS observations, 4280 time steps are achieved. For the dynamics of the system, we consider a uniform motion pattern with a constant velocity (see A.7). The training procedure is completed by maximizing the likelihood with Gaussian distribution assumption. The mean squared error of each approach for the test set are mentioned in the table 6, where the GIN ($73.12 \pm 2.21$ MSE) outperforms other approaches. In summary, this experiment indicates that the GIN can generalize with good performance to a real world dataset.

Table 6: MSE for NCLT experiment.

| Model | MSE[dB] |
|---|---|
| GIN(smooth) | 18.64±0.13 |
| Hybrid GNN | 20.73± 0.21 |
| KalmanNet | 22.2±0.17 |
| DSSM | 29.54±0.58 |
| Vanilla RNN | 40.21±0.52 |
| LGSSM | 24.38±0.17 |
| Observation | 25.47±0.08 |

## 7 CONCLUSION

The GIN, an approach for representation learning in both high and low dimensional SSMs, is introduced in this paper. The data flow is conducted by Bayesian filtering-smoothing, while, due to the usage of GRU based KG and SG network, the computational issues are tackled resulting in an efficient model with numerical stable results. In the presence of the dynamics, the GIN directly use them, otherwise it directly learns them in an end to end manner, which makes the GIN as a HW model with strong system identification abilities. Insightful representation for the uncertainty of the predictions is incorporated in this approach, while it outperforms its counterparts including LSTMs, GRUs and several generative models with variational inferences.

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

# A  APPENDIX

## A.1  GAUSSIAN STATE SPACE MODELS

In order to model the vectors of time series $\mathbf{w} = \mathbf{w}_{1:T} = [\mathbf{w}_1, ..., \mathbf{w}_T]$, Gaussian state space models (GSSMs) are commonly applied due to their filtering-smoothing ability. In fact, GSSMs model the first-order Markov process on the state space $\mathbf{x} = [\mathbf{x}_1, ..., \mathbf{x}_T]$, which can also include the external control input $\mathbf{u} = [\mathbf{u}_1, ..., \mathbf{u}_T]$ by multivariate normality assumption of the state

$$p_{\gamma_t}(\mathbf{x}_t|\mathbf{x}_{t-1}, \mathbf{u}_t) = \mathcal{N}(\mathbf{x}_t; \mathbf{F}_t\mathbf{x}_{t-1} + \mathbf{B}_t\mathbf{u}_t, \mathbf{Q}), \quad p_{\gamma_t}(\mathbf{w}_t|\mathbf{x}_t) = \mathcal{N}(\mathbf{w}_t; \mathbf{H}_t\mathbf{z}_t, \mathbf{R}). \quad (16)$$

For the cases, which are not controlled via external input, $\mathbf{B}_t$ matrix is simply $\mathbf{0}$ matrix. By Defining $\gamma_t$ as parameters which explain how the state state changes during the time, it contains the information of $\mathbf{F}_t, \mathbf{B}_t$ and $\mathbf{H}_t$ which are the state transition, control and emission matrices. In each step, the procedure is distorted via $\mathbf{Q}$ and $\mathbf{R}$ that are process noise and observation noise, respectively. It is common to initial the first state $\mathbf{x}_1 \sim \mathcal{N}(\mathbf{0}, \mathbf{\Sigma}_0)$, then the joint probability distribution of the GSSM is

$$p_\gamma(\mathbf{w}, \mathbf{x}|\mathbf{u}) = p_\gamma(\mathbf{w}|\mathbf{x})p_\gamma(\mathbf{x}|\mathbf{u}) = \prod_{t=1}^{T} p_{\gamma_t}(\mathbf{w}_t|\mathbf{x}_t).p(\mathbf{x}_1) \prod_{t=2}^{T} p_{\gamma_t}(\mathbf{x}_t|\mathbf{x}_{t-1}, \mathbf{u}_t). \quad (17)$$

GSSMs have substantial properties that we can utilize. Filtering and smoothing are among these properties which allow us to obtain the filtered and smoothed posterior based on the priors and observations. By applying classic Bayesian properties, we can have a strong tool to handle the missing data in the image imputation task.

## A.2  FILTERING AND SMOOTHING PARAMETERIZATION

The idea of Kalman filter applies two iterative steps, in the former one a prediction is made by the prior state information, while in the latter one an update is done based on the obtained observation. By normality assumption of known additive process and observation noise, the filter can go through the two mentioned steps. In the prediction step, the filter uses the transition matrix $\mathbf{F}$ to estimate the next priors $(\mathbf{x}_{t+1}^-, \mathbf{\Sigma}_{t+1}^-)$ which are the estimate of the the next states without any observation.

$$\mathbf{x}_{t+1}^- = \mathbf{F}\mathbf{x}_t^+, \text{and} \quad \mathbf{\Sigma}_{t+1}^- = \mathbf{F}\mathbf{\Sigma}_t^+\mathbf{F}^T + \mathbf{Q}, \text{and} \quad \mathbf{Q} = \sigma_{\text{trans}}^2\mathbf{I} \quad (18)$$

In the presence of new observation, the Kalman filter idea goes through the second step and modifies the predicted prior based on the new observation and emission matrix $\mathbf{H}$ that results in the next posterior $(\mathbf{x}_{t+1}^+, \mathbf{\Sigma}_{t+1}^+)$.

$$\mathbf{K}_{t+1} = \mathbf{\Sigma}_{t+1}^-\mathbf{H}^T(\mathbf{H}\mathbf{\Sigma}_{t+1}^-\mathbf{H}^T + \mathbf{R})^{-1}, \text{and} \quad (19)$$

$$\mathbf{x}_{t+1}^+ = \mathbf{x}_{t+1}^- + \mathbf{\Sigma}_{t+1}^-\mathbf{H}^T(\mathbf{H}\mathbf{\Sigma}_{t+1}^-\mathbf{H}^T + \mathbf{R})^{-1}(\mathbf{w}_t - \mathbf{H}\mathbf{x}_{t+1}^-) = \mathbf{x}_{t+1}^- + \mathbf{K}_{t+1}(\mathbf{w}_t - \mathbf{H}\mathbf{x}_{t+1}^-), \quad (20)$$

$$\mathbf{\Sigma}_{t+1}^+ = \mathbf{\Sigma}_{t+1}^- - \mathbf{\Sigma}_{t+1}^-\mathbf{H}^T(\mathbf{H}\mathbf{\Sigma}_{t+1}^-\mathbf{H}^T + \mathbf{R})^{-1}\mathbf{H}\mathbf{\Sigma}_{t+1}^-. \quad (21)$$

The whole observation update procedure can be considered as a weighted mean between the the next prior, that comes from state update, and new observation, where this weighting is a function of $\mathbf{Q}$ and $\mathbf{R}$ that has uncertainty nature.

We derive smoothing parameterization, where the key idea is to use Markov property, which states that $\mathbf{x}_t$ is independent of future observations $\mathbf{w}_{t+1:T}$ as long as $\mathbf{x}_{t+1}$ is known. However, we are not

aware of $\mathbf{x}_{t+1}$, but there is a distribution over it. So by conditioning on $\mathbf{x}_{t+1}$ and then marginalizing out it is possible to obtain $\mathbf{x}_t$ conditioned on $\mathbf{w}_{1:T}$.

$$
\begin{aligned}
p(\mathbf{x}_t|\mathbf{w}_{1:T}) &= \int p(\mathbf{x}_t|\mathbf{x}_{t+1}, \mathbf{w}_{1:T})p(\mathbf{x}_{t+1}|\mathbf{w}_{1:T})d\mathbf{x}_{t+1} \\
&= \int p(\mathbf{x}_t|\mathbf{x}_{t+1}, \mathbf{w}_{1:t}, \cancel{\mathbf{w}_{t+1:T}})p(\mathbf{x}_{t+1}|\mathbf{w}_{1:T})d\mathbf{x}_{t+1}
\end{aligned}
\tag{22}
$$

By using induction and and smoothed distribution for $t+1$:

$$
p(\mathbf{x}_{t+1}|\mathbf{w}_{1:T}) = \mathcal{N}(\mathbf{x}_{t+1|T}, \boldsymbol{\Sigma}_{t+1|T})
\tag{23}
$$

we calculate the filtered two-slice distribution as follows:

$$
.p(\mathbf{x}_t, \mathbf{x}_{t+1}|\mathbf{w}_{1:t}) = \mathcal{N}\left(\begin{pmatrix} \mathbf{x}_t^+ \\ \mathbf{x}_{t+1}^- \end{pmatrix}, \begin{pmatrix} \boldsymbol{\Sigma}_t^+ & \boldsymbol{\Sigma}_t^+\mathbf{F}_{t+1}^T \\ \mathbf{F}_{t+1}\boldsymbol{\Sigma}_t^+ & \boldsymbol{\Sigma}_{t+1}^- \end{pmatrix}\right)
\tag{24}
$$

by using Gaussian conditioning we have:

$$
p(\mathbf{x}_t|\mathbf{x}_{t+1}, \mathbf{w}_{1:t}) = \mathcal{N}(\mathbf{x}_t^+ + \mathbf{J}_t(\mathbf{x}_{t+1} - \mathbf{F}_{t+1}\mathbf{x}_t^+), \boldsymbol{\Sigma}_t^+ - \mathbf{J}_t\boldsymbol{\Sigma}_{t+1}^-\mathbf{J}_t^T)
\tag{25}
$$

where $\mathbf{J}_t = \boldsymbol{\Sigma}_t^+\mathbf{F}_{t+1}[\boldsymbol{\Sigma}_{t+1}^-]^{-1}$. We calculate the smoothed distribution for $t$ using the rules of iterated expectation and covariance:

$$
\begin{aligned}
\mathbf{x}_{t|T} &= \mathbb{E}\big[\mathbb{E}[\mathbf{x}_t|\mathbf{x}_{t+1}, \mathbf{w}_{1:T}]\,|\mathbf{w}_{1:T}\big] \\
&= \mathbb{E}\big[\mathbb{E}[\mathbf{x}_t|\mathbf{x}_{t+1}, \mathbf{w}_{1:t}]\,|\mathbf{w}_{1:T}\big] \\
&= \mathbb{E}\big[\mathbf{x}_t^+ + \mathbf{J}_t(\mathbf{x}_{t+1} - \mathbf{F}_{t+1}\mathbf{x}_t^+)\,|\mathbf{w}_{1:T}\big] \\
&= \mathbf{x}_t^+ + \mathbf{J}_t(\mathbf{x}_{t+1|T} - \mathbf{F}_{t+1}\mathbf{x}_t^+)
\end{aligned}
\tag{26}
$$

$$
\begin{aligned}
\boldsymbol{\Sigma}_{t|T} &= \text{cov}\big[\mathbb{E}[\mathbf{x}_t|\mathbf{x}_{t+1}, \mathbf{w}_{1:T}]\,|\mathbf{w}_{1:T}\big] + \mathbb{E}\big[\text{cov}[\mathbf{x}_t|\mathbf{x}_{t+1}, \mathbf{w}_{1:T}]\,|\mathbf{w}_{1:T}\big] \\
&= \text{cov}\big[\mathbb{E}[\mathbf{x}_t|\mathbf{x}_{t+1}, \mathbf{w}_{1:t}]\,|\mathbf{w}_{1:T}\big] + \mathbb{E}\big[\text{cov}[\mathbf{x}_t|\mathbf{x}_{t+1}, \mathbf{w}_{1:t}]\,|\mathbf{w}_{1:T}\big] \\
&= \text{cov}\big[\mathbf{x}_t^+ + \mathbf{J}_t(\mathbf{x}_{t+1} - \mathbf{F}_{t+1}\mathbf{x}_t^+)\,|\mathbf{w}_{1:T}\big] + \mathbb{E}\big[\boldsymbol{\Sigma}_t^+ - \mathbf{J}_t\boldsymbol{\Sigma}_{t+1}^-\mathbf{J}_t^T\,|\mathbf{w}_{1:T}\big] \\
&= \mathbf{J}_t\text{cov}\big[\mathbf{x}_{t+1} - \mathbf{F}_{t+1}\mathbf{x}_t^+\,|\mathbf{w}_{1:T}\big]\mathbf{J}_t^T + \boldsymbol{\Sigma}_t^+ - \mathbf{J}_t\boldsymbol{\Sigma}_{t+1}^-\mathbf{J}_t^T \\
&= \mathbf{J}_t\boldsymbol{\Sigma}_{t+1|T}\mathbf{J}_t^T + \boldsymbol{\Sigma}_t^+ - \mathbf{J}_t\boldsymbol{\Sigma}_{t+1}^-\mathbf{J}_t^T \\
&= \boldsymbol{\Sigma}_t^+ + \mathbf{J}_t(\boldsymbol{\Sigma}_{t+1|T} - \boldsymbol{\Sigma}_{t+1}^-)\mathbf{J}_t^T.
\end{aligned}
\tag{27}
$$

### A.3 PROCESS NOISE MATRIX

As stated in (18), we can elaborate the process noise matrix at time $t$ in more details

$$
\mathbf{Q}_t = \boldsymbol{\Sigma}_t^- - \mathbf{F}_t\boldsymbol{\Sigma}_{t-1}^+\mathbf{F}_t^T = \boldsymbol{\Sigma}_t^- - \mathbf{F}_t\big[\boldsymbol{\Sigma}_{t-1}^- - \mathbf{K}_{t-1}[\mathbf{H}_{t-1}\boldsymbol{\Sigma}_{t-1}^-\mathbf{H}_{t-1}^T + \mathbf{R}_{t-1}]^{-1}\mathbf{K}_{t-1}^T\big]\mathbf{F}_t^T
\tag{28}
$$

combining (18) into (28) results in

$$
\begin{aligned}
\mathbf{Q}_t = \boldsymbol{\Sigma}_t^- - \mathbf{F}_t\big[[\mathbf{F}_{t-1}\boldsymbol{\Sigma}_{t-2}^+\mathbf{F}_{t-1}^T + \mathbf{Q}_{t-1}] \\
- \mathbf{K}_{t-1}[\mathbf{H}_{t-1}[\mathbf{F}_{t-1}\boldsymbol{\Sigma}_{t-2}^+\mathbf{F}_{t-1}^T + \mathbf{Q}_{t-1}]\mathbf{H}_{t-1}^T + \mathbf{R}_{t-1}]^{-1}\mathbf{K}_{t-1}^T\big]\mathbf{F}_t^T
\end{aligned}
\tag{29}
$$

which is a function of $\mathbf{F}_t$, $\mathbf{Q}_{t-1}$, $\mathbf{F}_{t-1}$ and $\mathbf{H}_{t-1}$. In the GIN, $\hat{\mathbf{F}}_t$ and $\hat{\mathbf{H}}_t$ are learned by the *Dynamics Network* with the input of $\mathbf{x}_{t-1}^+$. From (20), $\mathbf{x}_{t-1}^+$ is derived as a function of both $\mathbf{F}_{t-1}$ and $\mathbf{H}_{t-1}$,

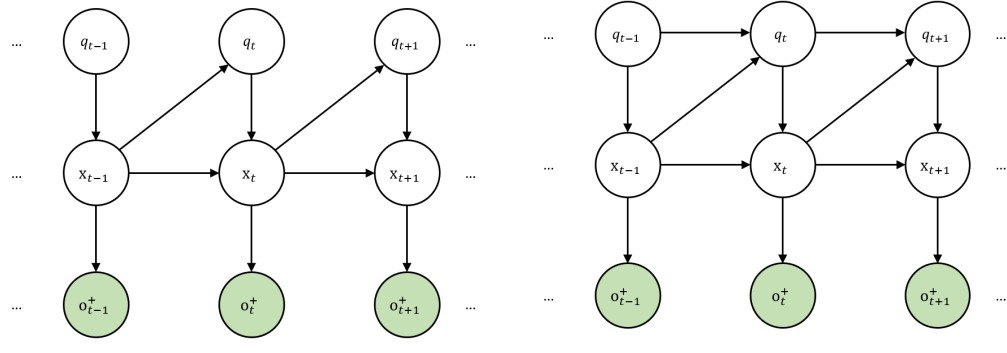

(a) Without recurrent dependency on $\mathbf{q}_t$.      (b) With recurrent dependency on $\mathbf{q}_t$.

Figure 9: Graphical models for low dimensional observations experiments.

meaning the learned $\hat{\mathbf{F}}_t$ carries the information of both $\mathbf{H}_{t-1}$ and $\mathbf{F}_{t-1}$. Therefore, one can rewrite the equation (29) as

$$\mathbf{Q}_t = \mathbf{g}\left(\hat{\mathbf{F}}_t\left(\mathbf{x}_{t-1}^+\right), \mathbf{Q}_{t-1}\right), \text{where} \quad \hat{\mathbf{F}}_t = \textit{Dynamics Network}\left(\mathbf{x}_{t-1}^+\left(\mathbf{H}_{t-1}, \mathbf{F}_{t-1}\right)\right). \tag{30}$$

where $\mathbf{g}$ is a nonlinear function mapping $\mathbf{x}_{t-1}^+$ and $\mathbf{Q}_{t-1}$ to $\mathbf{Q}_t$ and the graphical model for such choice of structure is in figure 9b. It is possible to go one step further and simplify $\mathbf{x}_{t-1}^+$ more, as it has $\mathbf{\Sigma}_{t-1}^-$ term in (20), combining it with (18) results in

$$\mathbf{x}_{t-1}^+ = \mathbf{x}_{t-1}^- + [\mathbf{F}_{t-1}\mathbf{\Sigma}_{t-2}^+\mathbf{F}_{t-1}^T + \mathbf{Q}_{t-1}]$$
$$\mathbf{H}_{t-1}^T\left(\mathbf{H}_{t-1}[\mathbf{F}_{t-1}\mathbf{\Sigma}_{t-2}^+\mathbf{F}_{t-1}^T + \mathbf{Q}_{t-1}]\mathbf{H}_{t-1}^T + \mathbf{R}_{t-1}\right)^{-1}(\mathbf{w}_t - \mathbf{H}_{t-1}\mathbf{x}_{t-1}^-) \tag{31}$$

indicating that not only $\mathbf{F}_{t-1}$ and $\mathbf{H}_{t-1}$, but also $\mathbf{Q}_{t-1}$ is included in $\mathbf{x}_{t-1}^+$, meaning that $\mathbf{Q}_t$ can be written solely as a function of $\mathbf{x}_{t-1}^+$ and the graphical model for such choice is in figure 9a.

$$\mathbf{Q}_t = \mathbf{g}\left(\hat{\mathbf{F}}_t\left(\mathbf{x}_{t-1}^+\right)\right), \text{where} \quad \hat{\mathbf{F}}_t = \textit{Dynamics Network}\left(\mathbf{x}_{t-1}^+\left(\mathbf{H}_{t-1}, \mathbf{F}_{t-1}, \mathbf{Q}_{t-1}\right)\right). \tag{32}$$

We call $\mathbf{g}$ as *Q Network*, where $\mathbf{g}$ can be modeled by a MLP (32) or a recurrent network (30), based on the mentioned explanations. In figure 11, it is shown how the *Q Network* is integrated into the whole model structure.

## A.4 OUTPUT DISTRIBUTION

In the case of grayscale images, consider each pixel, $y_i$, is one or zero with the probability of $p_i$ or $1 - p_i$ respectively, meaning that $P(Y = y) = p^y(1-p)^{1-y}$. By re-writing the probability equation into the exponential families form

$$f_\theta(y) = h(y).exp\left(\theta.y - \psi(\theta)\right) \rightarrow e^{log(p^y(1-p)^{1-y})} = e^{y\,log(\frac{p}{1-p}) + log(1-p)} \tag{33}$$

and by choosing $\theta = log(\frac{p}{1-p})$ and $\psi(\theta) = log(1-p)$, we can obtain $p = \frac{1}{1+e^{-\theta}}$. It means that by considering $\theta$ as the last layer of the decoder and applying a softmax layer, $p$ is obtained. Equivalently, one can calculate the deviance between real $p$ and estimation of it, $\hat{p}$, which is given by

$$D(p, \hat{p}) = \left[p\,log(\frac{p}{\hat{p}}) + (1-p)log(\frac{1-p}{1-\hat{p}})\right] \tag{34}$$

and minimize the deviance with respect to $\hat{p}$ as we did in (15).

Similarly, consider $x$, $\hat{x}_\theta$ and $\theta$ as the ground truth state, estimated state and the model variables respectively, where the residual follows Gaussian distribution $x = \hat{x}_\theta + \epsilon \sim \mathcal{N}(\hat{x}_\theta, \hat{\sigma}_\theta)$, where $\hat{\sigma}_\theta$ is the estimated variance. Then, the negative log likelihood is given by (35) as we obtained it in (14).

$$-log(\mathcal{L}) \propto \frac{1}{2}log(\hat{\sigma}_\theta) + \frac{(x - \hat{x}_\theta)^2}{2\hat{\sigma}_\theta} \tag{35}$$

### A.5   NOISE GENERATION PROCESS

In the high dimensional observation experiments, to show the noise robustness of the system, we use time correlated noise generation scheme. It makes the noise factors correlated over time by introducing a sequence of factors $f_t$ of the same length of the data sequence. Let $f_0 \sim \mathcal{U}(0,1)$ and $f_{t+1} = \min(\max(0, f_t + r_t), 1)$ with $r_t \sim \mathcal{U}(-0.2, 0.2)$, where $f_0$ is the initialized factor and $\mathcal{U}$ is the uniform distribution. Then by defining two thresholds, $t_1 \sim \mathcal{U}(0, 0.25)$ and $t_2 \sim \mathcal{U}(0.75, 1)$, $f_t < t_1$ are set to 0 and $f_t > t_2$ are set to 1 and the rest are splitted linearly within the range of $[0, 1]$. The $t$-th obtained observation is given by $\mathbf{o}_t = f_t \mathbf{i}_t + (1 - f_t)\mathbf{i}_t^{pn}$, where the $\mathbf{i}_t$ is the $t$-th true image and $\mathbf{i}_t^{pn}$ is the $t$-th generated pure noise.

### A.6   LORENZ ATTRACTOR DYNAMICS

There are three differential equations that model a Lorenz system, $x$ the convection rate, $y$ the horizontal temperature variation and $z$ the vertical temperature variation.

$$\frac{dx}{dt} = \sigma(y - x), \quad \frac{dy}{dt} = x(\rho - z) - y, \quad \frac{dz}{dt} = xy - \beta z \tag{36}$$

where the constant values $\sigma$, $\rho$ and $\beta$ are $10$, $28$ and $-\frac{8}{3}$, respectively. To construct a trajectory we use Lorenz system equations (36) with $dt = 10^{-5}$, then we sample from it with the step time of $\Delta t = 0.01$.

Based on the equations of the system (36), the state is $\mathbf{s_t} = [x_t, y_t, z_t]$ and we can write the dynamics of the system as $\mathbf{A_t}$ and obtain the transition matrix $\text{Exp}[\mathbf{A_t}] = \mathbf{F_t}$. To achieve this, we use the Taylor expansion of Exp function with 5 degrees.

$$\dot{\mathbf{s}}_t = \mathbf{A_t}\mathbf{s_t} = \begin{bmatrix} -10 & 10 & 0 \\ 28 - z & -1 & 0 \\ y & 0 & -\frac{8}{3} \end{bmatrix} \begin{bmatrix} x \\ y \\ z \end{bmatrix}, \text{ and } \quad \mathbf{F_t} = \text{Exp}[\mathbf{A_t}] = \mathbf{I} + \sum_{j=1}^{J} \frac{(\mathbf{A_t}.\Delta t)^j}{j!} \tag{37}$$

where $J$ is the degrees of expansion and $\mathbf{I}$ is the identity matrix. For the emission matrix we use $\mathbf{H_t} = \mathbf{I}$ and for process and observation noise standard deviation, we use $\mathbf{Q_t} = \frac{1}{100}\sigma^2\mathbf{I}$ and $\mathbf{R_t} = \sigma^2\mathbf{I}$, respectively.

### A.7   MOVEMENT MODEL DETAILS FOR THE NCLT EXPERIMENT

We assume that the segway robot is moved with a constant velocity, that the equations for such dynamics are given by

$$\frac{\partial p_1}{\partial t} = v_1, \quad \frac{\partial p_2}{\partial t} = v_2, \quad \frac{\partial v_1}{\partial t} = 0, \quad \frac{\partial v_2}{\partial t} = 0, \quad \mathbf{x}_t = [p_1, v_1, p_2, v_2], \quad \mathbf{y}_t = [p_1, p_2]. \tag{38}$$

By such assumptions for the motion's equations the transition, process noise distribution, emission and measurement noise distribution matrices can be obtained by

$$\mathbf{F} = \begin{bmatrix} 1 & \Delta t & 0 & 0 \\ 0 & 1 & 0 & 0 \\ 0 & 0 & 1 & \Delta t \\ 0 & 0 & 0 & 1 \end{bmatrix}, \quad \mathbf{Q} = \sigma^2 \begin{bmatrix} \Delta t & 0 & 0 & 0 \\ 0 & \Delta t & 0 & 0 \\ 0 & 0 & \Delta t & 0 \\ 0 & 0 & 0 & \Delta t \end{bmatrix}, \quad \mathbf{H} = \begin{bmatrix} 1 & 0 & 0 & 0 \\ 0 & 0 & 1 & 0 \end{bmatrix},$$
$$\mathbf{R} = \lambda^2 \begin{bmatrix} 1 & 0 \\ 0 & 1 \end{bmatrix}. \tag{39}$$

where $\Delta t = 1$ since the sampling frequency is 1Hz. Process and measurement variance parameters, $\sigma$ and $\lambda$, are unknown that the model will learn them. we split the whole sequence into training, testing and validation folds with the length of 3600 ( 18 sequences of length $T = 200$) , 280 (1 sequence of length $T = 280$) and 400 (2 sequences of length $T = 200$), respectively.

## A.8   NETWORK STRUCTURE AND PARAMETERS

In all experiments, Adam optimizer Kingma & Ba (2014) has been used on NVIDIA GeForce GTX 1050 Ti. We conduct a grid search for finding the hyperparameters to rule out the possibility of the models being trained with the suboptimal hyperparameters. To find the initial learning rate, by conducting a grid search between 0.001 and 0.2 with the increment of 0.005, we select the best one among them that corresponds to the highest log-likelihood. With an initial learning rate of 0.006 and an exponential decay with rate of 0.9 every 10 epochs, we employ back propagation through time Werbos (1990) to compute the gradients as we deploy GRU cells in the structure. Layer normalization technique Ba et al. (2016) is used to stabilize the dynamics in the recurrent structure and normalize the filter response. Elu + 1 activation function, can ensure the positiveness of the diagonal elements of the process, noise and covariance matrices.

In order to prevent the model being stuck in the poor local minima, e.g. focusing on the reconstruction instead of learning the dynamics obtained by filtering-smoothing, we find it useful to use two training tricks for an end-to-end learning:

1- Generating time correlated noisy sequences as consecutive observations, forces the model to learn the dynamics instead of focusing on reconstruction, e.g. figure 13 and 15.

2- For the first few epochs, only learn auto-encoder(MLPs) and globally learned parameters, e.g. $\mathbf{F}^{(k)}$ and $\mathbf{H}^{(k)}$, but not *Dynamics Network* parameters $\alpha_t(\mathbf{x}_{t-1})$. All the parameters are jointly learned, afterwards. This allows the system to learn good embedding and meaningful latent vectors at first, then learns how to employ $K$ different dynamics variables.

In the lack of dynamics, for the low dimensional observations we use $K = 5$, while for the high dimensional observations we use $K = 15$ as they need to learn more complex dynamics. In general, if the GIN is flexible enough, tuning the parameters is not difficult as the GIN is capable to learn how to prune unused elements by the *Dynamics Network*.

To prevent the model being stuck into mode collapse, we provided two solutions:

1- By introducing $k$ sets of $\mathbf{F}^k, \mathbf{H}^k$, where each set of $\mathbf{F}^k, \mathbf{H}^k$ models different dynamics, we introduce a loss term with a small constant factor which tries to increase the distance of each pair of $\mathbf{F}^k, \mathbf{H}^k$ set. Intuitively, the presence of different dynamics can easily modify the states in each update. We found this method as a potential solution to prevent the model go through the mode collapse.

2- Considering the negative distance of consecutive pairs of states as additional loss term with a small constant factor (the distance can be considered as euclidean difference of mean or KL of two consecutive states). Intuitively, this solution is forcing the states to not have overlap with each other and impose them to change in each update step.

In the simulation results, we have used the first option.

### A.8.1   PROPOSED ARCHITECTURE FOR HIGH AND LOW DIMENSIONAL OBSERVATIONS

The proposed structure to deal with high dimensional observations in the lack of the dynamics(the first three experiments in the paper) is shown in figure 10. While, the proposed structure to handle low dimensional observations in the presence of the dynamics(the last two experiments in the paper) is shown in the figure 11.

### A.8.2   EMPIRICAL RUNNING TIMES AND PARAMETERS

We present the number of parameters of the utilized cell structures in our experiments and their corresponding empirical running times for 1 epoch in the table 7 and 8. In the first row of each model

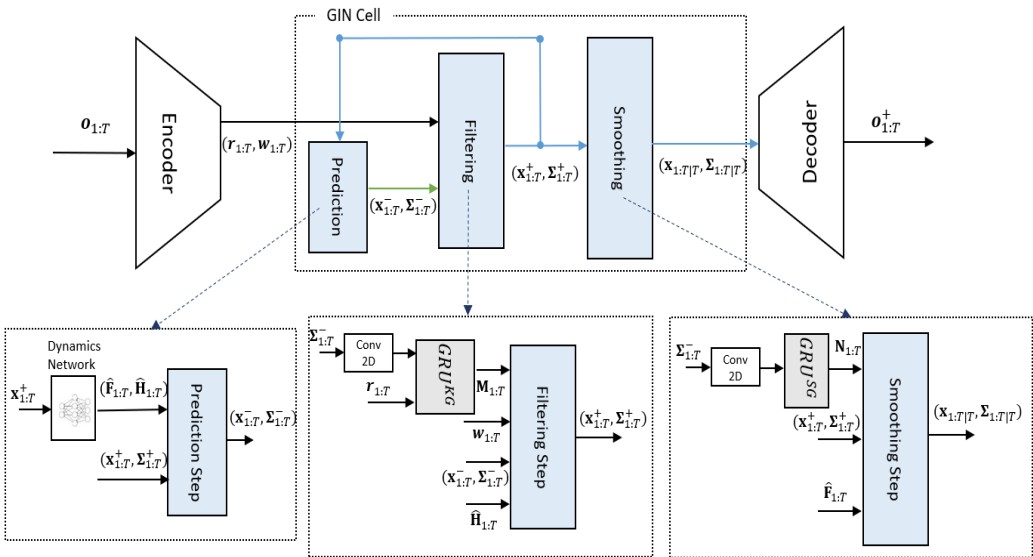

Figure 10: Proposed architecture for operating high dimensional observations in the lack of dynamics.

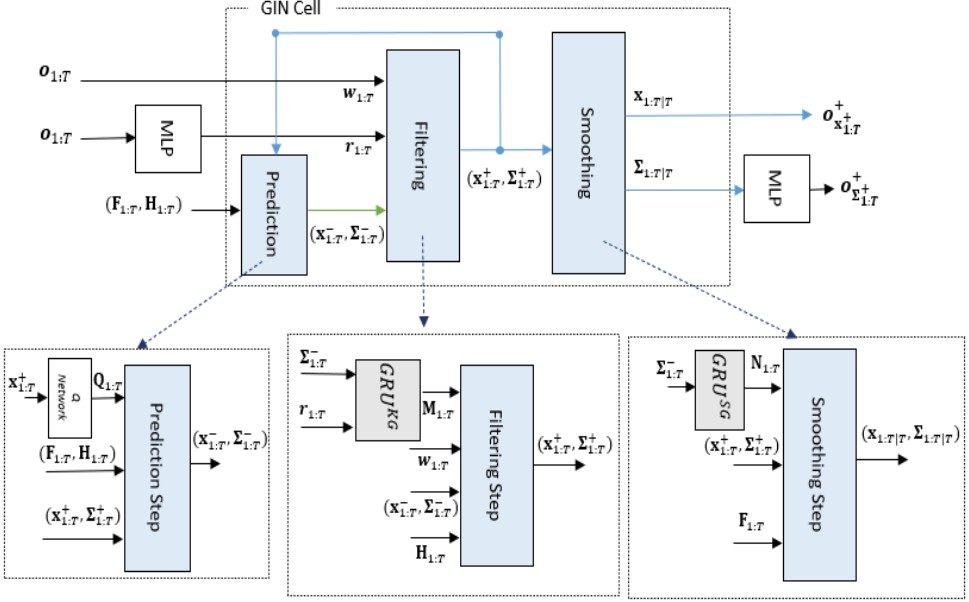

Figure 11: Proposed architecture for operating low dimensional observations in the presence of dynamics.

structure in the high dimensional experiments, we set the number of their parameters approximately equal to our GIN to indicate the outperformance of the GIN with the same number of the parameters, i.e. tables 1, 2 and 5, while we include the empirical running time and parameters for more complex structures as well. Extra running time of EM-variational approaches, like KVAE, is due to employing classic Bayesian equations because it increases the running time substantially when dealing with higher dimensional observations, however, the GIN circumvent this difficulty. The number of the parameters of the GIN are noticeably lower than other memory cells, e.g. LSTM and GRU, and EM-variational methods as we convert high dimensional sparse covariance matrices into lower dimensional covariance matrices by employing convolutional operator. It allows us to reduce the parameters of internal GRU cells quadratically.

### A.8.3 QUALITATIVE COMPARISON OF THE GIN TO RECENT RELATED WORK.

In table 9, we make a comparison to show whether algorithms are able to handle high and low dimensional observations, learn dynamics, use available-partial dynamics, estimate state appropriately, provide model's uncertainty estimates handling noisy data, handle missing data and perform direct optimization. Classic LGSSMs, e.g. EKF and UKF, work based on the linearization of the transition and emission equations and apply classic Bayesian updates over the linearized system with respect to the states. In other words, $(\mathbf{F}, \mathbf{H})$ in the classic LGSSMs are not data-deriven nor trainable. Despite classic LGSSMs, in the GIN we use a data-driven based network to learn dynamics, i.e. *Dynamics Network* in the paper.

### A.8.4 SINGLE-PENDULUM AND DOUBLE-PENDULUM EXPERIMENTS

DATA GENERATION.

Dataset consists of 1000 train, 100 valid and 100 test sequences with the length of 150. The sequences are distorted via generated noise, while in the informed imputation task half of the images are removed and boolean flags indicating the availability of the observations are passed to the cell instead. If the imputation task is in uninformed type, black images are considered as the observations instead of informing the cell with boolean flags.

ENCODER/DECODER AND THE DYNAMICS NETWORK ARCHITECTURE

To design the dynamics network, we use a MLP including 60 hidden units with Relu activation function and a softmax activation for the last layer. The state mean, with size of $n$, and number of the bases, with size of $k$, are the input and output of the dynamics, respectively. The structures of the encoder and decoder are in the table 10. In the table 10, $m$ is the transferred observation dimension that various values for this parameter are taken into account in the results. In the state estimation tasks, *out dim* is 4 and 8 for the single-pendulum and double-pendulum experiment, respectively. For the imputation task, number of the hidden units of the KG and SG network is set to 40 and 30, respectively. The convolutional layer applied over the covariance matrix has 8 filters with kernel size of 5.

### A.8.5 LORENZ ATTRACTOR AND NCLT EXPERIMENTS

In these two experiments that we have the knowledge of the dynamics, we employ a fully connected with the observations as its input and output dimension of 3 and 2 for Lorenz attractor and NCLT experiments, respectively, to obtain the observation noise, $\mathbf{r}$. The activation function is Elu + 1. Similarly another fully connected with the posterior state as its input and output dimension of 3 and 2 for Lorenz attractor and NCLT experiments, respectively, to attain the uncertainty estimates, $\mathbf{o}_\sigma^+$. To estimate the process noise matrix, a fully connect with the posterior state as the input and Elu + 1 activation function is used. Similarly, a GRU cell that maps the posterior states to the process noise matrix with 10 hidden units can be used.

Table 7: Empirical running times and parameters of high-low dimensional experiments.

| Cell | Single Pend | | Double Pend | | KITTI | |
|---|---|---|---|---|---|---|
| | Param | T/E | Param | T/E | Param | T/E |
| LSTM (units=25) | ∼18k | ∼56s | ∼18k | ∼56s | ∼45k | ∼83s |
| LSTM (units=50) | ∼36k | ∼70s | ∼36k | ∼71s | ∼70k | ∼95s |
| LSTM (units=100) | ∼76k | ∼98s | ∼76k | ∼96s | ∼120k | ∼131s |
| GRU (units=30) | ∼18k | ∼61s | ∼18k | ∼62s | ∼42k | ∼79s |
| GRU (units=50) | ∼27k | ∼65s | ∼27k | ∼67s | ∼53k | ∼84s |
| GRU (units=100) | ∼57k | ∼86s | ∼57k | ∼85s | ∼90k | ∼111s |
| KVAE (n=40) | ∼25k | ∼95s | ∼25k | ∼97s | ∼62k | ∼141s |
| KVAE (n=60) | ∼36k | ∼114s | ∼36k | ∼111s | ∼80k | ∼165s |
| RKN (n=100) | ∼25k | ∼57s | ∼25k | ∼58s | ∼45k | ∼79s |
| LGSSM (n=30) | ∼12k | ∼82s | ∼12k | ∼84s | ∼30k | ∼117s |
| LGSSM (n=45) | ∼15k | ∼98s | ∼15k | ∼97s | ∼36k | ∼136s |
| GIN (n=30) | ∼18k | ∼55s | ∼18k | ∼55s | ∼42k | ∼80s |
| GIN (n=45) | ∼25k | ∼59s | ∼25k | ∼58s | ∼48k | ∼83s |

Table 8: Low dimensional experiments.

| Cell | Lorenz | | NCLT | |
|---|---|---|---|---|
| | Param | T/E | Param | T/E |
| KalmanNet | | | ∼30k | ∼65s |
| GNN | ∼10k | ∼35s | ∼10k | ∼40s |
| RNN | | | ∼40k | ∼79s |
| DSSM | ∼40k | ∼76s | ∼40k | ∼81s |
| LGSSM | ∼6k | ∼20s | ∼6k | ∼22s |
| GIN | ∼9k | ∼28s | ∼9k | ∼31s |

Table 9: Learning the dynamics in LGSSM is shown with $\times / \checkmark$ because general LGSSMs, e.g. UKF and EKF, are not able to learn the dynamics. However, in our setting and parameterization we use a data driven-based network for obtaining $(\mathbf{F}, \mathbf{H})$ to make LGSSMs comparable with the GIN for high dimensional observation experiments.

| Model | high-d | low-d | learn dynamics | use dynamics | state est | uncertainty | missing-noise | dir opt |
|---|---|---|---|---|---|---|---|---|
| LSTM (Hochreiter & Schmidhuber, 1997) | $\checkmark$ | $\checkmark$ | $\checkmark$ | $\checkmark$ | $\checkmark$ | $\times$ | $\checkmark$ | $\checkmark$ |
| GRU (Cho et al., 2014) | $\checkmark$ | $\checkmark$ | $\checkmark$ | $\checkmark$ | $\checkmark$ | $\times$ | $\checkmark$ | $\checkmark$ |
| P2T (Wahlström et al., 2015) | $\checkmark$ | $\checkmark$ | $\checkmark$ | $\times$ | $\checkmark$ | $\times$ | $\times$ | $\checkmark$ |
| E2C (Watter et al., 2015) | $\checkmark$ | $\checkmark$ | $\checkmark$ | $\times$ | $\times$ | $\checkmark$ | $\times$ | $\times$ |
| BB-VI (Archer et al., 2015) | $\times$ | $\checkmark$ | $\checkmark$ | $\times$ | $\times$ | $\checkmark$ | $\checkmark$ | $\times$ |
| SIN (Krishnan et al., 2017) | $\checkmark$ | $\checkmark$ | $\checkmark$ | $\times$ | $\times$ | $\checkmark$ | $\checkmark$ | $\times$ |
| DVBF (Karl et al., 2016) | $\checkmark$ | $\checkmark$ | $\checkmark$ | $\times$ | $\times$ | $\checkmark$ | $\checkmark$ | $\times$ |
| VSMC (Naesseth et al., 2018) | $\checkmark$ | $\checkmark$ | $\checkmark$ | $\times$ | $\times$ | $\checkmark$ | $\checkmark$ | $\times$ |
| DSA (Li & Mandt, 2018) | $\checkmark$ | $\checkmark$ | $\checkmark$ | $\times$ | $\times$ | $\checkmark$ | $\times$ | $\times$ |
| KVAE (Fraccaro et al., 2017) | $\times$ | $\checkmark$ | $\checkmark$ | $\times$ | $\times$ | $\checkmark$ | $\checkmark$ | $\times$ |
| EKVAE (Klushyn et al., 2021) | $\times$ | $\checkmark$ | $\checkmark$ | $\times$ | $\times$ | $\checkmark$ | $\checkmark$ | $\times$ |
| rSLSD Linderman et al. (2017) | $\times$ | $\checkmark$ | $\times$ | $\checkmark$ | $\checkmark$ | $\checkmark$ | $\times$ | $\times$ |
| DeepAR Salinas et al. (2020) | $\times$ | $\checkmark$ | $\checkmark$ | $\times$ | $\checkmark$ | $\checkmark$ | $\times$ | $\checkmark$ |
| DSSM Rangapuram et al. (2018) | $\times$ | $\checkmark$ | $\checkmark$ | $\times$ | $\checkmark$ | $\checkmark$ | $\times$ | $\checkmark$ |
| HybridGNN Garcia Satorras et al. (2019) | $\times$ | $\checkmark$ | $\times$ | $\checkmark$ | $\checkmark$ | $\times$ | $\checkmark$ | $\checkmark$ |
| KalmanNet Revach et al. (2021) | $\times$ | $\checkmark$ | $\times$ | $\checkmark$ | $\checkmark$ | $\times$ | $\checkmark$ | $\checkmark$ |
| SSI Ruhe & Forré (2021) | $\times$ | $\checkmark$ | $\times$ | $\checkmark$ | $\checkmark$ | $\checkmark$ | $\checkmark$ | $\checkmark$ |
| LGSSM | $\times$ | $\checkmark$ | $\times / \checkmark$ | $\checkmark$ | $\checkmark$ | $\checkmark$ | $\checkmark$ | $\checkmark$ |
| GIN | $\checkmark$ | $\checkmark$ | $\checkmark$ | $\checkmark$ | $\checkmark$ | $\checkmark$ | $\checkmark$ | $\checkmark$ |

## A.9 VISUALIZATION AND THE IMPUTATION

Graphical results of informed, uninformed and noisy observations for image imputation task for both single and double pendulum experiments can be found in 12, 13, 14 and 15 figures. Inference for the trained smoothing and filtering distributions of all high dimensional experiments are in 16, 17, 18, 19, 20, 21, 22, 23, 24, 25, 26, 27, 28, 29, 30, 31, 32, 33, 34, 35, 36, 37, 38 and 39 figures, where we generated samples from the smoothing distribution, $f(\mathbf{x}_t | \mathbf{w}_{1:T})$, and the filtering distribution, $f(\mathbf{x}_t | \mathbf{w}_{1:t})$. Then we fit density on the generated samples. This visualization shows the effectiveness of the GIN in reducing the uncertainty of the estimates compare to LGSSM and KVAE. Finally, the results of NCLT experiment are in figure 47.

Table 10: The structure of the encoder and decoder for single and double pendulum experiments.

| Encoder | Decoder |
|---|---|
| $6 \times 6$ Conv, 12, max pooling $2 \times 2$, stride $2 \times 2$ | $\mathbf{o}_x^+$: fully connected: *out dim*, linear activation |
| LayerNormalizer() | $\mathbf{o}_\Sigma^+$: fully connected, Elu + 1 activation |
| $4 \times 4$ Conv, 12, max pooling $2 \times 2$, stride $2 \times 2$ | if *imputation task*: |
| LayerNormalizer() |     fully connected: 144 |
| fully connected: 40 |     $6 \times 6$ Trns Conv, 16, stride $4 \times 4$ |
| $\mathbf{w}$: fully connected: $m$, linear activation |     LayerNormalizer() |
| $\mathbf{r}$: fully connected, Elu + 1 activation |     $4 \times 4$ Trns Conv, 12, stride $2 \times 2$ |
| |     LayerNormalizer() |
| |     $\mathbf{o}_i^+$: $1 \times 1$ Trns Conv, stride $1 \times 1$, softmax |

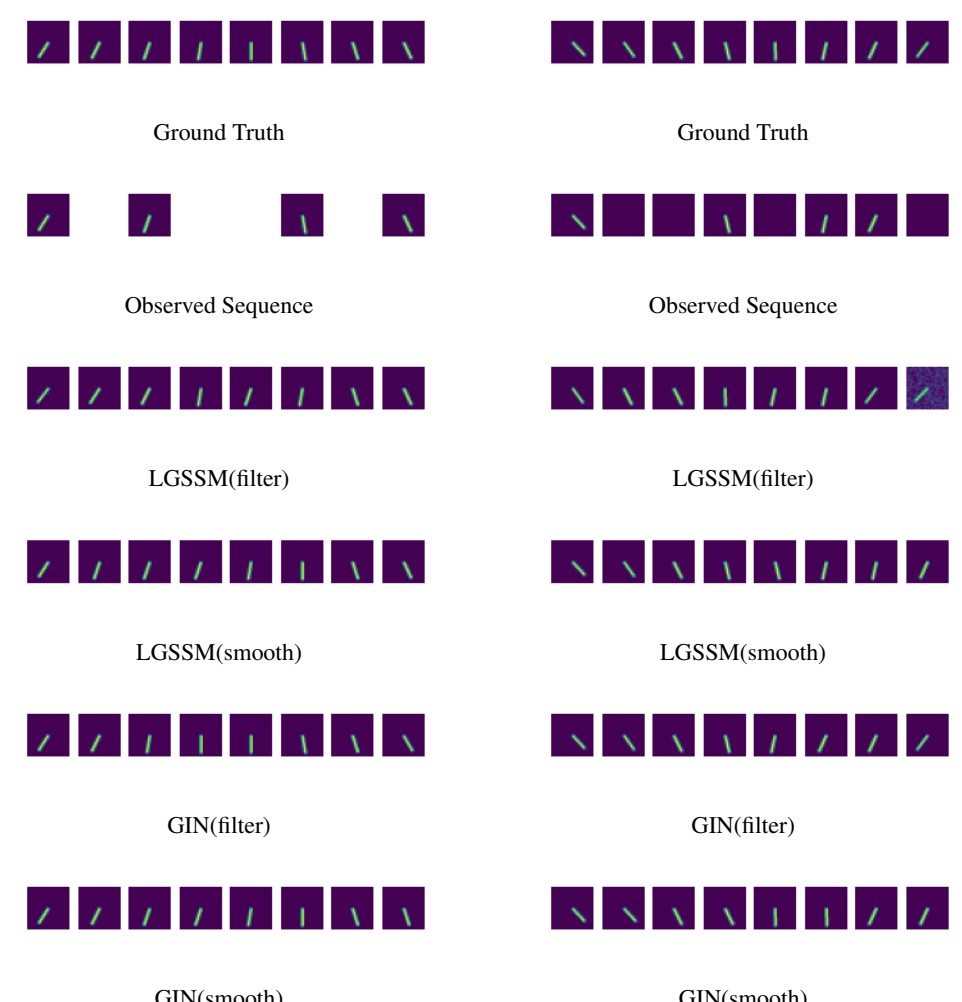

Figure 12: Informed(left column) and uninformed(right column) image imputation task for the single pendulum experiments.

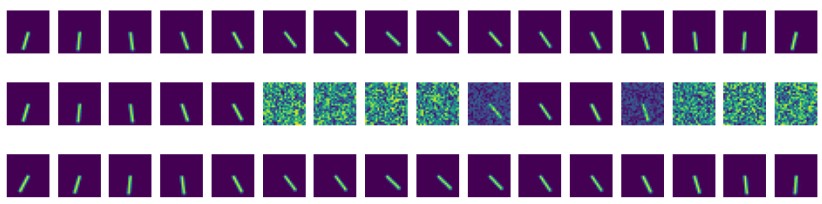

Figure 13: Image imputation task for the single pendulum experiment exposed to the noisy observations, where the generated noise has correlation with the time. Each figure, beginning from top to bottom, indicates the ground truth, noisy observation and the imputation results of the GIN.

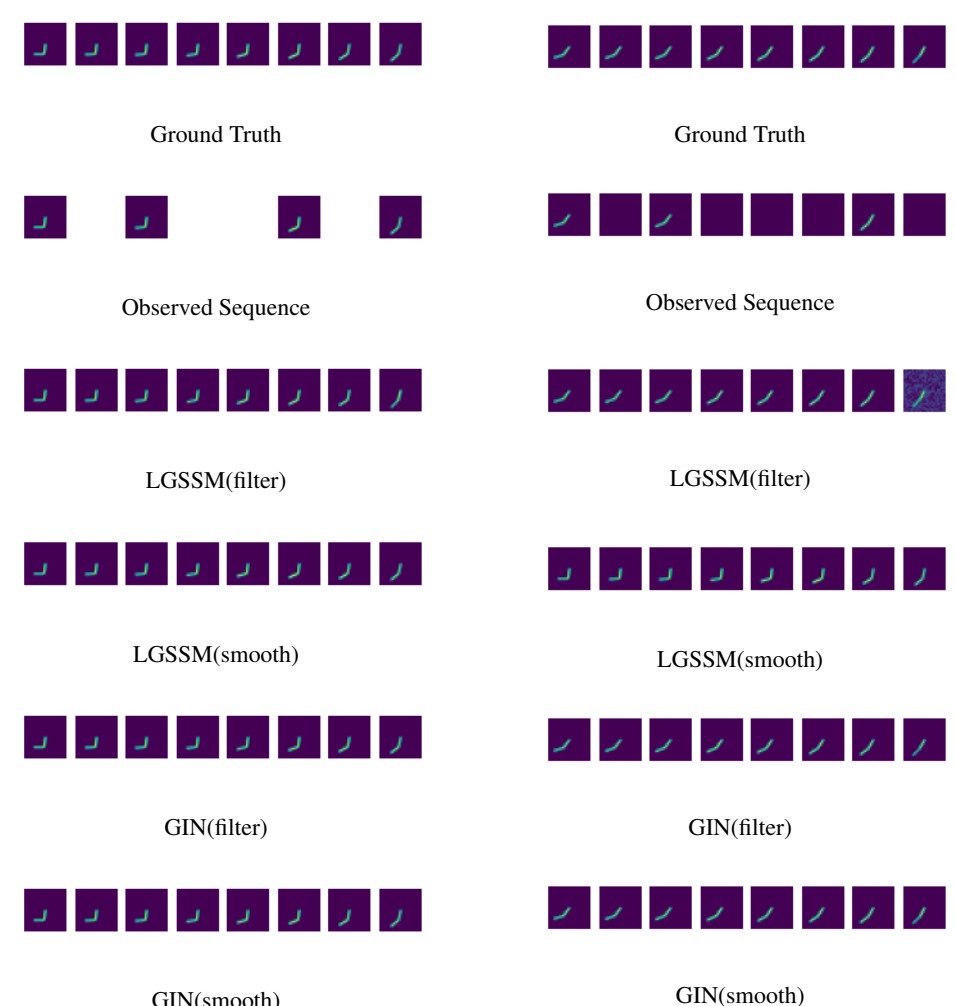

Figure 14: Informed(left column) and uninformed(right column) image imputation task for the double pendulum experiments.

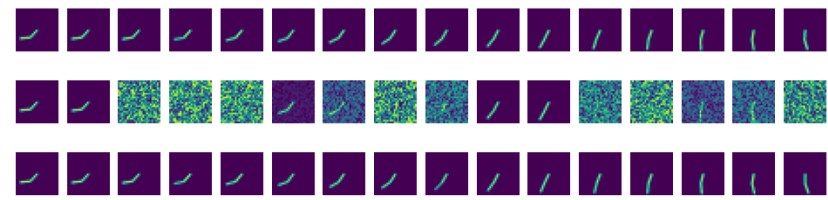

Figure 15: Image imputation task for the double pendulum experiment exposed to the noisy observations, where the generated noise has correlation with the time. Each figure, beginning from top to bottom, indicates the ground truth, noisy observation and the imputation results of the GIN.

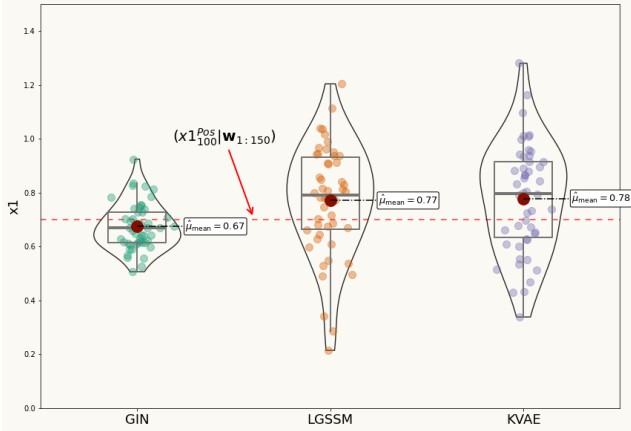

Figure 16: Inference for the single pendulum $x1$ position at 100-th time step. Generated samples from smoothened distribution, $f(x1_{100}|\mathbf{w}_{1:150})$, trained by the GIN, LGSSM and KVAE, respectively. The dashed red line $(x1_{100}^{\text{Pos}}|\mathbf{w}_{1:150})$ is the ground truth state with distribution of $\delta(x1_{100} - 0.7)$. We calculate the sample mean and fit a distribution on the samples for further visualization and comparison purpose.

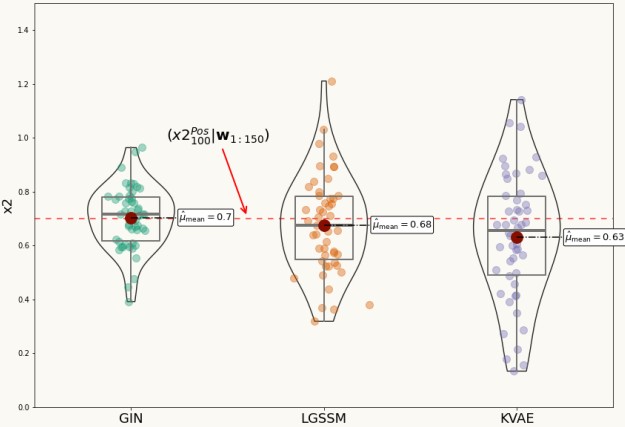

Figure 17: Inference for the single pendulum $x2$ position at 100-th time step. Generated samples from smoothened distribution, $f(x2_{100}|\mathbf{w}_{1:150})$, trained by the GIN, LGSSM and KVAE, respectively.

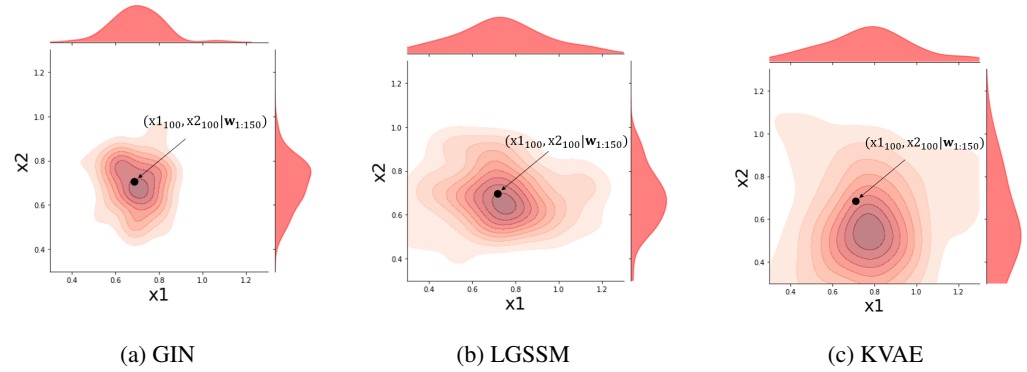

(a) GIN        (b) LGSSM        (c) KVAE

Figure 18: Generated samples from the trained smoothened joint distribution of the single pendulum position, $(x1, x2)$, at 100-th time step for the GIN, LGSSM and KVAE, respectively. The ground truth is shown with a black point.

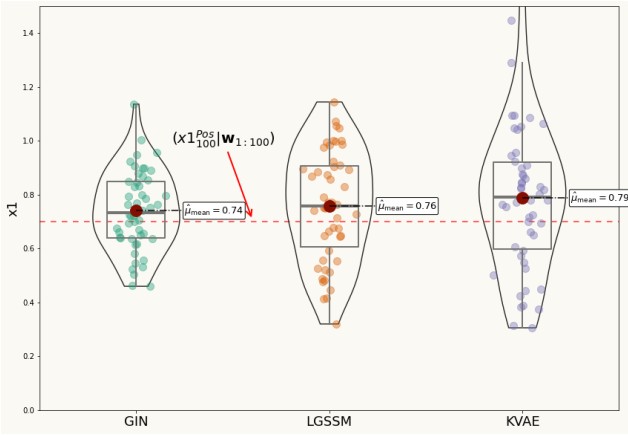

Figure 19: Inference for the single pendulum $x1$ position at 100-th time step. Generated samples from filter distribution, $f(x1_{100}|\mathbf{w}_{1:100})$, trained by the GIN, LGSSM and KVAE, respectively. The dashed red line $(x1_{100}^{\text{Pos}}|\mathbf{w}_{1:100})$ is the ground truth state with distribution of $\delta(x1_{100} - 0.7)$.

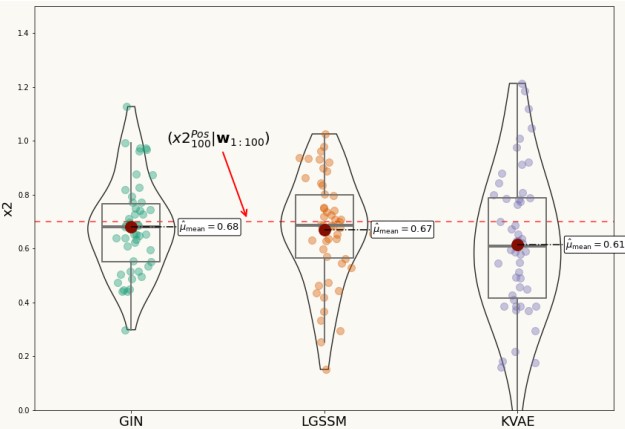

Figure 20: Inference for the single pendulum $x2$ position at 100-th time step. Generated samples from filter distribution, $f(x2_{100}|\mathbf{w}_{1:100})$, trained by the GIN, LGSSM and KVAE, respectively.

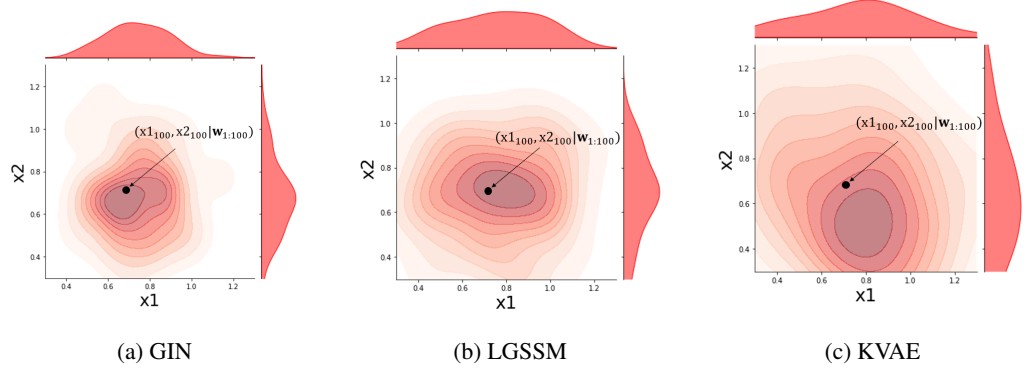

| (a) GIN | (b) LGSSM | (c) KVAE |

Figure 21: Generated samples from the trained filter joint distribution of the single pendulum position, $(x1, x2)$, at 100-th time step for the GIN, LGSSM and KVAE, respectively. The ground truth is shown with a black point.

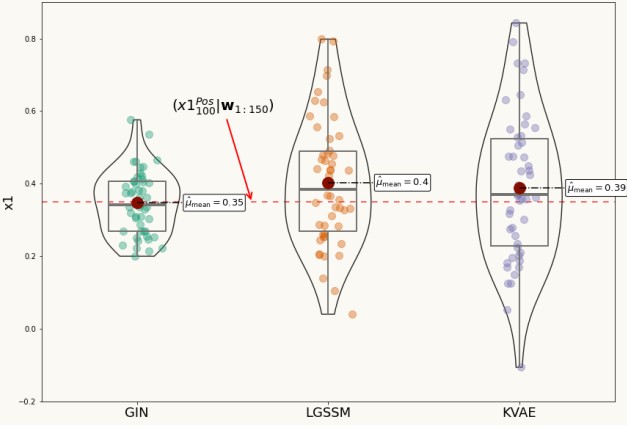

Figure 22: Inference for the double pendulum $x1$ position at 100-th time step. Generated samples from smoothened distribution, $f(x1_{100}|\mathbf{w}_{1:150})$, trained by the GIN, LGSSM and KVAE, respectively. The dashed red line $(x1_{100}^{\text{Pos}}|\mathbf{w}_{1:150})$ is the ground truth state with distribution of $\delta(x1_{100} - 0.35)$.

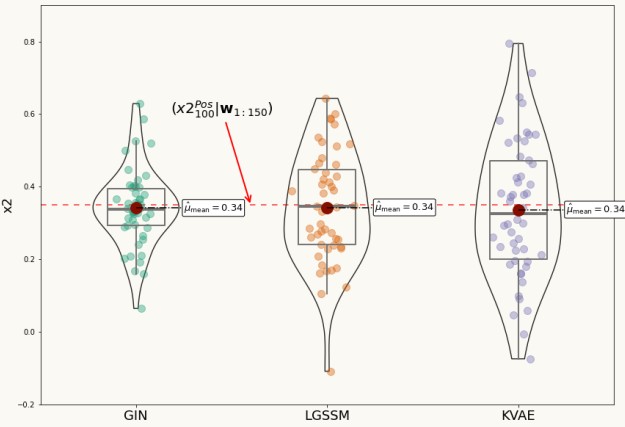

Figure 23: Inference for the double pendulum $x2$ position at 100-th time step. Generated samples from smoothened distribution, $f(x2_{100}|\mathbf{w}_{1:150})$, trained by the GIN, LGSSM and KVAE, respectively. The dashed red line $(x2_{100}^{\text{Pos}}|\mathbf{w}_{1:150})$ is the ground truth state with distribution of $\delta(x2_{100} - 0.35)$.

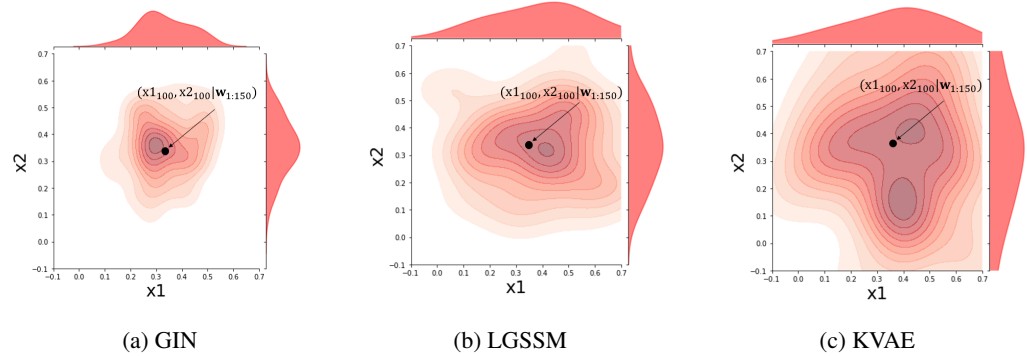

(a) GIN        (b) LGSSM        (c) KVAE

Figure 24: Generated samples from the trained smoothened joint distribution of the double pendulum first joint position, $(x1, x2)$, at 100-th time step for the GIN, LGSSM and KVAE, respectively. The ground truth is shown with a black point.

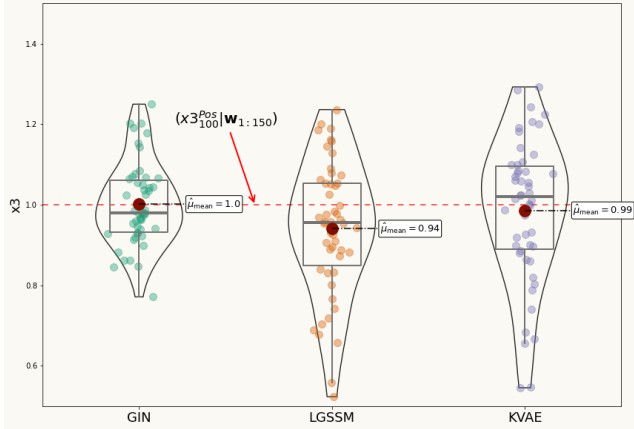

Figure 25: Inference for the double pendulum $x3$ position at $100$-th time step. Generated samples from smoothened distribution, $f(x3_{100}|\mathbf{w}_{1:150})$, trained by the GIN, LGSSM and KVAE, respectively. The dashed red line $(x3_{100}^{\text{Pos}}|\mathbf{w}_{1:150})$ is the ground truth state with distribution of $\delta(x3_{100}-1)$.

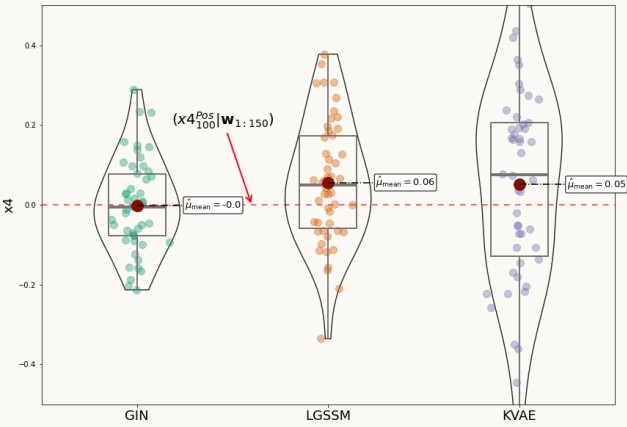

Figure 26: Inference for the double pendulum $x4$ position at $100$-th time step. Generated samples from smoothened distribution, $f(x4_{100}|\mathbf{w}_{1:150})$, trained by the GIN, LGSSM and KVAE, respectively.

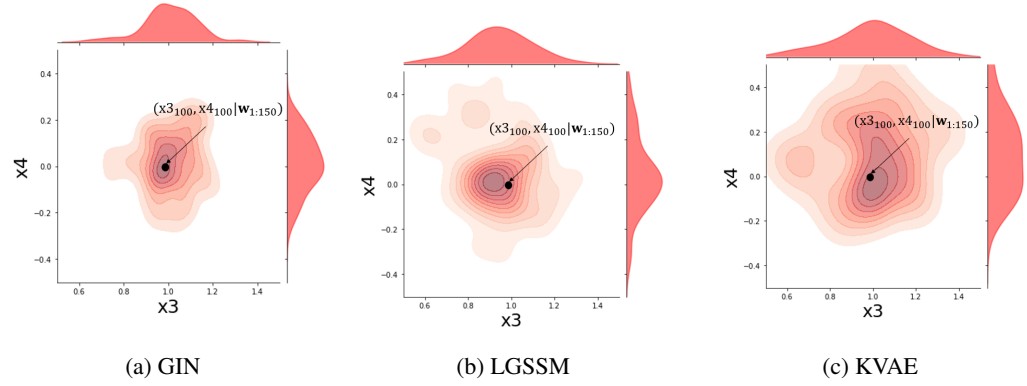

(a) GIN        (b) LGSSM        (c) KVAE

Figure 27: Generated samples from the trained smoothened joint distribution of the double pendulum second joint position, $(x3, x4)$, at $100$-th time step for the GIN, LGSSM and KVAE, respectively. The ground truth is shown with a black point.

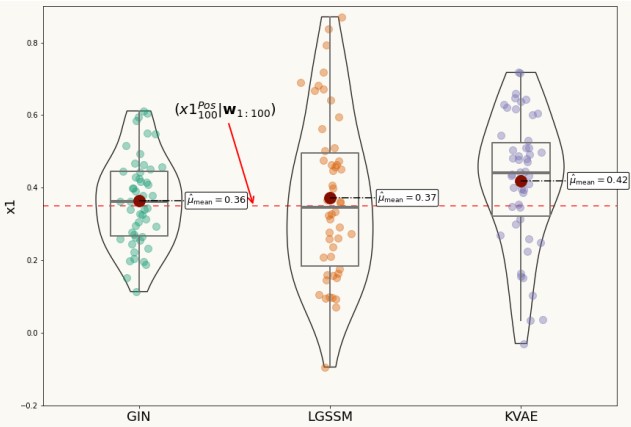

Figure 28: Inference for the double pendulum $x1$ position at 100-th time step. Generated samples from filter distribution, $f(x1_{100}|\mathbf{w}_{1:100})$, trained by the GIN, LGSSM and KVAE, respectively. The dashed red line $(x1_{100}^{\text{Pos}}|\mathbf{w}_{1:100})$ is the ground truth state with distribution of $\delta(x1_{100} - 0.35)$.

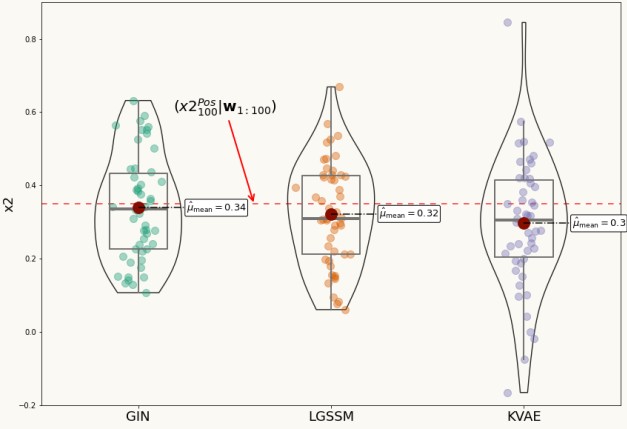

Figure 29: Inference for the double pendulum $x2$ position at 100-th time step. Generated samples from filter distribution, $f(x2_{100}|\mathbf{w}_{1:100})$, trained by the GIN, LGSSM and KVAE, respectively. The dashed red line $(x2_{100}^{\text{Pos}}|\mathbf{w}_{1:100})$ is the ground truth state with distribution of $\delta(x2_{100} - 0.35)$.

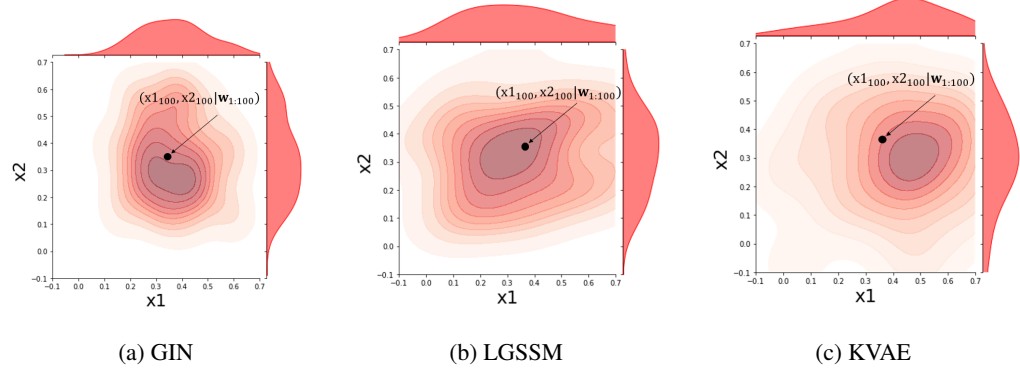

(a) GIN        (b) LGSSM        (c) KVAE

Figure 30: Generated samples from the trained filter joint distribution of the double pendulum first joint position, $(x1, x2)$, at 100-th time step for the GIN, LGSSM and KVAE, respectively. The ground truth is shown with a black point.

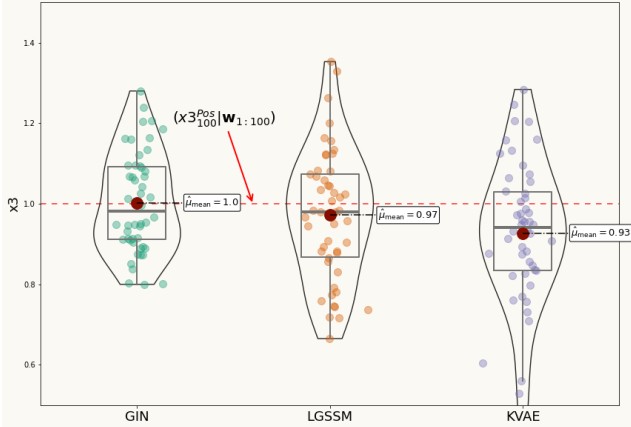

Figure 31: Inference for the double pendulum $x3$ position at 100-th time step. Generated samples from filter distribution, $f(x3_{100}|\mathbf{w}_{1:100})$, trained by the GIN, LGSSM and KVAE, respectively. The dashed red line $(x3_{100}^{\text{Pos}}|\mathbf{w}_{1:100})$ is the ground truth state with distribution of $\delta(x3_{100} - 1)$.

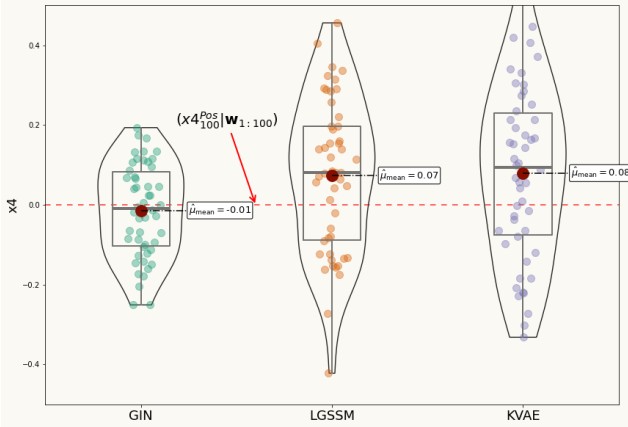

Figure 32: Inference for the double pendulum $x4$ position at 100-th time step. Generated samples from filter distribution, $f(x4_{100}|\mathbf{w}_{1:100})$, trained by the GIN, LGSSM and KVAE, respectively.

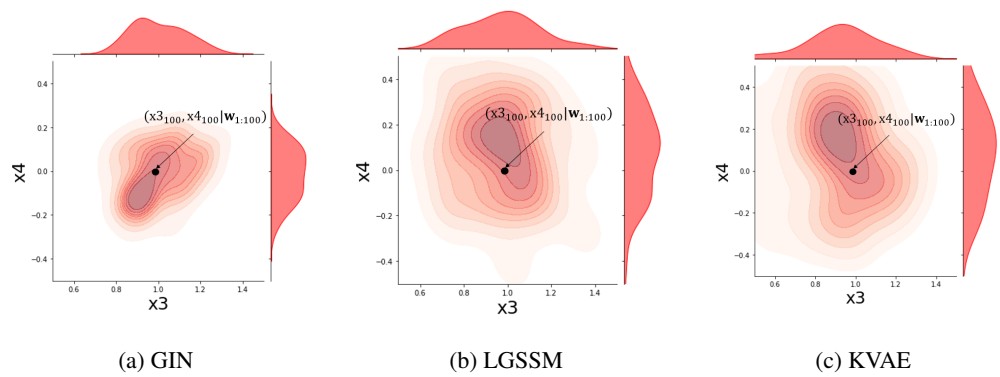

(a) GIN  (b) LGSSM  (c) KVAE

Figure 33: Generated samples from the trained filter joint distribution of the double pendulum second joint position, $(x3, x4)$, at 100-th time step for the GIN, LGSSM and KVAE, respectively. The ground truth is shown with a black point.

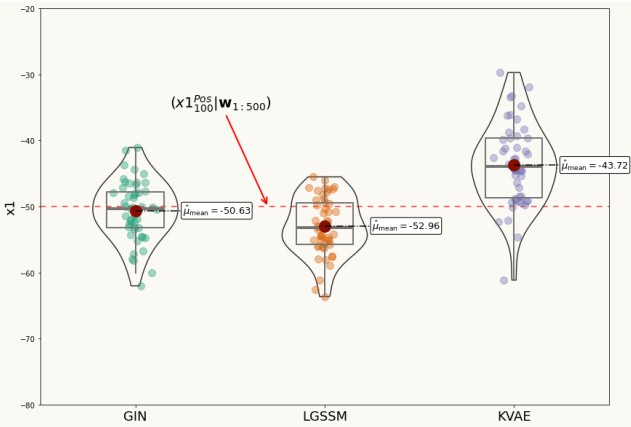

Figure 34: Inference for the visual odometry $x1$ position at 100-th time step. Generated samples from smoothened distribution, $f(x1_{100}|\mathbf{w}_{1:500})$, trained by the GIN, LGSSM and KVAE, respectively. The dashed red line $(x1_{100}^{\text{Pos}}|\mathbf{w}_{1:500})$ is the ground truth state with distribution of $\delta(x1_{100}+50)$.

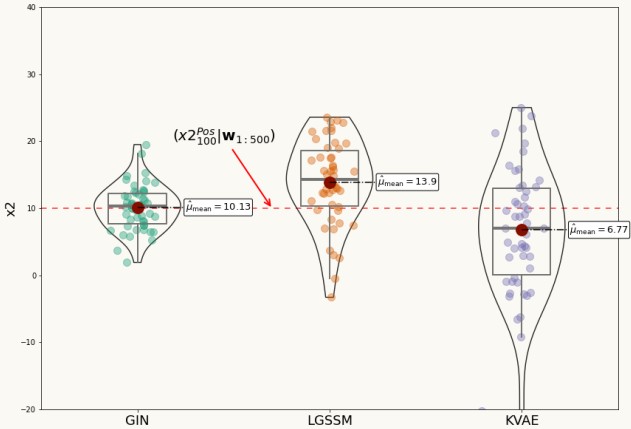

Figure 35: Inference for the visual odometry $x2$ position at 100-th time step. Generated samples from smoothened distribution, $f(x2_{100}|\mathbf{w}_{1:500})$, trained by the GIN, LGSSM and KVAE, respectively. The dashed red line $(x2_{100}^{\text{Pos}}|\mathbf{w}_{1:500})$ is the ground truth state with distribution of $\delta(x1_{100}-10)$.

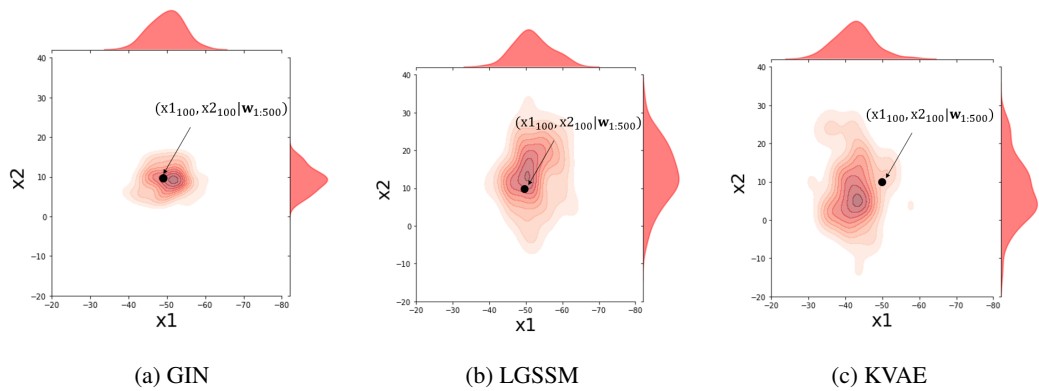

   (a) GIN        (b) LGSSM        (c) KVAE

Figure 36: Generated samples from the trained smoothened joint distribution of the visual odometry joint position, $(x1, x2)$, at 100-th time step for the GIN, LGSSM and KVAE, respectively. The ground truth is shown with a black point.

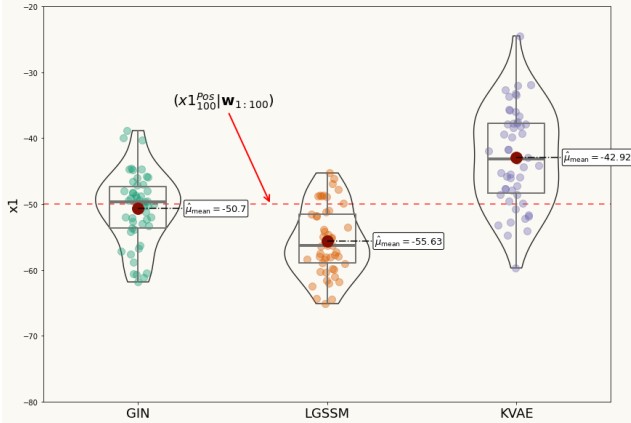

Figure 37: Inference for the visual odometry $x1$ position at 100-th time step. Generated samples from filter distribution, $f(x1_{100}|\mathbf{w}_{1:100})$, trained by the GIN, LGSSM and KVAE, respectively. The dashed red line $(x1_{100}^{\text{Pos}}|\mathbf{w}_{1:100})$ is the ground truth state with distribution of $\delta(x1_{100} + 50)$.

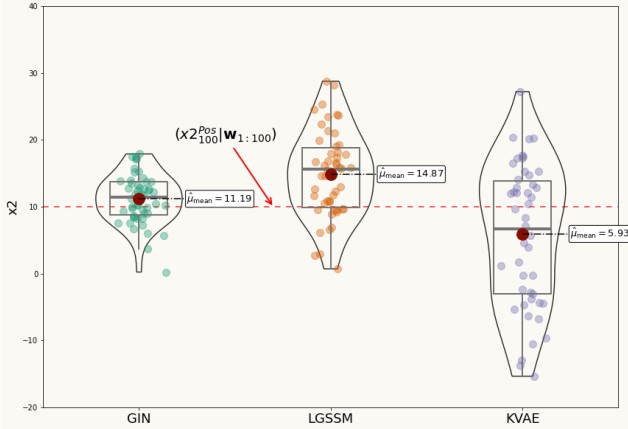

Figure 38: Inference for the visual odometry $x2$ position at 100-th time step. Generated samples from filter distribution, $f(x2_{100}|\mathbf{w}_{1:100})$, trained by the GIN, LGSSM and KVAE, respectively. The dashed red line $(x2_{100}^{\text{Pos}}|\mathbf{w}_{1:100})$ is the ground truth state with distribution of $\delta(x1_{100} - 10)$.

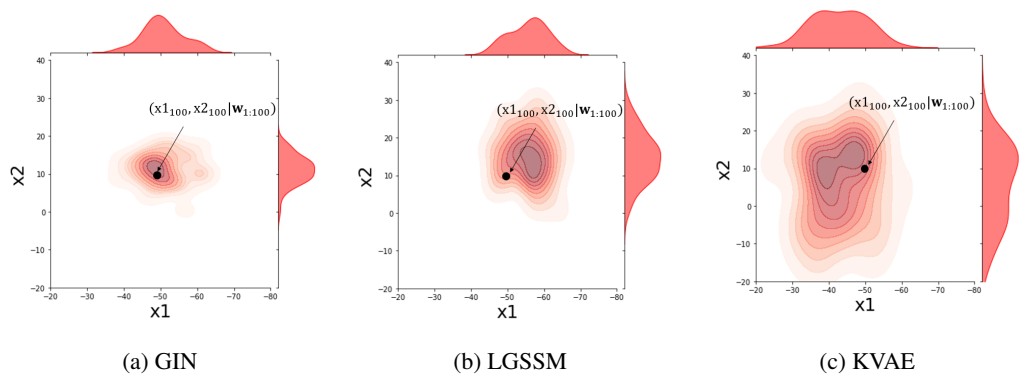

(a) GIN        (b) LGSSM        (c) KVAE

Figure 39: Generated samples from the trained filter joint distribution of the visual odometry joint position, $(x1, x2)$, at 100-th time step for the GIN, LGSSM and KVAE, respectively. The ground truth is shown with a black point.

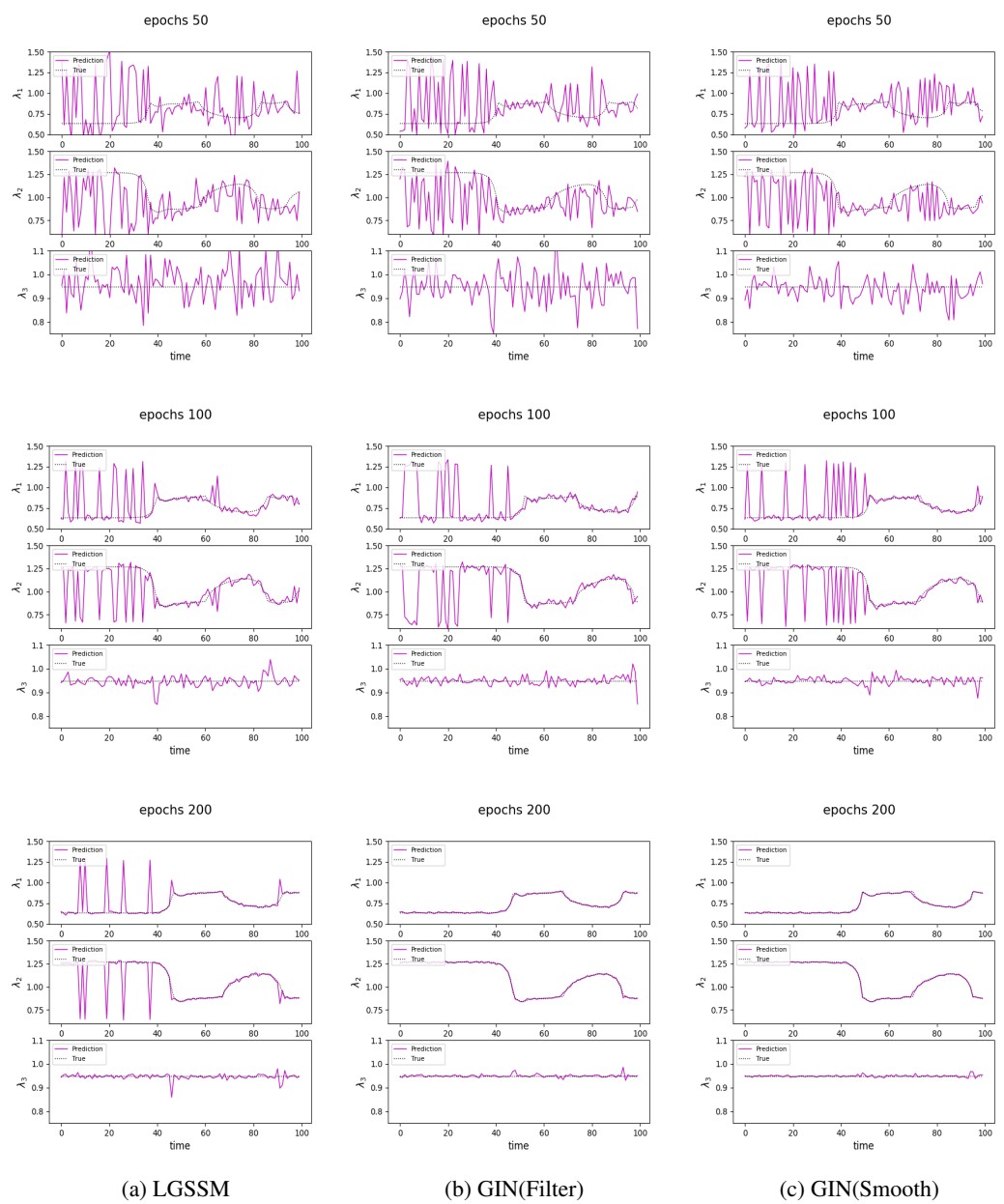

(a) LGSSM                    (b) GIN(Filter)                    (c) GIN(Smooth)

Figure 42: Eigenvalues of the learned transition matrix $\hat{\mathbf{F}}_t$ and their corresponding true values in the first 100 time steps for Lorenz attractor experiment. Despite the low dimensional experiments in the paper that we give the dynamics $(\mathbf{F}, \mathbf{H})$ to the model, here we show the GIN ability for learning the dynamics, when we do not provide the dynamics information, i.e. $(\mathbf{F}_t, \mathbf{H}_t)$ in (37).

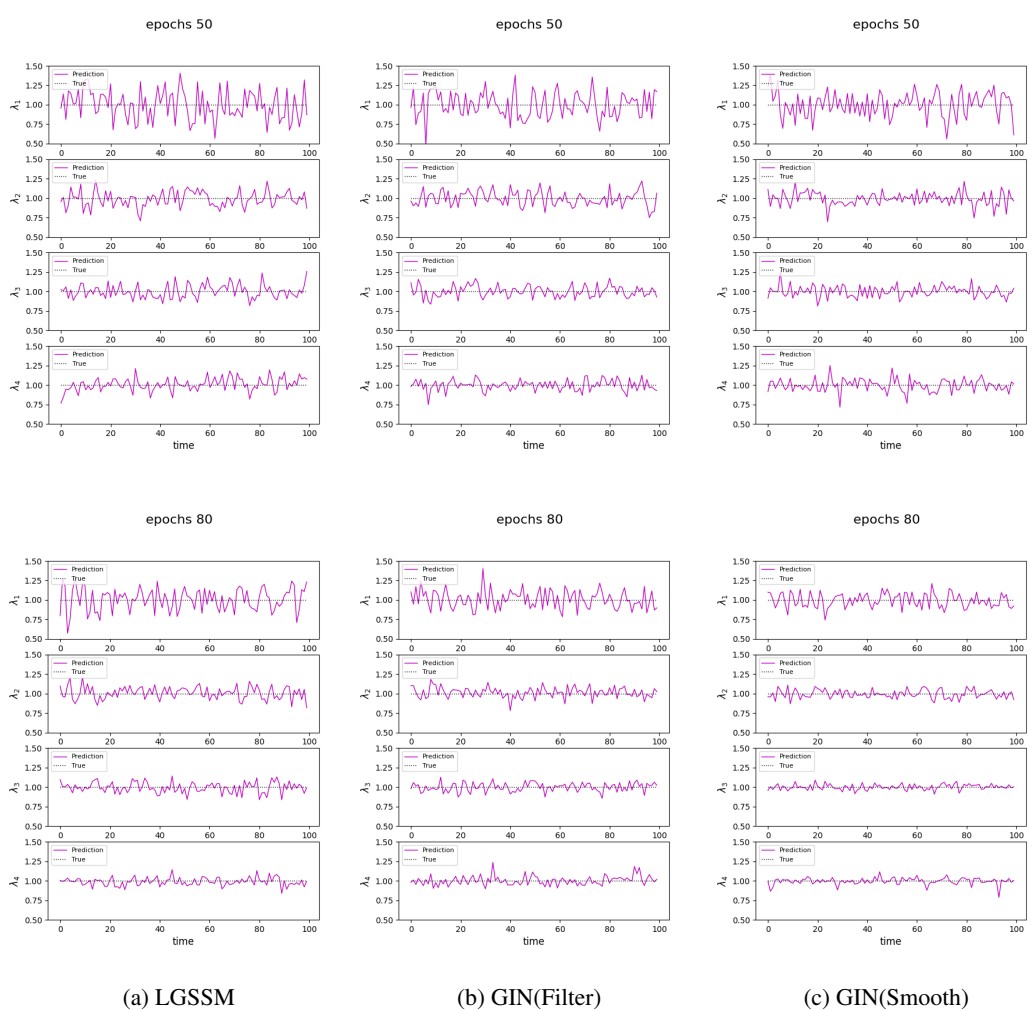

(a) LGSSM      (b) GIN(Filter)      (c) GIN(Smooth)

Figure 44: Eigenvalues of the learned transition matrix $\hat{\mathbf{F}}_t$ and their corresponding true values in the first 100 time steps for NCLT dataset experiment. Despite the low dimensional experiments in the paper that we provided the dynamics $(\mathbf{F}, \mathbf{H})$ for the model, here we show the GIN ability for learning the dynamics, when we do not provide the dynamics information, i.e. $(\mathbf{F}_t, \mathbf{H}_t)$ in (39).

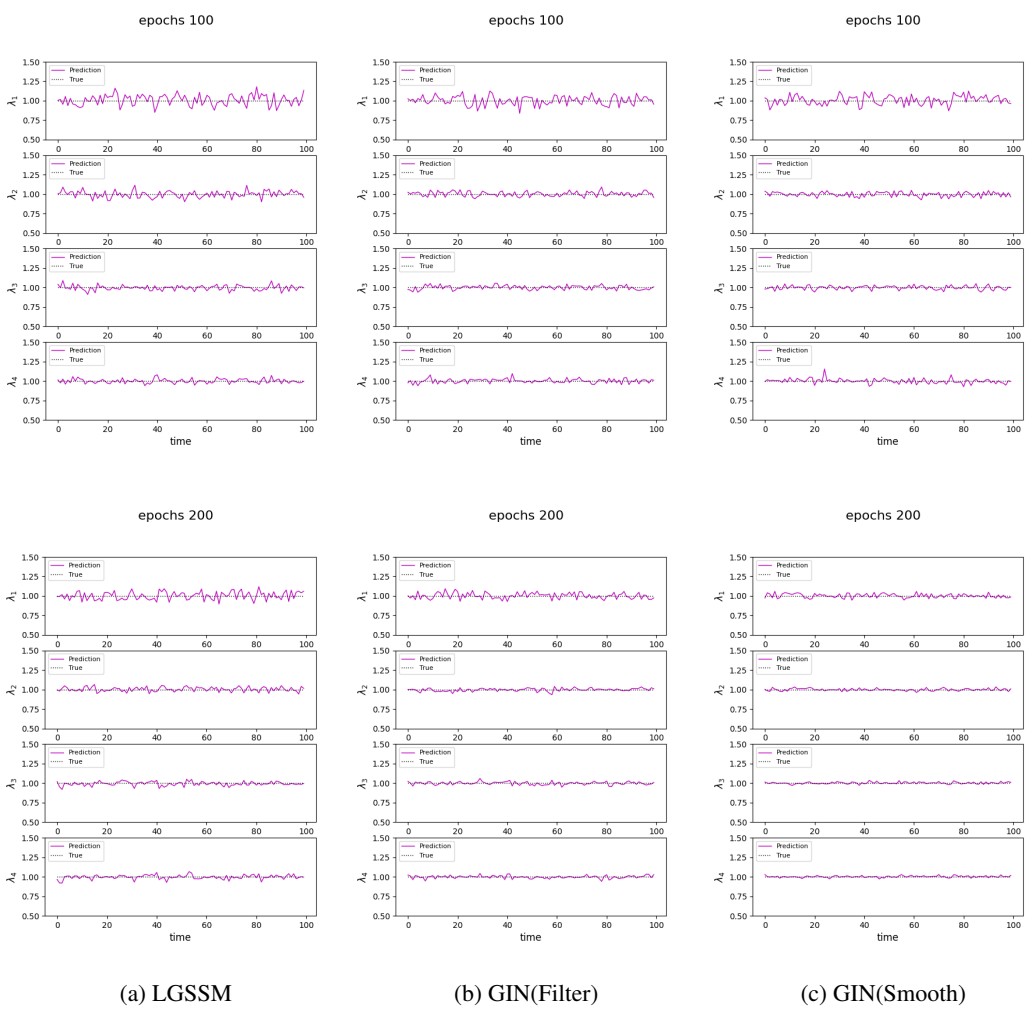

(a) LGSSM        (b) GIN(Filter)        (c) GIN(Smooth)

Figure 46: Eigenvalues of the learned transition matrix $\hat{\mathbf{F}}_t$ and their corresponding true values in the first 100 time steps for NCLT dataset experiment. Despite the low dimensional experiments in the paper that we provided the dynamics $(\mathbf{F}, \mathbf{H})$ for the model, here we show the GIN ability for learning the dynamics, when we do not provide the dynamics information, i.e. $(\mathbf{F}_t, \mathbf{H}_t)$ in (39).

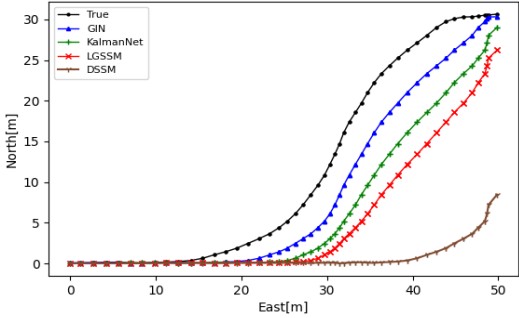

Figure 47: NCLT dataset position for the first 50 observations: ground truth positions and the generated trajectories with the GIN, LGSSM, KalmanNet and DSSM approaches are illustrated.

## A.10   MSE RESULTS FOR THE STATE ESTIMATION AND ESTIMATED KG-SG

We compare the learned KG-SG matrices via the GRU cells with their corresponding ground truth for the first 100 time steps of the low dimensional experiments. We calculate the element-wise squared difference of the learned KG and its ground truth, $\Delta\mathrm{KG}_t = \mathrm{Tr}\big((\hat{\mathrm{KG}}_t - \mathrm{KG}_t)^T(\hat{\mathrm{KG}}_t - \mathrm{KG}_t)\big)$, and take the average of all $\Delta\mathrm{KG}_t$, while similar procedure holds for the SG. The results are provided in table 11.

Table 11: Comparison of leaned KG-SG matrices and ground truth KG-SG. Lorenz attractor and NCLT experiments with dynamics refer to the situation, where are given the dynamics form, i.e. (37)-(39).

| Exp | 50 epoch | | 80 epoch | | 100 epoch | | 200 epoch | |
|---|---|---|---|---|---|---|---|---|
| | $\Delta$ KG | $\Delta$ SG | $\Delta$ KG | $\Delta$ SG | $\Delta$ KG | $\Delta$ SG | $\Delta$ KG | $\Delta$ SG |
| Lorenz with dynamics | 2.83 | 2.49 | 1.52 | 1.26 | 0.69 | 0.62 | 0.46 | 0.49 |
| NCLT with dynamics | 1.91 | 2.15 | 1.39 | 1.24 | 0.93 | 0.95 | 0.67 | 0.73 |
| Lorenz without dynamics | 12.31 | 13.26 | 8.55 | 7.93 | 5.40 | 6.29 | 4.19 | 4.58 |
| NCLT without dynamics | 9.88 | 9.62 | 6.73 | 7.11 | 5.11 | 4.83 | 3.29 | 3.10 |

The MSE results for the single and double pendulum experiments are in the table 12 and 13. In addition to (7), where $\mathbf{F}$ matrix includes the effects of the process noise, two other mentioned solutions introduced in section 4, are included in the MSE results as well. Using GRU cell and MLP for mapping $\mathbf{x}^+$, as their input, to $\mathbf{Q}$, as their output, where the former one is shown by GRU($\mathbf{Q}$) and the latter one by MLP($\mathbf{Q}$) in the tables.

Table 12: MSE for single pendulum experiment.

| Model | MSE |
|---|---|
| LSTM (units = 25, $m = 15$) | 0.092±0.003 |
| LSTM (units = 25, $m = 20$) | 0.092±0.005 |
| LSTM (units = 25, $m = 40$) | 0.090±0.005 |
| LSTM (units = 100, $m = 15$) | 0.089±0.002 |
| LSTM (units = 100, $m = 20$) | 0.089±0.005 |
| LSTM (units = 100, $m = 40$) | 0.090±0.004 |
| GRU (units = 30, $m = 15$) | 0.095±0.006 |
| GRU (units = 30, $m = 20$) | 0.093±0.002 |
| GRU (units = 30, $m = 40$) | 0.094±0.005 |
| GRU (units = 100, $m = 15$) | 0.091±0.002 |
| GRU (units = 100, $m = 20$) | 0.092±0.004 |
| GRU (units = 100, $m = 40$) | 0.091±0.008 |

| Model | $\hat{\mathbf{F}}(\mathbf{Q})$ | MLP($\mathbf{Q}$) | GRU($\mathbf{Q}$) |
|---|---|---|---|
| LGSSM filter($m = 15, n = 30, K = 10$) | 0.089±0.009 | 0.088±0.005 | 0.088±0.006 |
| LGSSM filter($m = 15, n = 30, K = 15$) | 0.088±0.011 | 0.087±0.007 | 0.086±0.004 |
| LGSSM filter($m = 15, n = 45, K = 10$) | 0.085±0.004 | 0.084±0.007 | 0.084±0.009 |
| LGSSM filter($m = 15, n = 45, K = 15$) | 0.084±0.005 | 0.083±0.004 | 0.082±0.004 |
| LGSSM filter($m = 20, n = 40, K = 10$) | 0.084±0.009 | 0.082±0.014 | 0.082±0.011 |
| LGSSM filter($m = 20, n = 40, K = 15$) | 0.083±0.012 | 0.081±0.005 | 0.080±0.014 |
| LGSSM filter($m = 20, n = 60, K = 10$) | 0.079±0.005 | 0.078±0.012 | 0.076±0.009 |
| LGSSM filter($m = 20, n = 60, K = 15$) | 0.077±0.006 | 0.075±0.011 | 0.074±0.008 |
| LGSSM smooth($m = 15, n = 30, K = 10$) | 0.086±0.011 | 0.083±0.004 | 0.084±0.007 |
| LGSSM smooth($m = 15, n = 30, K = 15$) | 0.085±0.012 | 0.084±0.008 | 0.083±0.012 |
| LGSSM smooth($m = 15, n = 45, K = 10$) | 0.081±0.008 | 0.080±0.009 | 0.079±0.003 |
| LGSSM smooth($m = 15, n = 45, K = 15$) | 0.081±0.014 | 0.078±0.007 | 0.077±0.011 |
| LGSSM smooth($m = 20, n = 40, K = 10$) | 0.082±0.005 | 0.078±0.004 | 0.076±0.013 |
| LGSSM smooth($m = 20, n = 40, K = 15$) | 0.080±0.003 | 0.076±0.006 | 0.074±0.010 |
| LGSSM smooth($m = 20, n = 60, K = 10$) | 0.076±0.008 | 0.073±0.002 | 0.070±0.009 |
| LGSSM smooth($m = 20, n = 60, K = 15$) | 0.073±0.013 | 0.071±0.011 | 0.068±0.013 |
| GIN filter($m = 15, n = 30, K = 10$) | 0.078±0.013 | 0.076±0.005 | 0.075±0.004 |
| GIN filter($m = 15, n = 30, K = 15$) | 0.078±0.014 | 0.075±0.009 | 0.074±0.012 |
| GIN filter($m = 15, n = 45, K = 10$) | 0.074±0.010 | 0.073±0.008 | 0.072±0.009 |
| GIN filter($m = 15, n = 45, K = 15$) | 0.073±0.015 | 0.074±0.011 | 0.071±0.005 |
| GIN filter($m = 20, n = 40, K = 10$) | 0.072±0.005 | 0.072±0.008 | 0.070±0.002 |
| GIN filter($m = 20, n = 40, K = 15$) | 0.071±0.007 | 0.071±0.004 | 0.071±0.009 |
| GIN filter($m = 20, n = 60, K = 10$) | 0.067±0.009 | 0.066±0.005 | 0.065±0.006 |
| GIN filter($m = 20, n = 60, K = 15$) | 0.065±0.013 | 0.064±0.009 | 0.063±0.010 |
| GIN smooth($m = 15, n = 30, K = 10$) | 0.071±0.007 | 0.070±0.003 | 0.068±0.009 |
| GIN smooth($m = 15, n = 30, K = 15$) | 0.070±0.008 | 0.068±0.011 | 0.068±0.007 |
| GIN smooth($m = 15, n = 45, K = 10$) | 0.065±0.011 | 0.065±0.009 | 0.064±0.012 |
| GIN smooth($m = 15, n = 45, K = 15$) | 0.064±0.008 | 0.066±0.007 | 0.063±0.009 |
| GIN smooth($m = 20, n = 40, K = 10$) | 0.064±0.005 | 0.065±0.003 | 0.062±0.008 |
| GIN smooth($m = 20, n = 40, K = 15$) | 0.063±0.004 | 0.064±0.011 | 0.063±0.007 |
| GIN smooth($m = 20, n = 60, K = 10$) | 0.059±0.009 | 0.061±0.012 | 0.057±0.006 |
| GIN smooth($m = 20, n = 60, K = 15$) | 0.058±0.005 | 0.057±0.009 | 0.056±0.004 |

Table 13: MSE for double pendulum experiment.

| Model | MSE |
|---|---|
| LSTM (units = 50, $m = 15$) | 0.172±0.012 |
| LSTM (units = 50, $m = 20$) | 0.166±0.009 |
| LSTM (units = 50, $m = 40$) | 0.167±0.011 |
| LSTM (units = 100, $m = 15$) | 0.164±0.006 |
| LSTM (units = 100, $m = 20$) | 0.162±0.009 |
| LSTM (units = 100, $m = 40$) | 0.159±0.010 |
| GRU (units = 50, $m = 15$) | 0.194±0.014 |
| GRU (units = 50, $m = 20$) | 0.189±0.013 |
| GRU (units = 50, $m = 40$) | 0.188±0.015 |
| GRU (units = 100, $m = 15$) | 0.173±0.009 |
| GRU (units = 100, $m = 20$) | 0.169±0.014 |
| GRU (units = 100, $m = 40$) | 0.166±0.018 |

| Model | $\hat{\mathbf{F}}(\mathbf{Q})$ | MLP($\mathbf{Q}$) | GRU($\mathbf{Q}$) |
|---|---|---|---|
| LGSSM filter($m = 15$, $n = 30$, $K = 10$) | 0.154±0.013 | 0.159±0.021 | 0.153±0.009 |
| LGSSM filter($m = 15$, $n = 30$, $K = 15$) | 0.152±0.008 | 0.153±0.010 | 0.152±0.012 |
| LGSSM filter($m = 15$, $n = 45$, $K = 10$) | 0.144±0.011 | 0.141±0.015 | 0.139±0.013 |
| LGSSM filter($m = 15$, $n = 45$, $K = 15$) | 0.142±0.007 | 0.138±0.012 | 0.137±0.017 |
| LGSSM filter($m = 20$, $n = 40$, $K = 10$) | 0.144±0.012 | 0.137±0.009 | 0.138±0.013 |
| LGSSM filter($m = 20$, $n = 40$, $K = 15$) | 0.141±0.007 | 0.137±0.008 | 0.136±0.016 |
| LGSSM filter($m = 20$, $n = 60$, $K = 10$) | 0.129±0.009 | 0.126±0.014 | 0.122±0.015 |
| LGSSM filter($m = 20$, $n = 60$, $K = 15$) | 0.127±0.012 | 0.124±0.013 | 0.119±0.009 |
| LGSSM smooth($m = 15$, $n = 30$, $K = 10$) | 0.147±0.009 | 0.148±0.014 | 0.144±0.015 |
| LGSSM smooth($m = 15$, $n = 30$, $K = 15$) | 0.146±0.014 | 0.146±0.013 | 0.142±0.017 |
| LGSSM smooth($m = 15$, $n = 45$, $K = 10$) | 0.139±0.017 | 0.136±0.009 | 0.133±0.017 |
| LGSSM smooth($m = 15$, $n = 45$, $K = 15$) | 0.137±0.009 | 0.135±0.017 | 0.133±0.012 |
| LGSSM smooth($m = 20$, $n = 40$, $K = 10$) | 0.136±0.014 | 0.131±0.022 | 0.132±0.011 |
| LGSSM smooth($m = 20$, $n = 40$, $K = 15$) | 0.134±0.011 | 0.129±0.014 | 0.129±0.022 |
| LGSSM smooth($m = 20$, $n = 60$, $K = 10$) | 0.123±0.019 | 0.116±0.016 | 0.115±0.013 |
| LGSSM smooth($m = 20$, $n = 60$, $K = 15$) | 0.120±0.010 | 0.112±0.009 | 0.108±0.014 |
| GIN filter($m = 15$, $n = 30$, $K = 10$) | 0.126±0.014 | 0.125±0.012 | 0.125±0.011 |
| GIN filter($m = 15$, $n = 30$, $K = 15$) | 0.124±0.015 | 0.124±0.019 | 0.121±0.009 |
| GIN filter($m = 15$, $n = 45$, $K = 10$) | 0.115±0.011 | 0.114±0.015 | 0.110±0.017 |
| GIN filter($m = 15$, $n = 45$, $K = 15$) | 0.114±0.019 | 0.112±0.020 | 0.110±0.011 |
| GIN filter($m = 20$, $n = 40$, $K = 10$) | 0.113±0.013 | 0.111±0.009 | 0.111±0.013 |
| GIN filter($m = 20$, $n = 40$, $K = 15$) | 0.111±0.009 | 0.109±0.018 | 0.108±0.009 |
| GIN filter($m = 20$, $n = 60$, $K = 10$) | 0.099±0.018 | 0.094±0.017 | 0.095±0.021 |
| GIN filter($m = 20$, $n = 60$, $K = 15$) | 0.097±0.009 | 0.093±0.009 | 0.091±0.008 |
| GIN smooth($m = 15$, $n = 30$, $K = 10$) | 0.115±0.011 | 0.116±0.009 | 0.113±0.017 |
| GIN smooth($m = 15$, $n = 30$, $K = 15$) | 0.113±0.015 | 0.113±0.018 | 0.112±0.014 |
| GIN smooth($m = 15$, $n = 45$, $K = 10$) | 0.105±0.009 | 0.101±0.014 | 0.101±0.015 |
| GIN smooth($m = 15$, $n = 45$, $K = 15$) | 0.102±0.013 | 0.100±0.011 | 0.098±0.008 |
| GIN smooth($m = 20$, $n = 40$, $K = 10$) | 0.101±0.008 | 0.098±0.010 | 0.094±0.015 |
| GIN smooth($m = 20$, $n = 40$, $K = 15$) | 0.098±0.017 | 0.095±0.014 | 0.092±0.007 |
| GIN smooth($m = 20$, $n = 60$, $K = 10$) | 0.086±0.013 | 0.081±0.008 | 0.079±0.009 |
| GIN smooth($m = 20$, $n = 60$, $K = 15$) | 0.083±0.009 | 0.079±0.006 | 0.076±0.013 |

## A.11 ALGORITHMS

---

**Algorithm** High-Dimensional Observations Training

---

**Input:** Ground Truth $\mathbf{gt}_{1:T}$, Observations $\mathbf{o}_{1:T}$, last posteriors $(\mathbf{x}_{1:T}^+, \mathbf{\Sigma}_{1:T}^+)$, initial posterior $(\mathbf{x}_0^+, \mathbf{\Sigma}_0^+)$
$\alpha_{1:T}$ = *Dynamics Network* $(\mathbf{x}_{0:T-1}^+)$
Obtain $\hat{\mathbf{F}}_{1:T}$ and $\hat{\mathbf{H}}_{1:T}$ by (13)
$(\mathbf{x}_{1:T}^-, \mathbf{\Sigma}_{1:T}^-)$ = *Prediction Step* $((\mathbf{x}_{0:T-1}^+, \mathbf{\Sigma}_{0:T-1}^+), \hat{\mathbf{F}}_{1:T})$
$(\mathbf{w}_{1:T}, \mathbf{r}_{1:T})$ = *encoder* $(\mathbf{o}_{1:T})$
$\mathbf{K}_{1:T} = \mathbf{\Sigma}_{1:T}^- \hat{\mathbf{H}}_{1:T} \mathbf{M}_{1:T} \mathbf{M}_{1:T}^T, \quad \mathbf{M}_{1:T} = GRU^{KG}(Conv2D(\mathbf{\Sigma}_{1:T}^-), \mathbf{r}_{1:T})$
$\mathbf{J}_{1:T} = \mathbf{\Sigma}_{1:T}^+ \hat{\mathbf{F}}_{1:T}^T \mathbf{N}_{1:T} \mathbf{N}_{1:T}^T, \quad \mathbf{N}_{1:T} = GRU^{SG}(Conv2D(\mathbf{\Sigma}_{1:T}^-))$
$(\mathbf{x}_{1:T}^+, \mathbf{\Sigma}_{1:T}^+)$ = *Filtering Step* $(\mathbf{x}_{1:T}^-, \mathbf{\Sigma}_{1:T}^-, \mathbf{K}_{1:T}, \mathbf{w}_{1:T}, \hat{\mathbf{H}}_{1:T})$
$(\mathbf{x}_{1:T|T}, \mathbf{\Sigma}_{1:T|T})$ = *Smoothing Step* $(\mathbf{x}_{1:T}^+, \mathbf{\Sigma}_{1:T}^+, \mathbf{J}_{1:T}, \hat{\mathbf{F}}_{1:T})$
$\mathbf{o}_{1:T}^+$ = *decoder* $(\mathbf{x}_{1:T|T}, \mathbf{\Sigma}_{1:T|T})$
$\mathcal{L}_{1:T}$ = - Likelihood $(\mathbf{gt}_{1:T}, \mathbf{o}_{1:T}^+)$
Backward Propagation ()

---

---

**Algorithm** Low-Dimensional Observations Training

---

**Input:** Ground Truth $\mathbf{gt}_{1:T}$, Observations $\mathbf{y}_{1:T}$, last posteriors $(\mathbf{x}_{1:T}^+, \mathbf{\Sigma}_{1:T}^+)$, initial posterior $(\mathbf{x}_0^+, \mathbf{\Sigma}_0^+)$
**if** Dynamics are not known **then**
    $\alpha_{1:T}$ = *Dynamics Network* $(\mathbf{x}_{0:T-1}^+)$
    Obtain $\hat{\mathbf{F}}_{1:T}$ and $\hat{\mathbf{H}}_{1:T}$ by (13)
    $(\mathbf{w}_{1:T}, \mathbf{r}_{1:T})$ = MLP $(\mathbf{y}_{1:T})$
    $(\mathbf{x}_{1:T}^-, \mathbf{\Sigma}_{1:T}^-)$ = *Prediction Step* $(\mathbf{x}_{0:T-1}^+, \mathbf{\Sigma}_{0:T-1}^+, \hat{\mathbf{F}}_{1:T})$
    $\mathbf{K}_{1:T} = \mathbf{\Sigma}_{1:T}^- \hat{\mathbf{H}}_{1:T} \mathbf{M}_{1:T} \mathbf{M}_{1:T}^T, \quad \mathbf{M}_{1:T} = GRU^{KG}(\mathbf{\Sigma}_{1:T}^-, \mathbf{r}_{1:T})$
    $\mathbf{J}_{1:T} = \mathbf{\Sigma}_{1:T}^+ \hat{\mathbf{F}}_{1:T}^T \mathbf{N}_{1:T} \mathbf{N}_{1:T}^T, \quad \mathbf{N}_{1:T} = GRU^{SG}(\mathbf{\Sigma}_{1:T}^-)$
    $(\mathbf{x}_{1:T}^+, \mathbf{\Sigma}_{1:T}^+)$ = *Filtering Step* $(\mathbf{x}_{1:T}^-, \mathbf{\Sigma}_{1:T}^-, \mathbf{K}_{1:T}, \mathbf{w}_{1:T}, \hat{\mathbf{H}}_{1:T})$
    $(\mathbf{x}_{1:T|T}, \mathbf{\Sigma}_{1:T|T})$ = *Smoothing Step* $(\mathbf{x}_{1:T}^+, \mathbf{\Sigma}_{1:T}^+, \mathbf{J}_{1:T}, \hat{\mathbf{F}}_{1:T})$
    $\mathbf{o}_{1:T}^+$ = MLP $(\mathbf{x}_{1:T|T}+, \mathbf{\Sigma}_{1:T|T})$
    $\mathcal{L}_{1:T}$ = - Likelihood $(\mathbf{gt}_{1:T}, \mathbf{o}_{1:T}^+)$
    Backward Propagation ()
**else**
    *Q network* = MLP $(\mathbf{x}_{0:T-1}^+)$ or GRU $(\mathbf{x}_{0:T-1}^+, \mathbf{Q}_{1:T})$
    $(\mathbf{F}_{1:T}, \mathbf{H}_{1:T})$ = Dynamics
    $\mathbf{r}_{1:T}$ = *trainable layer*$(\mathbf{y}_{1:T})$
    $\mathbf{q}_{1:T}$ = *Q network*$(\mathbf{x}_{0:T-1}^+)$
    $(\mathbf{x}_{1:T}^-, \mathbf{\Sigma}_{1:T}^-)$ = *Prediction Step* $((\mathbf{x}_{0:T-1}^+, \mathbf{\Sigma}_{0:T-1}^+), \mathbf{Q}_{1:T}, \mathbf{F}_{1:T})$
    $\mathbf{K}_{1:T} = \mathbf{\Sigma}_{1:T}^- \mathbf{H}_{1:T} \mathbf{M}_{1:T} \mathbf{M}_{1:T}^T, \quad \mathbf{M}_{1:T} = GRU^{KG}(\mathbf{\Sigma}_{1:T}^-, \mathbf{r}_{1:T})$
    $\mathbf{J}_{1:T} = \mathbf{\Sigma}_{1:T}^+ \mathbf{F}_{1:T}^T \mathbf{N}_{1:T} \mathbf{N}_{1:T}^T, \quad \mathbf{N}_{1:T} = GRU^{SG}(\mathbf{\Sigma}_{1:T}^-)$
    $(\mathbf{x}_{1:T}^+, \mathbf{\Sigma}_{1:T}^+)$ = *Filtering Step* $(\mathbf{x}_{1:T}^-, \mathbf{\Sigma}_{1:T}^-, \mathbf{K}_{1:T}, \mathbf{y}_{1:T}, \mathbf{H}_{1:T})$
    $(\mathbf{x}_{1:T|T}, \mathbf{\Sigma}_{1:T|T})$ = *Smoothing Step* $(\mathbf{x}_{1:T}^+, \mathbf{\Sigma}_{1:T}^+, \mathbf{J}_{1:T}, \mathbf{F}_{1:T})$
    $\sigma_{1:T|T}$ = *Trainable Layer* $(\mathbf{\Sigma}_{1:T|T})$
    $\mathbf{o}_{1:T}^+$ = $[\mathbf{x}_{1:T|T}, \sigma_{1:T|T}]$
    $\mathcal{L}_{1:T}$ = - Likelihood $(\mathbf{gt}_{1:T}, \mathbf{o}_{1:T}^+)$
    Backward Propagation ()
**end if**

---

### A.12 PYTHON INFERENCE CODE

To demonstrate the simplicity of our proposed GIN, we include intuitive inference code with Tensorflow library for both the high dimensional and low dimensional experiments. The code runs with Python 3.6+. The entire code to reproduce the experiments are available in Github repository.

#### A.12.1 PYTHON INTUITIVE CODE FOR HIGH DIMENSIONAL EXPERIMENTS.

```python
import tensorflow.keras as k
import Prediction
import Filtering
import Smoothing
import DynamicsNetwork
import Encoder
import Decoder
import DataGen

class GIN_CELL(k.layers.Layer):
    def __init__(self, initial_states):
        self.x_tm1_+, self.Sigma_tm1_+ = initial_states
        self.filter_states = [[self.x_tm1_+, self.Sigma_tm1_+]]
    def call(self, inputs):
        w_1:T, r_1:T = inputs
        for w_t, r_t in (w_1:T, r_1:T):
            F_hat_t, H_hat_t = DynamicsNetwork(self.x_tm1_+)
            x_t_-, Sigma_t_- = Prediction(F_hat_t, H_hat_t,...
            self.x_tm1_+, self.Sigma_tm1_+)
            x_t_+, Sigma_t_+ = Filtering(x_t_-, Sigma_t_-,...
            w_t, r_t, H_hat_t)
            self.filter_states.append([x_t_+, Sigma_t_+])
            self.x_tm1_+, self.Sigma_tm1_+ = x_t_+, Sigma_t_+
        x_1:T_T, Sigma_1:T_T = Smoothing(self.filter_states,...
        Sigma_1:T_-, F_hat_1:T)
        return x_1:T_T, Sigma_1:T_T

class GIN(k.models.Model):
    def __init__(self, initial_states):
        self.x_0_+, self.Sigma_0_+ = initial_states
        self.GIN_CELL_OBJ = self.GIN_CELL(self.x_0_+, self.Sigma_0_+)
        self.Encoder = Encoder
        self.Decoder = Decoder

    def call(self, o_1:T):
        w_1:T, r_1:T = self.Encoder(o_1:T)
        x_1:T_T, Sigma_1:T_T = self.GIN_CELL_OBJ(w_1:T, r_1:T)
        o_1:T_+ = self.Decoder(x_1:T_T, Sigma_1:T_T)
        return o_1:T_+

Data = DataGen()
o_1:T_+ = GIN(Data)
```

### A.12.2  PYTHON INTUITIVE CODE FOR LOW DIMENSIONAL EXPERIMENTS.

```python
import tensorflow.keras as k
import Prediction
import Filtering
import Smoothing
import MLP_ENC
import MLP_DEC
import DataGen
import Dynamics

class GIN_CELL(k.layers.Layer):
    def __init__(self, initial_states):
        self.x_tm1_+, self.Sigma_tm1_+ = initial_states
        self.filter_states = [[self.x_tm1_+, self.Sigma_tm1_+]]

    def call(self, inputs):
        w_1:T, r_1:T, F_1:T, H_1:T = inputs
        for w_t, r_t, F_t, H_t in (w_1:T, r_1:T, F_1:T, H_1:T):
            x_t_-, Sigma_t_- = Prediction(F_t, H_t, self.x_tm1_+,...
            self.Sigma_tm1_+)
            x_t_+, Sigma_t_+ = Filtering(x_t_-, Sigma_t_-, w_t, r_t, H_t)
            self.filter_states.append([x_t_+, Sigma_t_+])
            self.x_tm1_+, self.Sigma_tm1_+ = x_t_+, Sigma_t_+
        x_1:T_T, Sigma_1:T_T = Smoothing(self.filter_states,...
        Sigma_1:T_-, F_1:T)
        return x_1:T_T, Sigma_1:T_T

class GIN(k.models.Model):
    def __init__(self, initial_states, Dynamics):
        self.x_0_+, self.Sigma_0_+ = initial_states
        self.F_1:T, self.H_1:T = Dynamics
        self.GIN_CELL_OBJ = self.GIN_CELL(self.x_0_+, self.Sigma_0_+)
        self.MLP_ENC = MLP_ENC
        self.MLP_DEC = MLP_DEC

    def call(self, o_1:T):
        r_1:T = self.MLP_ENC(o_1:T)
        x_1:T_T, Sigma_1:T_T = self.GIN_CELL_OBJ(o_1:T, r_1:T,...
        self.F_1:T, self.H_1:T)
        Sigma_o_1:T_+ = self.MLP_DEC(Sigma_1:T_T)
        x_o_1:T_+ = x_1:T_T
        return x_o_1:T_+, Sigma_o_1:T_+

Data = DataGen()
Dynamics_Matrices = Dynamics()
x_o_1:T_+, Sigma_o_1:T_+ = GIN(Data, Dynamics_Matrices)
```

