# OpenReview forum: "Gated Inference Network: Inferencing and Learning State-Space Models"
_ICLR.cc/2023/Conference — Submitted to ICLR 2023_

### Official Review · Reviewer_SVRC · 2022-10-24

**Confidence:** 3
**Correctness:** 3
**Technical Novelty And Significance:** 4
**Empirical Novelty And Significance:** 3
**Recommendation:** 6

**Clarity, Quality, Novelty And Reproducibility:**

The paper is well written, easy to follow and quite detailed. The experimental setup and the method is very well described (except maybe the loss, assuming there is added regularization terms missing), includes a 24 page long appendix describing the experiments giving pseudocodes, code examples, and the detailed architecture description. This clearly result in good reproducibility.

While I am not perfectly up to date in deep learning extensions of classical Systems theory ideas (this is reflected in my confidence score) the model seems quite novel.

Questions:
What exactly this statement means (in the introduction):
"Moreover, in the variational inference approaches that usually implemented in the context of variational auto encoders
for dimension reduction, they do not have access to the loss directly and have to minimize its lower
bound instead, which reduce the ability of learning dynamics and affect the performance of the model."

While it is true that the exact Bayesian loss (the negative logarithm of the exact posterior over the model parameters) is not available, as it is not available for you either of course, but the expected likelihood of the reconstruction is part of the ELBO with the Kullback-Leibler regularizer. What ability is really reduced in VB relative to your approach? Can you clarify?

(MINOR IMPORTANCE) In the case of low dimensional observation spaces NeuralODE models [1,2,3] can be quite competitive (for example in the case of the Lorenz-system. Do you have a specific reason not to consider these?
[1] Rubanova, Yulia, Ricky TQ Chen, and David K. Duvenaud. "Latent ordinary differential equations for irregularly-sampled time series." Advances in neural information processing systems 32 (2019).
[2] De Brouwer, Edward, et al. "GRU-ODE-Bayes: Continuous modeling of sporadically-observed time series." Advances in neural information processing systems 32 (2019).
[3] Kidger, Patrick, et al. "Neural controlled differential equations for irregular time series." Advances in Neural Information Processing Systems 33 (2020): 6696-6707.

(MAJOR IMPORTANCE) Maximizing the equation (13) as likelihood seems problematic given choosing high covariance would always give high log likelihood. Do you use any regularization or a prior on the covariance?

**Strength And Weaknesses:**

Strengths:
- Ability to handle high dimensional inputs, such as images directly
- The method builds on existing knowledge from Systems theory, utilizing inductive biases from the field.
- The experimental setup and the method is very well described
- No costly matrix inversion in the dynamics part

Weaknesses:
- The case of observational uncertainty is solved in an ad-hoc way. While a Kalman filter uses some prior assumption about noise levels, here the un-encoded observations are point observations. It is clear that in the case of high dimensional input this problem is hard to solve better than it is addressed here, for low dimensional cases it may be possible.
- It is not exactly clear how the likelihood without regularization / priors is sufficient to learn a probabilistic representation.


**Summary Of The Paper:**

The paper suggests a state space model to handle both high and low dimensional observation spaces with highly nonlinear observation model, such as animated images of physical systems. The internal state representation is probabilistic, taking account uncertainties. The model is a nonlinear extension of the Hammersein-Wiener model, where the state transition is also allowed to be nonlinear.

**Summary Of The Review:**

The paper is well written and the method seems quite novel.  The claims against Variational Methods should be a bit more clarified. It seems the likelihood itself missing a regularization what a variational objective would have provided. Clarifying what is happening there would be quite illuminating.

---

> ### Author Response · Authors · 2022-11-15
> **Reply-1**
>
>
> We would like to thank you for the detailed review and appreciate your invested time.  In the following, we will address each question separately:
>
> *The case of observational uncertainty is solved in an ad-hoc way...*
>
> **R1**
>
> We agree that the observation noise is obtained in an ad-hoc manner, because inferring the observation noise for high dimensional experiments is difficult.
> However, for the low dimensional experiments we can infer the observation noise in two ways
>
>
>  (i) Using the similar ad-hoc way that maps the point observations into the observation noise with a non-linear transformation, i.e. MLP, that we used in our experiments.
>
>
>
> (ii) Similar to the proposed structure for learning and inferring the process noise, i.e. Eq (30)-(32), we can obtain a similar formulation for the observation noise.
>
> $R_t = \mathrm{Cov}(w_t) - H_t \Sigma_{t}^- H_t^T
> = \mathrm{Cov}(w_t) - H_t \big[ Q_{t} + F_{t}        \Sigma_{t-1}^+             F_{t}^T \big] H_t^T
>      $
>
> where the it can be furthered expanded to
>
> $
> R_t = \mathrm{Cov}(w_t) - H_t \bigg[ Q_{t} + F_{t}
>      \big[
>      \Sigma_{t-1}^- - K_{t-1}
>          [\mathrm{Cov}(w_{t-1})]
>      K_{t-1}^T
>      \big]
>      F_{t}^T \bigg] H_t^T = g(w_{t}, w_{t-1})
> $
>
> where $g(w_{t}, w_{t-1})$ can be modeled by another MLP with the input of $(w_{t}, w_{t-1})$ or their difference $(w_{t} - w_{t-1})$. It is possible to go one step further and expand the previous equation more
>
> $
> R_t = \mathrm{Cov}(w_t) - H_t \bigg[ Q_{t} + F_{t}
>      \big[
>      \Sigma_{t-1}^- - K_{t-1}
>          [H_{t-1} \Sigma_{t-1}^- H_{t-1}^T + R_{t-1}]
>      K_{t-1}^T
>      \big]
>      F_{t}^T \bigg] H_t^T
>      = g(w_{t},R_{t-1})
> $
>
> where $g(w_{t}, R_{t-1})$ can be modeled by a GRU cell with state dependency on $R_{t-1}$ and $w_{t}$. Such structure allows us to incorporate the prior knowledge of observation noise for its estimation.
>
>
> Although, we can obtain a more intuitive model for inferring the observation noise by (ii), but it enforces additional complexity to the overall model.
>
> *It is not exactly clear how the likelihood without regularization / priors ...*
>
> **R2**
>
> Please note that we have used Wishart distribution as a prior for our estimated covariance matrix, which pushes the estimated covariance toward a scale of identity matrix and the scale is a hyper parameter for each experiment. Such prior prevents the covariance to grow and accordingly prevents getting higher log-likelihood due to the high covariance. To prevent any confusion for the readers, we write an explanation about it in the Fitting section of the revision.
>
> *What exactly this statement means (in the introduction)...*
>
> **R3**
>
> In the VAE-based approaches, as they do not have access to the log likelihood (because the posterior distribution is not available), they propose an approximation of the posterior and try to maximize the lower bound of the log likelihood by introducing an additional latent variable with variational inference (There are several VAE based models developed, but we chose EKVAE for the comparison in our experiments because it outperforms other VAE-based models for the dynamics learning). However, we do not use variational inference in our approach and optimize the likelihood directly as it is shown with $o^+$ in figure 4. We agree that similar to VB approaches, we do not have access to the log-likelihood but the inference method in our approach is different from VB.
>
> *(MINOR IMPORTANCE) In the case of low dimensional...*
>
> **R4**
>
> Thank you for the mentioned works. Neural ODE-based approaches for learning the density function can be considered as a good comparison. However, as they calculate $-\mathrm{Tr}(\frac{\partial f}{\partial \mathbf{z}})$, [1], to measure the volume change resulted from change of variables, their approach may not be scalable for high dimensional experiments. For the low dimensional experiments, these approaches learn and infer the dynamics with data as they assume the dynamics are not available(similar to DSSM's performance in figure 7 for Lorenz attractor experiment). Accordingly, it is most likely that these approaches cannot provide competing results compare to the GIN or Hybrid GNN.
>
> *[1] Ricky T. Q. Chen, Yulia Rubanova, Jesse Bettencourt, David Duvenaud.
>     Neural Ordinary Differential Equations. Advances in Neural Information Processing Systems,
> 34, 2018.*

---

### Official Review · Reviewer_jiSL · 2022-10-25

**Confidence:** 3
**Correctness:** 3
**Technical Novelty And Significance:** 3
**Empirical Novelty And Significance:** 2
**Recommendation:** 5

**Clarity, Quality, Novelty And Reproducibility:**

I had a hard time following the notation in this paper. The flow of the presentation can largely be improved and the paper can largely benefit from some reorganization. In terms of novelty, deep Kalman filters and variational versions existed before. This paper introduces a specific parameterization in the same context. This specific parameterization did not exist before, but the overall framework is not novel. The results seem to be reproducible although I did not try to reproduce them myself.

**Strength And Weaknesses:**

**Weaknesses**

**Notation and wording**

Overall, the notation and wording of this paper are confusing and not standard. Please add a notation section and clarify what each variable is representing. More specific comments are below.

* What are latent noisy observations? If a variable is latent it means it's not observed, this wording is confusing for the reader, please fix this.
* What does the superscript + mean? This notation is not common in the statistics literature.
* My current understanding is that $x$ corresponds to dynamics, $w$ corresponds to transformed inputs, and $o$ denotes inputs. Since there are so many letters and new notations used in the paper it would largely benefit from a notation section and will help the reader understand the main contributions better.

**Statistical Grounding**

From a statistical viewpoint, after reading the paper multiple times it is still unclear to me what the generative model and its corresponding graphical model are in this framework. What inference algorithms are used and what approximations are made? More specific comments are below.

* What is the graphical or statistical model?
* Do we start by a model of $x_t|x_{t-1},w_{1:t-1}$?
* If so, does this determine a full joint distribution over all the variables?
* Starting from $x_t|x_{t-1},w_{1:t-1}$ do we first marginalize out $x_{t-1}$ and get $x_t|w_{1:t-1}$ and then to get $x_t|w_{1:t}$ after observing $w_t$?
* What happens if we don't want to use Gaussian distributions? For example, what if the input data is not continuous? Does this framework still hold? What parts do we need to change to adjust for different generative assumptions? If this works only for the Gaussian case this should somehow be reflected in the abstract to clarify the scope of the presented method.
* The Kalman framework seems to rely on linear approximations of the dynamics (where the matrices are given by nonlinear transformations of their inputs). Is the linear approximation a good enough local approximation?
* The paper seems to lack a solid theoretical or statistical grounding. What type of inference algorithm is used and what is the underlying statistical model? The inference algorithm (if theoretically valid) has to be approximate since the model has nonlinearities and we cannot integrate them out. It looks like the KG and SG matrices are approximated using GRUs for the high-dimensional cases. How do we know if the GRU functions are finding the right KG and SG? What is the justification for using identity functions for low-dimensional cases? How is this reflected in the inference algorithm?


**Presentation**

The presentation and the flow of the paper can be largely improved. I would recommend the authors use the following flow for their presentation.

* First write a summary of Kalman Filtering for completeness to set the context and familiarize the readers with notation (you can add a notation section before this).
* Write a statistical model, draw a graphical model, include parameters, random variables, and noise, and use a better naming convention (latent vs. observed, which variables are following what type of dynamics or equations.
* Write your parameterization (eq. 7-11) and which functions are approximated by GRUs and how this fixed the issue with KF such as matrix inversion, high-d data, covariance parameterization, include a note about what are the approximations to what cost function and how this relates to VI and other approximations.

**Comparisons**

* The method includes lots of matrix multiplications, how does the time complexity and scalability of your method compare to alternatives e.g. VAEs? Can you include a paragraph about the time complexity and report some results on the wall clock time for your methods and comparisons with others?
* The comparison against E2C sounds like a strawman, E2C is an RL algorithm where the representations are learned for a completely different purpose.
* Many other comparisons aren't included both in the introduction and in the compared methods such as SVAE, ARHMM, SLDS,
* For the results reported in tables, did you run all the models until convergence?
* How is the model complexity accounted for? How does the number of parameters across different models correspond to the number of units?
* You are comparing stochastic and deterministic models. This is not a fair comparison. There are various methods in the VAE literature with specific latent and approximating family structures and algorithms for improving the inference such as importance-weighted autoencoders for tighter bounds, etc. For other models such as ARHMMs and SLDS models, there are specific inference algorithms designed such as Laplace-EM to help with convergence and generalizations. These are not included in the comparisons.
* Test log-likelihoods are not comparable across different models.
* I don't quite understand the sample generation plots, do we want the variance to be small and more densely concentrated around the mean? Although a single parameter has generated the data, we might expect a larger set of parameters to produce similar observations. How should this be quantified properly?

**Minor Issues**
Eq 3, 4: what do you mean by proportional to?
Eq 5, "can be written as a function of $F_t$". This needs to be written down in the main text, it's an important piece of what follows in the manuscript.
I don't understand eq. 6, it is not a standard notation. Can you clarify this?


## Post Rebuttal

**Presentation** I thank the authors for their detailed responses and clarifications. After reading the author's responses and comments from other reviewers I still think that the paper will largely benefit from a new organization to help clarify it from a statistical viewpoint. Specifically, the graphical models and the notation is not consistent with the literature on state space models (R1, R2, R3, R4) and it makes it hard to identify the contributions and follow the notations. A common strategy for organizing a state space model is to first include a generative process (accompanied by its corresponding graphical model) where there is a clear distinction between random variables, parameters, choice of distributions, functions, etc (R5). Then determine the inference strategy (variational inference, max likelihood, max a posteriori, sampling-based inference, etc.) and determine what approximations are performed. This organization will clearly resolve questions such as the linearity of the model vs. approximation (R7). Including the literature of Kalman filtering as the base notation in the main text will help determine where this method lies in the literature of SSMs (R8). After reading this paper multiple times I still don't have a clear picture of what the generative process is and how this method lies in the statistical models of time series literature.

**Comparisons** My comments and concerns regarding the comparisons did not spark the introduction of new results and comparisons which is against my expectation (R11, R12). In addition, the existing results require more information to be accompanied for completeness such as (effective) model complexity (R13). I also didn't find some of my original questions to be addressed fully such as R15.

I do believe that the paper poses a really interesting perspective and contribution, and it would be beneficial work to share with the community. Hence I increase my score slightly and encourage the authors to do another iteration of refinement. However, I think the paper is not quite ready for prime time as it is.






**Summary Of The Paper:**

The paper proposes Gated Inference Network, a recurrent architecture for handling noise, missing data, filtering, and smoothing in time series. The methods build on the literature on Kalman Filters and use specific parameterizations for the gain and smoothing matrices representing them as nonlinear transformations of their inputs parameterized by GRUs. Comparisons are done against deterministic architectures such as LSTM and GRU and stochastic models with variational approaches. Both high-dimensional and low-dimensional synthetic datasets are considered for the comparisons showing the scalability of GIN compared to alternatives.

**Summary Of The Review:**

The paper could largely benefit from a clear organization and logical flow. Comparisons are not comprehensive and some methods are not considered. The methods presented in the paper lack a solid theoretical and statistical grounding, making it difficult to compare against the existing statistical literature on time series models.

---

> ### Author Response · Authors · 2022-11-15
> **Reply-1**
>
>
> Thank you for your comprehensive comments and invested time.  In the following, we will address each question separately:
>
> *What are latent noisy observations? If a variable*
>
> **R1**
>
> In order to prevent any confusion, we have fixed this and change it to 'transferred observation'. In other words, $w_{1:T}$ is called transferred observation throughout the paper.
>
> *What does the superscript + mean?...*
>
> **R2**
>
> We have used superscript + to represent the posterior state mean and covariance, i.e. the mean vector is determined with $x^+$ and the covariance matrix is $\Sigma^+$.
> We have borrowed similar notation from [1], while for the flow of the paper we use similar organization from [6].
>
> *My current understanding is that...*
>
> **R3**
>
> For more clarity, we have provided notation information in the beginning of Parameterization section. We use $o_{1:T}$ as observations, $w_{1:T}$ as the transferred observations, $x_{1:T}$ as the states and $(F_{1:T},H_{1:T})$ as the dynamics of the system. Other parameters and variables like noises are explained and the notations are defined in the revision.
>
> *What is the graphical or statistical model? *
>
> **R4**
>
> We have added the graphical models for high and low dimensional experiments in the figure 3 and 9, respectively. The reason for the different models is that in the lack of dynamics    $(F_{1:T},H_{1:T})$ (in the high dimensional experiments), we need to learn and infer the dynamics as well. But in the presence of dynamics (low dimensional experiments), we already obtain them (Eq (37)-(39)). In the following, we explain each one:
>
> - High dimensional observations graphical model (figure 3): The dynamics in time step $t$ have dependency from the state in time $t-1$, where this dependency is modeled with *Dynamics Network* in the paper. Because of that we have a connection from $x_{t-1}$ to $dyn_{t}=(F_{t},H_{t})$.
>     In order to obtain $x_{t}$ in GSSMs, we need the inferred dynamics and $x_{t-1}$ so that we have a connection from $dyn_{t}$ and  $x_{t-1}$ to $x_{t}$.
>     It is worth noting that based on the 'learning process noise' subsection and Eq (6), we argue that the process noise effect, $Q$, is included in the learned transition matrix $\hat{F}$, thus we do not include $q$ variable for the process noise into the graphical model. However, we can infer the process noise separately as well, which is elaborated in Eq (30) and Eq (32). The corresponding results when we infer the process noise separately with Eq (30) are in table 12 and 13 in the third column. While, the corresponding results when we infer the process noise separately with Eq (32) are in table 12 and 13 in the second column.
>
> - Low dimensional observations graphical model (figure 9): Including the effects of the process noise $Q$ into the learned transition matrix $\hat{F}$ is not feasible in this case since we do not learn the dynamics as they are already available (Eq (37)-(39)). Thus, we can model the system as figure 9b to infer the process noise via eq (30), or we can model the system as figure 9a to infer the process noise via eq (32). Please note that if we do not have access to the dynamics $(F_{t},H_{t})$ or lack them for any reason, we can switch the model structure to figure 3 in order to learn the dynamics.
>
> *Do we start by a model of $x|x_{t-1}$ ...*
>
> **R5**
>
> We start by $x_{t}|x_{t-1}, w_{1:t-1}$ in the filtering step, where we get the transferred observation as $w_{1:t-1}|o_{1:t-1}$ and by marginalizing out $w_{1:t-1}$, we get $x_{t}|x_{t-1}, o_{1:t-1}$. Then in each time, first we marginalize out $x_{t-1}$ and then by observing $o_t$ (or equivalently $w_t$ and marginalizing it out), we obtain $x_t|o_{1:t}$. After observing all observations $o_{1:T}$, we go through the smoothing step where we obtain $x_t|o_{1:T}$. Finally, we get $o_{1:T}^+|x_{1:T}, o_{1:T}$ and by marginalizing out $x_{1:T}$, we get $o_{1:T}^+|o_{1:T}$, which is obtained in the fitting section.
>
> *What happens if we don't want to use Gaussian*
>
> **R6**
>
>  Similar to [7], we can consider any distribution and accordingly modify $e(.)$ and $d(.)$
> structures such that they output the parameters of that distribution. However, we can perform similar idea of End-to-end Optimized Image Compression [7] for the discrete data by adding some uniform noise to the quantized variables in order to obtain a soft continues distribution ready for calculating gradients.
>
> *The Kalman framework seems to rely on linear approximations...*
>
> **R7**
>
>  $\hat{F}(x_{t})$ and $\hat{H}(x_{t})$ are state dependent and non-linear functions of the states, which separates the GIN from EKF-UKF.
> We learn $K$ state transition and emission matrices $\hat{F}^k (x_{t-1}^+)$ and $\hat{H}^k(x_{t-1}^+)$, and combine each one with the state dependent coefficient $\alpha^k(x_{t-1}^+$),
> where a separated neural network with softmax activation is utilized to learn $\alpha^k(x_{t-1}^+)$ that we call it *Dynamics Network*.

---

> ### Author Response · Authors · 2022-11-15
> **Reply-2**
>
> *It looks like the KG and SG matrices are approximated using GRUs...*
>
> **R8**
>
> As we maximize the log likelihood, we can calculate the gradients w.r.t KG-SG in each time step with (truncated) BPTT. We expect the approximated KG and SG matrices by GRUs converge to the right KG -SG after some convincing number of iterations. To address your concern better, we compare the learned KG-SG matrices via the GRU cells with their corresponding ground truth for
> the first 100-time steps of the low dimensional experiments(here we do not provide the system dynamics for the model to show its ability for learning the dynamics), where the ground truth KG-SG are calculated by systems dynamics in Eq (37)-(39). We calculate the element-wise squared
> difference of the learned KG-SG and their ground truth and
> take the average of all errors. We report these errors for 50,80,100 and 200 iterations in table 11 in the revision.
>
>
> It should be noted that for the high dimensional experiments, as we assume the dynamics are not available and we model the states and dynamics with higher dimensions(for example in the pendulum experiment we set the state size to $n$ and $F \in
> R^{n \times n}$), it is not intuitive to compare the higher dimensional learned KG-SG with low dimensional systems KG-SG. But we can sample from the constructed distributions to check their Gaussianity.
> One of the reasons that we sample from the constructed distributions, i.e. figure 8 and figures 16-39, is that to show after training phase, the Gaussianity of the states are almost preserved.
>
> For the low dimensional observations, we use identity functions to model $e(.)$ and $d(.)$, i.e. $w=o$ and $o^+=x$, because we do not need to map the observation into smaller space. However, we need to model the observation noise $R$ and the prediction covariance, where we use separated MLPs for each one. Please see figure 11 as we draw the structure with details.
>
> *The presentation and the flow of the paper...*
>
> **R9**
>
> Thank you for the recommendation about the organization of the paper. We apply some changes that summarize them in the following.
> In the beginning of the parameterization section, we refer the readers to the background materials including the classic Kalman filter-smooth equations in the appendix (because of large number of results and figures and the page limitation). Then, we provide notations, which explain all the parameters and notations we have used in the paper. We have modified all of the confusing names and words, e.g. latent obs to transferred obs. We added the graphical model for both high and low dimensional experiments in figure 3 and 9 respectively, including the parameters inside them.
>
> *The method includes lots of matrix multiplications...*
>
> **R10**
>
> Tables 7 and 8 include the information you mentioned, where we report the number of the parameters and running time for 1 iteration for each model.  In the first row of each model
> structure in the high dimensional experiments, we set the number of their parameters approximately equal to our GIN to indicate the outperformance of the GIN in both log-likelihood and running time aspects. Extra running time of variational approaches, like KVAE, is due to employing
> classic Bayesian equations because it increases the running time substantially, when dealing with higher dimensional observations. With the same state size, the parameters complexity of the GIN is noticeably lower than other memory cells, e.g. LSTM and GRU, and
> variational methods as we convert high dimensional sparse covariance matrices into lower dimensional covariance matrices by employing convolutional operator, where it allows us to reduce the parameters of internal GRU cells quadratically.
>
> *Many other comparisons aren't included...*
>
> **R11**
>
> In the related works section, we add a paragraph about SLDS models with a brief explanation about their structures [9][10]. Likewise, we give a summary of ARHMM models [11] and potential drawbacks of them. Then, we explain how another ARHMM [12] model addresses the problems of the conventional ones. Additionally, we insert the mentioned papers in the table 9 for an overall comparison. It is worth noting that there is one connection between the GIN and SLDSs. Similar to [10], we are modeling $K$ dynamics sets, where each set has dependency from the state and is shown as $(\hat{F}^k,\hat{H}^k)$. Each set models different dynamics, that will dominate when the corresponding element of the dynamics network is high. Despite the SLDS models that select the switches as discrete values, we consider a soft combination of all switches, i.e. $K$ sets of dynamics, where the coefficient of each switch is determined by *Dynamics Network*.

---

> ### Author Response · Authors · 2022-11-15
> **Reply-3**
>
> *You are comparing stochastic and deterministic models...*
>
> **R12**
>
> We agree that the comparison of stochastic and deterministic approaches is not fair. But, for the completeness of the comparisons we have included all of them, where a similar but relatively concise tables are used in [1] and [6] that we expand them with more approaches. Among the VI based approaches for learning the dynamics of the system, we found out that the EKVAE [13] approach outperforms many other VI-based approaches and DSSMs papers. They show that maximizing
> the evidence lower bound may not infer and learn the underlying dynamics appropriately. So that they propose a constrained optimization framework as a general approach for training DSSMs, where they introduce the extended Kalman VAE and
> combine extended Kalman filtering/smoothing with amortized variational inference and a neural linearization. They also propose a method for appropriate initialization of the state parameters. Therefore, we select their approach as a good candidate for the sake of comparison in the experiments.
>
> *How is the model complexity accounted for...*
>
> **R13**
>
> For the model complexity we consider two aspects:(i) empirical running time (ii) the number of parameters. In order to check the timing complexity, we set the number of the parameters for each model almost same and measure the iteration time for each one. To check the parameters complexity, we assign the same state size to all of the models. Because in the GIN we code the sparse covariance matrices into smaller versions with convolutional operator, we prevent the complexity of the GRU parameters grow quadratically and the complexity is comparatively lower. Further details are in tables 7 and 8.
> The units number for the LSTM and GRU model in the tables 1 and 2 corresponds to their unit number. While $n$ is the state size and $m$ is the transferred observation size. For example, in the first row of the table 1, we have LSTM with 50 units and the input size of the LSTM is $m$, which is shared among all models.
>
> *For the results reported in tables...*
>
> **R14**
>
> The results are run until the convergence with the recommended hyper parameters of each paper.
>
> *I don't quite understand the sample generation...*
>
> **R15**
>
> We have three goals by sampling from the learned state distribution
>
> - After convincing number of epochs, the right KG-SG matrices are modeled such that the states are Gaussian.
>
> - The generated samples be close to the ground truth
>
> - The uncertainty of the samples generated by the GIN be lower than its counterparts in order to (i) be closer to the ground truth and (ii) prevent high fake log-likelihood because of huge uncertainty.
>
> We also include a comparison between the learned dynamics and the ground truth dynamics for 100 consecutive time steps. This may address your concern about the comparison between a set of learned dynamics and their ground truth. The comparisons are attached in the appendix figures 40-46, where we compare the eigenvalues of the learned dynamics with those of the ground truth.
>
> *Eq 3, 4: what do you mean by ...*
>
> **R16**
>
> In the Eq (3)-(4), the proportional means that by having a non-linear transformations like $f(.)$ and $g(.)$, we can map $(\Sigma_t^-, R_t)$ to KG and $\Sigma_{t+1}^-$ to SG such that KG=$f(\Sigma_t^-, R_t)$ and SG=$g(\Sigma_{t+1}^-)$.
> We have added the equation, in which $Q_{t}$ is written as a function of $F_t$ in the main text. In Eq (6), which is Eq (7) in the revised version, we are obtaining the covariance of the prior state at time $t$, i.e. $x_{t}|w_{1:t-1}$.
>  In order to do this, we get $x_{t}|x_{t-1},w_{1:t-1} = \mathcal{N}(F_{t} x_{t-1},Q_t)$ and marginalize out $x_{t-1}$ to get  $x_{t}|w_{1:t-1}=\mathcal{N}(F_{t} x_{t-1}^+ , F_{t} \Sigma_{t-1}^+ F_{t}^T+ Q_t)
> =\mathcal{N}(x_{t}^-,  \Sigma_{t}^-)$.
>
> As we explained in Eq (5)-(6), we can learn the transition matrix $\hat{F}_{t}$ such that it learns the effects of $Q_t$ as well.
>
> This means that we can write $\Sigma_{t}^- $ as  Eq (7) (in the revision).
>
> *[1] Philipp Becker, Harit Pandya, Gregor Gebhardt, Cheng Zhao, C James Taylor, and Gerhard Neumann.
>     Recurrent kalman networks: Factorized inference in high-dimensional deep feature spaces. In
>     International Conference on Machine Learning, pp. 544–552. PMLR, 2019,*
>
>
> *[2] Marco Fraccaro, Simon Kamronn, Ulrich Paquet, Ole Winther. A Disentangled Recognition and Nonlinear Dynamics Model for Unsupervised Learning. Part of Advances in Neural Information Processing Systems 30 (NIPS 2017).*
>
> *[3] Yuri Burda, Roger Grosse, Ruslan Salakhutdinov. Importance Weighted Autoencoders. Submitted to ICLR 2015.*
>
> *[4] Guy Revach, Nir Shlezinger, Xiaoyong Ni, Adria Lopez Escoriza, Ruud JG van Sloun, and Yonina C Eldar. Kalmannet: Neural network aided kalman filtering for partially known dynamics.*

---

> ### Author Response · Authors · 2022-11-15
> **Reply-4**
>
> *[5] Alexej Klushyn, Richard Kurle, Maximilian Soelch, Botond Cseke, and Patrick van der Smagt. Latent
> matters: Learning deep state-space models. Advances in Neural Information Processing Systems,
> 34, 2021.*
>
> *[6] David Ruhe and Patrick Forré. Self-supervised inference in state-space models. Submitted to ICLR 2022.*
>
> *[7] Diederik P Kingma and Max Welling. Auto-encoding variational bayes. arXiv preprint
> arXiv:1312.6114, 2013.*
>
> *[8] Johannes Ballé, Valero Laparra, Eero P. Simoncelli. End-to-end Optimized Image Compression. Submitted to ICLR 2017.*
>
> *[9] Zoubin Ghahramani and Geoffrey E Hinton. Variational learning for switching state-space models.
> Neural computation, 12(4):831–864, 2000.*
>
> *[10] Scott Linderman, Matthew Johnson, Andrew Miller, Ryan Adams, David Blei, and Liam Paninski.
> Bayesian learning and inference in recurrent switching linear dynamical systems. In Artificial
> Intelligence and Statistics, pp. 914–922. PMLR, 2017.*
>
> *[11] David Salinas, Valentin Flunkert, Jan Gasthaus, and Tim Januschowski. Deepar: Probabilistic
> forecasting with autoregressive recurrent networks. International Journal of Forecasting, 36(3):
> 1181–1191, 2020.*
>
> *[12] Syama Sundar Rangapuram, Matthias W Seeger, Jan Gasthaus, Lorenzo Stella, Yuyang Wang, and
> Tim Januschowski. Deep state space models for time series forecasting. Advances in neural
> information processing systems, 31, 2018.*
>
> *[13] Alexej Klushyn, Richard Kurle, Maximilian Soelch, Botond Cseke, and Patrick van der Smagt. Latent
> matters: Learning deep state-space models. Advances in Neural Information Processing Systems,
> 34, 2021.*

---

### Official Review · Reviewer_dayt · 2022-10-27

**Confidence:** 3
**Correctness:** 3
**Technical Novelty And Significance:** 3
**Empirical Novelty And Significance:** 2
**Recommendation:** 3

**Clarity, Quality, Novelty And Reproducibility:**

The clarity needs to be much improved. It is not clear which problem the authors are focusing on since they have plugged together a 'high-dimensional, lack of dynamics' study and a 'low-dimensional, with the presence of dynamics' study. There needs to be some structure to understand why these two parts are considered in the same framework.

The ideas in the paper seem promising, but there may be limitations due to the structure of their proposed model (e.g., linear dynamics). These need to be stated clearly.

**Strength And Weaknesses:**

1. For many case studies, the authors do not estimate the latents but assume the latent form. In a typical dynamics and latent estimation study, we are not provided the latent form (for example, position and velocity in the Michigan NCLT dataset). What happens if the authors want to estimate the latents as well as the dynamics?
2. The paper sets out to perform system identification and learn the parameters of the state space model, but the authors only compare the predicted / estimated states with the ground truth. Certain properties of the recovered matrices can be compared with ground truth, for example, their eigenvalues. The authors need to show that their methods can recover ground truth dynamics.
3. The paper lacks motivation for the architecture of GIN. For example, is the goal of the encoder and decoder simply to reduce dimensionality? What if the model does not have an encoder and decoder, for example in the case of a low-dimensional input (the authors state that they use an MLP in this case, but is this necessary)?
4. The data that the authors used to perform the imputation task is simple although it outperforms other methods shown in the paper, the data has a high amount of temporal correlations, so it may not be hard to achieve an imputation task. Imputation in a complex real-world dataset would be more helpful to show the ability of GIN.
5. The “lack of dynamics” needs to be explained more clearly: it is not clear whether the dynamics are not being inferred or there is no succinct generative dynamical model for the simulations.

Minor comments:
1. The notation of x_{t+} and x_{t-} appear on page 3 first, but their definitions are in the caption of Figure 3 which is later shown.
2. The introduction does not clearly state the problem that the authors are considering. There are many typos and colloquial sentences, e.g., “a bunch of approaches”.


**Summary Of The Paper:**

The authors proposed a novel Gated Inference Network (GIN) to infer and learn state space models in both high and low dimensions. They use concepts in Kalman filtering to obtain estimates of the dynamics. They show that GIN is better at learning state space representations with disentangled dynamics features than existing approaches, they are also robust with noise and able to impute missing data. GIN is efficient in handling the case of unknown dynamics, the models are learned directly from the observations which are fed into encoders to obtain latent observations.

**Summary Of The Review:**

Due to the lack of clarity and the weaknesses mentioned above, I am not recommending acceptance in the current state of the submission.

---

> ### Author Response · Authors · 2022-11-15
> **Reply-1**
>
> We would like to thank you and appreciate your clear review and invested time.  In the following, we will address each question separately:
>
> *For many case studies, the authors do not estimate the latents ...*
>
> **R1**
>
> For more clarity we have provided notation information in the beginning of 'Parameterization' section. We use $o_{1:T}$, as observations, $w_{1:T}$ as the transferred observations, $x_{1:T}$ as the states and $(F_{1:T},H_{1:T})$ as the dynamics of the system.
> In all of the experiments, we perform filtering-smoothing parameterization on the states, $x_{1:T}$, for the sake of state estimation. In the first three experiment we assume the dynamics, $(F_{1:T},H_{1:T})$, are not known and they are going to be estimated and inferred as well as the states. As you stated, we are not provided with the latent form in the first three experiments because neither we know the dynamics nor the latent state form (for example, for the single pendulum experiment we are just given the high dimensional observations $o_{1:T}$, but we do not know that the state size is 4 which includes $(x,y,\dot{x},\dot{y})$ position and velocity in each axis and the dynamics for state transition and emission).
>
> While in the last two experiments, the dynamics $(F_{1:T},H_{1:T})$ are already known (please see the dynamics equations (37)-(39) in appendix) and accordingly considering latent form and size (for example position and velocity in NCLT dataset experiment) could be reasonable for the state estimation.
>
> *The paper sets out to perform system identification and learn*
>
> **R2**
>
> - For the high dimensional experiments, as we are not provided with the latents and dynamics form, the size of the state space and dynamics are hyper-parameters. For example, by referring to the tables 1 and 2, the size of the latent state is defined by $n$. Likewise for the learned dynamics because we are trying to learn the high dimensional dynamics. Afterwards, the high dimensional estimated states with the size of $n$ are transferred to low dimensional states (4 in the case of single pendulum) and the log likelihood is obtained. Because of such choice, showing the eigenvalues of the high dimensional learned dynamics, for example $\mathbf{F} \in \mathbb{R}^{n \times n}$, and comparing it with the ground truth dynamics $\mathbf{F} \in \mathbb{R}^{4 \times 4}$ may not be intuitive. However, to check whether the dynamics are learned appropriately or not for high dimensional observations, we provided (i) consecutive uninformed images imputation (figure 5) and highly distorted consecutive frames reconstruction (figures 13 and 15)
>
> - For the low dimensional experiments, we set the size of the state and ground truth same to each other (4 in the NCLT dataset experiment and 3 in the Lorenz attractor experiment). However, in these two experiments, we know the dynamics (Eq (37)-(39)) and do not learn them. But we attach additional results to address your concern:
>     As mentioned in the introduction, the GIN can handle high-low dimensional observations with lack-presence of the dynamics. Accordingly, we can conduct our two low dimensional experiments without the presence of the dynamics(Eq (37)-(39)) and learn the dynamics as well as the states. In such cases, comparing the learned dynamics and ground truth dynamics is feasible because the size of the learned dynamics are same with the ground truth. The comparison between the learned dynamics and ground truth are attached in the appendix (because of the large numbers of figures and results and also the page limit) figures [40-46], where we compare the eigenvalues of the learned dynamics with those of the ground truth. (Please note that the implementation of such experiments are already provided in the github repo in the submission time)
>
> *The data that the authors used to perform the imputation...*
>
> **R3**
>
> We agree with you that the imputation tasks in the paper are simple but there are two reasons for this choice:
>
> - Comparison with the SOTA: same imputation task is conducted in [1], [2] and [5] (however we have to note that the experiment in [2] is bounding ball but the complexity of the dynamics and the frame size are almost similar.)
>
> - Modeling higher dimensional observation with additional noise distortion and complexities are conducted in the visual odometry experiment where the observations (1241$\times$376$\times$3) are comparatively larger than single-double pendulum experiment. In this experiment, we achieved competitive results with the SOTA for inferring the states meaning that the model has the ability of learning the dynamics. However, for the reconstruction of the high dimensional observations, i.e. when ${o}_{1:T}^+$ are high dimensional images, we need better choice of $e(.)$ and $d(.)$ which is separated from inferring and learning the states and dynamics. We can put this as one of the potential extensions and future works of the current version.

---

> ### Author Response · Authors · 2022-11-15
> **Reply-2**
>
> *The paper lacks motivation for the architecture of GIN...*
>
> **R4**
>
> In our overall structure, i.e. figure 2, the role of $e(.)$ function is to provide $w_{1:T}$ and observation noise $R_{1:T}$ for the transition block, while the role of $d(.)$ function is to output the states and its covariance, i.e. $o_{1:T}^+$.
> For the high dimensional experiments $e(.)$ function, (i) reduces the dimensionality of the original observations $o_{1:T}$ to the transferred observations $w_{1:T}$. (ii) Also it constructs the the transferred observation noise $R_{1:T}$ with some positive activation function. $d(.)$ outputs the mean vector and covariance matrix of $o_{1:T}^+$ (please see figure 10). The same structure with similar purpose is employed in [1], [2] and [3].
> While for the lower dimension observations, we do not need to reduce the dimensionality of the original observations $o_{1:T}$. In other words $o_{1:T} = w_{1:T}$, however we still need to model the observation noise $R_{1:T}$ that we use MLP for this sake (please see figure 11). Similar approach is used in [4].
> Overally, as stated in the introduction, with a simple adjustment in the structure of $e(.)$ and $d(.)$, the model can handle a wide range of applications and provide competitive results with the SOTA in each framework, i.e. high-low dimensional observations with lack-presence of the dynamics.
>
> *The “lack of dynamics” needs to be explained more clearly...*
>
> **R5**
>
> We modified the introduction and add a notation section in which we have addressed your concern. Here is what we have added: "We define the transition matrices $F_{1:T}$ and emission matrices $H_{1:T}$ as the dynamics of the model, where lack of dynamics in the first three experiments means that ($F_{1:T}, H_{1:T}$) are not know and are going to be trained(graphical model is in figure 3), while the presence of dynamics in our last two experiments means that ($F_{1:T}, H_{1:T}$) are known(graphical models are in figures 9)."
>
>
> *The notation of $x_{t+}$ and $x_{t-}$ appear on page 3 first...*
>
> **R6**
>
> We addressed this issue in the notation in the beginning of the parameterization section.
>
> *The clarity needs to be much improved. It is not clear which problem the authors are focusing...*
>
> **R7**
>
> We modify the contributions (i) and (ii) in the introduction section to point to the problem better. Moreover, we clarify the terms more in the provided notation to give the readers better intuition about what 'lack-presence' of dynamics means. One of the reasons we include high-low dimensional experiments is that to show by a simple adjustment of the non-linear transferring function $e(.)$ and $d(.)$, we can handle the both cases. While, the ability of learning the dynamics (in the lack of them) and using the dynamics (in the presence of them), makes the GIN applicable to a wide range of applications. For example, it can be used for high dimensional images and video frames where we are not aware of the underlying dynamics of the system. It can be employed for the forecasting seasonal-periodic time series data, where the underlying dynamics are not known as well, e.g. electricity and traffic dataset [6] or results in figures 40-46. It can be used when the dynamics are available, e.g. low dimensional experiments in the main paper. Where, based on the numerical results, it can provide competitive performance compare to the literature of each framework.
>
>
> *[1] Philipp Becker, Harit Pandya, Gregor Gebhardt, Cheng Zhao, C James Taylor, and Gerhard Neumann.
>     Recurrent kalman networks: Factorized inference in high-dimensional deep feature spaces. In
>     International Conference on Machine Learning, pp. 544–552. PMLR, 2019,*
>
>
> *[2] Marco Fraccaro, Simon Kamronn, Ulrich Paquet, Ole Winther. A Disentangled Recognition and Nonlinear Dynamics Model for Unsupervised Learning. Part of Advances in Neural Information Processing Systems 30 (NIPS 2017).*
>
> *[3] Yuri Burda, Roger Grosse, Ruslan Salakhutdinov. Importance Weighted Autoencoders. Submitted to ICLR 2015.*
>
> *[4] Guy Revach, Nir Shlezinger, Xiaoyong Ni, Adria Lopez Escoriza, Ruud JG van Sloun, and Yonina C Eldar. Kalmannet: Neural network aided kalman filtering for partially known dynamics.*
>
> *[5] Alexej Klushyn, Richard Kurle, Maximilian Soelch, Botond Cseke, and Patrick van der Smagt. Latent
> matters: Learning deep state-space models. Advances in Neural Information Processing Systems,
> 34, 2021.*
>
> *[6] Hsiang-Fu Yu, Nikhil Rao, and Inderjit S Dhillon. Temporal regularized matrix factorization for high-dimensional time series prediction. In D. D. Lee, M. Sugiyama, U. V. Luxburg, I. Guyon, and R. Garnett, editors, Advances in Neural Information Processing Systems 29, pages 847–855. Curran Associates, Inc., 2016.*

---

### Author Response · Authors · 2022-11-15
**Revision**

We would like to thank all of the reviewers. We provide the revision based on the comments pointed by the reviewers, where the summary of the changes are as follow:

1- Introduction and contributions are modified to improve the clarity and show what kind of problems we are focusing on and why those problems are important.

2- Couples of related works that were not covered in first version, are added now. The comparison table 9 is updated as well.

3- Reference to the background materials, e.g. classic Kalman filter-smooth, is added for the completeness and the notations and variables with their naming are defined in the beginning of parameterization section.

4- Graphical model for both high and low dimensional with explanations are added in figures 3 and 9.

5- A comparison between the learned dynamics and ground truth dynamics are added in figures 40-46

6- Learned KG-SG matrices are compared with their corresponding ground truths for low dimensional experiments in table 11.

---

### Decision · Program_Chairs · 2023-01-20

**Decision:**

Reject

**Justification For Why Not Higher Score:**

Getting the paper ready for publication requires substantial updates and therefore also a new round of reviews.

**Justification For Why Not Lower Score:**

N/A

**Metareview: Summary, Strengths And Weaknesses:**

This paper proposes a state space model approach to time-series that uses a GRU network to learn the Kalman gain rather than using potentially unstable matrix inversions.

The numerical results appear convincing, however, the reviewers have reservations mainly with regarding to have the paper is presented. This has prompted the reviewers to propose significant reorganisation of the paper.

The paper definitely merits publication but it is recommended that it is rejected for this conference giving the authors time to properly update the paper for the next.